# An assessment of basal melt parameterisations for Antarctic ice shelves

Clara Burgard[1], Nicolas C. Jourdain[1], Ronja Reese[2], Adrian Jenkins[2], and Pierre Mathiot[1]

[1]Univ. Grenoble Alpes, CNRS, IRD, Grenoble INP, IGE, 38000 Grenoble, France
[2]Department of Geography and Environmental Sciences, Northumbria University, Newcastle Upon Tyne, UK

**Correspondence:** C. Burgard (clara.burgard@univ-grenoble-alpes.fr)

**Abstract.** Ocean-induced ice-shelf melt is one of the largest uncertainty factors in the Antarctic contribution to future sea-level rise. Several parameterisations exist, linking oceanic properties in front of the ice shelf to melt at the base of the ice shelf, to force ice-sheet models. Here, we assess the potential of a range of these existing basal melt parameterisations to emulate basal melt rates simulated by a cavity-resolving ocean model on the circum-Antarctic scale. To do so, we perform two cross-validations, over time and over ice shelves respectively, and re-tune the parameterisations in a perfect model approach, to compare the melt rates produced by the newly tuned parameterisations to the melt rates simulated by the ocean model. We find that the quadratic dependence of melt to thermal forcing without dependency on the individual ice-shelf slope and the plume parameterisation yield the best compromise, in terms of integrated shelf melt and spatial patterns. The box parameterisation, which separates the sub-shelf circulation into boxes, the PICOP parameterisation, which combines the box and plume parameterisation, and quadratic parameterisations with dependency on the ice slope yield basal melt rates further from the model reference. The linear parameterisation cannot be recommended as the resulting integrated ice-shelf melt is comparably furthest from the reference. When using offshore hydrographic input fields in comparison to properties on the continental shelf, all parameterisations perform worse, however the box and the slope-dependent quadratic parameterisations yield the comparably best results. In addition to the new tuning, we provide uncertainty estimates for the tuned parameters.

## 1 Introduction

The Antarctic ice sheet has been losing mass at a rapid pace in past decades, increasing the Antarctic contribution to sea-level rise from 0.14$\pm$0.02 mm yr$^{-1}$ between 1992 and 2001 to 0.55$\pm$0.07 mm yr$^{-1}$ between 2012 and 2016 (Oppenheimer et al., 2019). Most of this mass loss has been attributed to an acceleration in ice flow across the grounding line, i.e. from the grounded part to the floating ice shelves at the outskirts of the ice sheet (e.g. Mouginot et al., 2014; Rignot et al., 2014; Scheuchl et al., 2016; Khazendar et al., 2016; Shen et al., 2018; The IMBIE Team, 2018).

Ice shelves themselves can moderate the pace of the mass loss. Being several hundreds of meters thick and locally constrained by land or pinning points, they act as natural barriers to restrain the grounded ice-sheet flow into the ocean. Ice shelves have been thinning all around Antarctica in past decades (Rignot et al., 2013; Paolo et al., 2015; Adusumilli et al., 2020), driven by an increasing amount of warm circumpolar deep water (CDW) intruding on to the continental shelf and into the cavities below

the ice shelves (Jacobs et al., 2011; Wouters et al., 2015; Khazendar et al., 2016; Jenkins et al., 2018). Thinning reduces the ice shelves' buttressing potential, which means that the restraining force that they exert on the ice outflow at the grounding line is lower and more ice is discharged into the ocean. In some bedrock configurations, increased melt can trigger marine ice sheet instabilities (e.g. Weertman, 1974; Schoof, 2007; Gudmundsson et al., 2012). This is why ocean-induced sub-shelf melt, which we call *basal melt* in the following, is a crucial component for simulations of the Antarctic contribution to future sea-level

evolution. Still, it is currently one of the main sources of uncertainty in such projections (e.g. Edwards and the ISMIP6 Team, 2021; Hill et al., 2021).

Basal melt is a result of positive thermal forcing, i.e. water above the local freezing point getting in contact with the lower side of the ice shelf. To represent basal melting accurately in models, we therefore need to accurately simulate the hydrographic properties of the water entering the ice-shelf cavity and to resolve the circulation of the water masses within the cavity.

Ideally, this would be done in a coupled ocean–ice-sheet simulation resolving the ocean circulation in the cavity below the ice shelf (e.g. De Rydt and Gudmundsson, 2016; Seroussi et al., 2017). However, running such simulations on a circum-Antarctic scale is computationally expensive and this approach is therefore currently not suitable for large ensembles or multi-centennial time scales. Furthermore, most global climate models, such as the ones used in the most recent phases of the Coupled Model Intercomparison Project (CMIP, Taylor et al., 2012; Eyring et al., 2016), still poorly represent the ocean dynamics along the

Antarctic margins and do not include ice-shelf cavities (Beadling et al., 2020; Heuzé, 2021). As a consequence, the Antarctic contribution to sea-level rise is often computed by standalone ice-sheet models or ice-sheet models coupled to a coarse-resolution ocean model, which can be called "coupling of intermediate complexity" (Kreuzer et al., 2021). In both cases, basal melting is parameterised based on ocean properties simulated by the ocean model for the region in front of the ice shelf (e.g. Jourdain et al., 2020; Reese et al., 2020).

On the one hand, ice-sheet models need information about the spatial distribution of melt below the ice shelf. On the other hand, non-cavity-resolving ocean models only provide the hydrographic properties in front of the ice shelf ("far field" in the following). Several parameterisations of varying complexity have been developed in the last 20 years to derive melt rates from far-field ocean properties on the circum-Antarctic scales (Beckmann and Goosse, 2003; Holland et al., 2008; DeConto and Pollard, 2016; Reese et al., 2018a; Lazeroms et al., 2018, 2019; Favier et al., 2019; Jourdain et al., 2020). However, assumptions

in the various formulations differ, giving rise to a large variety of melt patterns (Favier et al., 2019). As observations of the hydrographic properties in front of ice shelves are sparse, it is challenging to evaluate the performance and uncertainty of the different basal melt parameterisations and therefore make a recommendation on which one to incorporate in standalone ice-sheet models or ocean–ice-sheet couplings of intermediate complexity.

Favier et al. (2019) evaluated various melt parameterisations through a comparison between standalone ice-sheet simulations

with parameterised melt and a small ensemble of coupled ice-sheet–ocean simulations (resolving the ocean circulation and melt beneath the ice shelf). This work was based on an idealised modelling setup consisting of an evolving but relatively small cavity, with idealised cooling and warming transitions similar to the MISMIP+ and MISOMIP framework (Asay-Davis et al., 2016). Their parameterisation ranking was then used as a basis to choose the standard melt parameterisation of ISMIP6 (Jourdain et al., 2020; Nowicki et al., 2020). However, when these recommendations were applied to ice-sheet models with realistic and

diverse ice-shelf geometries, substantial empirical temperature corrections had to be applied to reproduce observational melt rates in the various sectors of Antarctica, and the pattern and sensitivity to ocean warming were still questionable (Jourdain et al., 2020; Reese et al., 2020). Previous studies also had to apply sector-dependent corrections or calibrations (Lazeroms et al., 2018; DeConto and Pollard, 2016).

In this study, we assess the potential of the diverse parameterisations to represent melt rates without basin- or ice-shelf-dependent temperature correction or calibration. To do so, we explore their ability to emulate an ensemble of circum-Antarctic ice-shelf-resolving ocean simulations representing a total of 127 years of basal melt rates responding to a variety of ocean conditions. This assessment is particularly relevant for the application of the parameterisations in pan-Antarctic ice-sheet simulations. In Sec. 2, we describe the ensemble of ocean simulations that we use as our "virtual" reality, the data, and the different parameterisations we assess. We also revisit the formulation of several simple parameterisations to emphasize the physical hypotheses behind them. In Sec. 3, we conduct cross-validations to assess how the resulting melt rates compare between parameterisations and how they compare with the melt rates simulated by the ocean model, and we propose newly tuned "best-estimate" parameters. In Sec. 4, we investigate uncertainties around the parameters and discuss recommendations and limitations for applications in pan-Antarctic ice-sheet simulations.

## 2 Evaluation framework

We use a perfect-model approach to assess and re-tune different basal melt parameterisations proposed by previous literature, from simple to more complex ones. This means that we use the ocean state and melt rates simulated by a cavity-resolving ocean model as a "virtual" reality (Fig. 1). There are several advantages to this method. First, we have a larger amount of data, both over time and space, than we have from observations. Second, a model provides a self-consistent framework where the ocean properties in front of the ice shelf, which we feed into the different parameterisations, perfectly match the melt rates at the base of the ice shelf. This perfect link is currently not achievable with observational estimates, except at a few specific locations. In the following we present the ocean model and its configuration, the basal melt parameterisations, and the tuning and evaluation method. Note that the perfect model approach relies on the assumption that NEMO results in a realistic approximation of the circulation in the ice-shelf cavity and melt behaviour at the ice-ocean interface. We discuss the limitations of this assumption in Sec. 4.1.1.

### 2.1 The ocean model NEMO

#### 2.1.1 Basic model setup

Our study is based on simulations conducted with the version 4.0.4 of the 3-D primitive-equation coupled ocean–sea-ice model NEMO (Nucleus for European Modelling of the Ocean, NEMO Team, 2019). It is run in a global configuration referred to as eORCA025 (Storkey et al., 2018), which is a grid of 0.25° resolution in longitude, i.e. a resolution of 8 km in both directions at 70°S, which is sufficient to capture the basic ocean circulation below multiple Antarctic ice shelves (Mathiot et al., 2017;

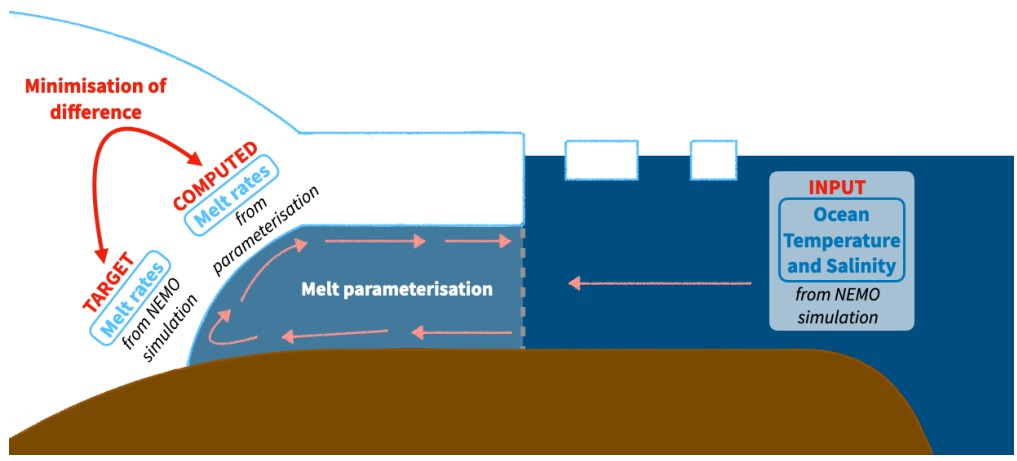

**Figure 1.** Schematic of the perfect model approach to assess the different parameterisations.

Bull et al., 2021). The nonlinear free surface is defined through the time varying $z^\star$ vertical coordinate (Adcroft and Campin, 2004). A new vertical grid was developed, with 121 vertical levels (vs 75 commonly used) and a depth-dependent resolution of 1 m in the surface layers, 20 m between 100 m and 1000 m depth and 200 m at 6000 m depth. As most ice shelves have their grounding lines above or near 1000 m depth and their front below 100 m, this enables a quasi uniform vertical resolution across the Antarctic ice shelves.

In this version of NEMO, the SI3 model represents sea-ice dynamics, thermodynamics, brine inclusions and subgrid-scale sea-ice thickness variations (NEMO Sea Ice Working Group, 2019). Also, we use the Lagrangian iceberg model developed by Marsh et al. (2015) and improved by Merino et al. (2016) to account for sub-surface currents and temperatures. The Antarctic calving fluxes are constant and based on the satellite estimates by Rignot et al. (2013). The observed fluxes are imposed at the front of individual ice shelves with a uniform random distribution for all grid cells of a given ice shelf front.

The basal melt rate of ice shelves is represented by the three equations as described in Asay-Davis et al. (2016) implemented into NEMO by Mathiot et al. (2017):

1. the heat balance at the ice-ocean interface:

$$c_{\mathrm{oc}}\rho_{\mathrm{oc}}\gamma_T(T_{\mathrm{oc}} - T_f) = -L_i f_w - \rho_i c_i \kappa \frac{T_a - T_{\mathrm{oc}}}{h_{\mathrm{isf}}} \tag{1}$$

2. the salt balance at the ice-ocean interface:

$$\rho_{\mathrm{oc}}\gamma_S(S_{\mathrm{oc}} - S_b) = -S_b f_w \tag{2}$$

3. the pressure and salinity-dependent freezing temperature:

$$T_f = \lambda_1 S_b + \lambda_2 + \lambda_3 z_{\mathrm{draft}} \tag{3}$$

where $c_{oc}$ and $c_i$ are, respectively, the specific heat capacity of the seawater and the ice, $\rho_{oc}$ and $\rho_i$ are, respectively, the density of the seawater and the ice, $T_{oc}$ and $S_{oc}$ are the temperature and salinity averaged over a boundary layer below the ice shelf, $T_f$ is the freezing point, $L_i$ is the latent heat of fusion, $f_w$ is the meltwater mass flux, $\kappa$ is the thermal diffusivity of ice, $T_a$ is the atmospheric surface temperature, here assumed constant at -20°C, $h_{isf}$ is the ice shelf thickness, $S_b$ is the salinity at the ice-ocean interface, $\lambda_1$, $\lambda_2$ and $\lambda_3$ are the coefficients for the freezing point equation (close to the ones listed in Table 2), $z_{draft}$ is the depth of the ice-shelf draft (negative below sea level), and $\gamma_T$ and $\gamma_S$ are the exchange velocities for temperature and salt:

$$\gamma_T = C_d^{1/2}\Gamma_T\sqrt{U^2 + U_{tide}^2} \tag{4}$$

$$\gamma_S = C_d^{1/2}\Gamma_S\sqrt{U^2 + U_{tide}^2} \tag{5}$$

where $U$ is the ocean velocity in the top boundary layer, $U_{tide}$ is the tidal velocity, $C_d$ is the ice-ocean drag coefficient, set to $2.5 \times 10^{-3}$, and $\Gamma_T$ and $\Gamma_S$ are the heat and salt exchange coefficients and are set respectively to 0.014 and $4 \times 10^{-4}$ as in Hausmann et al. (2020) and Bull et al. (2021). The top boundary layer is the layer over which temperature, salinity, and the ocean horizontal and vertical velocity components $u_{oc}$ and $v_{oc}$ (where $U = \sqrt{u_{oc}^2 + v_{oc}^2}$) are averaged. Its thickness is set to 20 m.

In contrast to Mathiot et al. (2017), we prescribe a constant but spatially varying tidal velocity $U_{tide}$ in Eq. (4) and Eq. (5) at the upper interface (ice shelf/ocean). Following the recommendations of Jourdain et al. (2019), it is calculated as 0.656 times the mean barotropic tidal velocity derived from constituents M2, S2, N2, K1, Q1 and O1 of the CATS 2008 model (Padman et al., 2008; Howard et al., 2019) using Eq. (7) of Jourdain et al. (2019). While the conclusions of Jourdain et al. (2019) were limited to the Amundsen Sea sector, more recent work gives confidence that this method to represent tide-induced melt is relevant at the circum-Antarctic scale (Hausmann et al., 2020; Huot et al., 2021; Richter et al., 2020).

The model bathymetry is derived from ETOPO1 in the open ocean (Amante and Eakins, 2009), GEBCO (IHO and BODC IOC, 2003) on the continental shelves (excluding Antarctic continental shelf). The Antarctic continental shelf bathymetry and ice shelf draft are based on Bedmachine Antarctica version 2 (Morlighem, 2020; Morlighem et al., 2020). The simulations are forced with the climatological geothermal heat flux from Goutorbe et al. (2011), and atmospheric forcing from JRA55-do version 1.4 (Tsujino et al., 2018). The turbulent and momentum fluxes are computed using the NCAR bulk formulae algorithm (Large and Yeager, 2009). The ocean initial conditions are based on the WOA2018 climatological temperature and salinity over 1981-2010 (Locarnini et al., 2018; Zweng et al., 2018; Garcia et al., 2019). The sea-ice initial conditions are taken from a 1980-2004 model climatology based on an upgrade of eORCA025 GO6 simulations (Storkey et al., 2018). Initial sea-ice and ocean velocities are set to 0.

More details on the parameterisations common to all our simulations are presented in Appendix A.

## 2.1.2 Ensemble of simulations

We use an ensemble of four NEMO simulations that are based on different values of a small number of parameters. This is an ensemble of opportunity, i.e. not specifically designed for this study, but it covers large ocean temperature variations, with differences of up to $\pm 2°C$ in front of some ice shelves (Fig. 2), which is an amplitude comparable to typical RCP8.5 changes during the 21st century (Barthel et al., 2020). Also, the different simulations cover several decades, and therefore include interannual variability, which can affect basal melt rates (Hoffman et al., 2019).

The differences between the four simulations of the ensemble are listed in Table 1. None of the changed parameters has a significant impact on the physics of ocean—ice-shelf interactions, and they mostly change the physical ocean properties outside the cavities. For example, for all simulations except HIGHGETZ, we assigned a land barrier along the 350 m isobath of Bear Ridge East Flank in the Amundsen Sea to mimic the sea-ice blocking induced by grounded icebergs in that region, like in Bett et al. (2020) and Nakayama et al. (2014).

**Table 1.** Description of the differences between the four simulations used. The AABW (AntArctic Bottom Water) restoring is described in Dufour et al. (2012). GM mixing is a parameterisation of adiabatic eddy mixing (Gent and McWilliams, 1990) used where local resolution is coarser than half the local Rossby radius. S2016 and M2015 are the iceberg size distribution provided by Stern et al. (2016) and Marsh et al. (2015), respectively. The land barrier at Bear Ridge mimics the sea-ice blocking induced by grounded icebergs in that region. Ice set 1 and Ice set 2 make use of different sea ice parameters. In Ice-set 2 (compared to Ice-set 1), the adaptive EVP rheology is turned off, the ice–ocean drag coefficient is set to 0.005 instead of 0.012, the snow conductivity changed from 0.31 to 0.35 W/m/K, and the frazil ice formation scheme is turned off. "Bug 2626" stands for a bug on the distribution of solar and non-solar radiation originally present in NEMO-v4.0.4 and corrected since then.

| Simulation | Simulation period (Analysis period) | Isopycnal diffusion [m²/s] | Eddy-induced velocity coefficient [m²/s] | GM mixing | AABW restoring | Iceberg distribution | Barrier Bear Ridge | Sea ice parameters | Bug 2626 | Getz geometry |
|---|---|---|---|---|---|---|---|---|---|---|
| REALISTIC | 1979-2018 (1989-2018) | 150 | 150 | Yes | No | S2016 | Yes | Ice set 2 | No | Shallow |
| COLDAMU | 1959-2008 (1980-2008) | 150 | n/a | No | No | M2015 | Yes | Ice set 1 | No | Shallow |
| WARMROSS | 1959-2008 (1980-2008) | 300 | 300 | Yes | Yes | M2015 | Yes | Ice set 1 | No | Shallow |
| HIGHGETZ | 1959-2018 (1980-2018) | 150 | n/a | No | No | M2015 | No | Ice set 1 | Yes | Realistic |

The only difference affecting ocean–ice-shelf interactions is the ice-shelf topography of Getz. The total integrated basal melt rate of Getz in the HIGHGETZ simulation reaches 500 Gt/yr due to an underestimated thermocline depth that allows

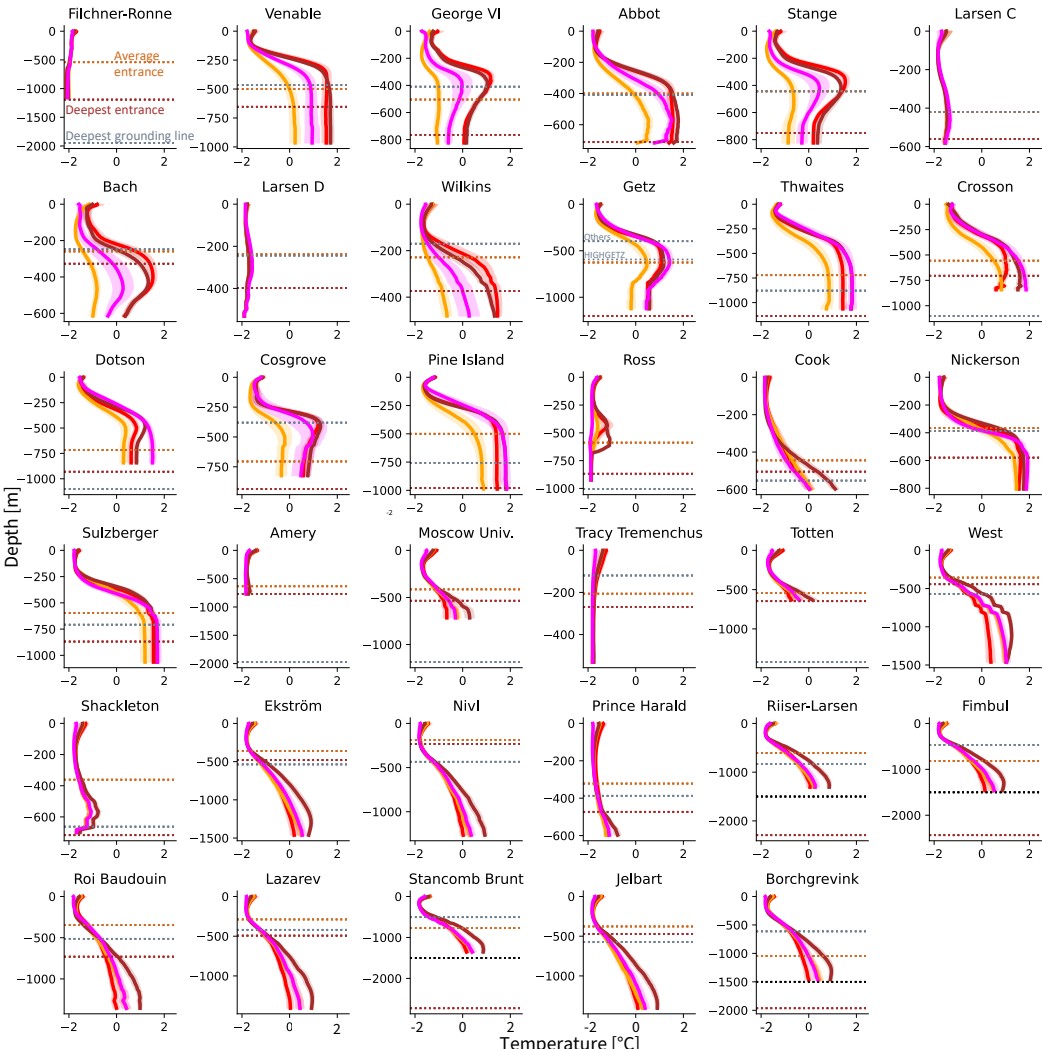

Averaged temperature profiles over 50 km in front of the ice shelf for the different simulations of the ensemble

**Figure 2.** Comparison of mean input temperature profiles between the four different simulations (REALISTIC, COLDAMU, WARMROSS, HIGHGETZ), averaged over 50 km on the continental shelf in front of the ice shelf. The shading represents the interannual variability (one standard deviation over time). The horizontal dotted lines show the average depth of the bed at the ice front (light brown), the largest depth of the bed at the ice front (dark brown) and the largest depth of the grounding line (grey). The deepest grounding line point for Getz ice shelf is different in the HIGHGETZ compared to the other simulations due to the artificial thinning in the latter (see Sec. 2.1.2). For a few ice shelves, the deepest entrance is deeper than 1500 m depth but the profiles are defined over the continental shelf (depth shallower than -1500 m). A black dotted line denotes the 1500 m limit in this case. For Stange and Larsen C ice shelves, the average entrance depth and the deepest grounding line overlap.

circumpolar deep water to reach most of the ice shelf draft. This is a long-standing bias in our NEMO simulations (Mathiot et al., 2017; Jourdain et al., 2017). As a consequence, the Getz ice shelf was artificially thinned by $\sim 200$ m in all other simulations of our ensemble. This correction reduces the integrated basal melt rate to $\sim 170$ Gt/yr (see Fig. E1), closer to observational estimates (Rignot et al., 2013; Adusumilli et al., 2020).

As suggested by the simulation names, REALISTIC is the simulation closest to realistic conditions, COLDAMU remains relatively cold in the Amundsen Sea, and WARMROSS triggers a warm state over the East Ross Shelf and much fresher high salinity shelf water (HSSW). More details, including an evaluation of the REALISTIC simulation compared to observations, are provided in Appendix B.

For this study, we start by interpolating the NEMO output bilinearly to a stereographic grid of 5 km spacing as all parameterisations are coded for stereographic grids, which are commonly used for ice sheet models. All pre-processing and analysis is conducted using this regridded data. In a second step, we cut out the different ice shelves according to longitude and latitude boundaries (details found in Burgard (2022)). In a third step, we only keep the largest ice shelves. The effective resolution of physical ocean models, i.e. the resolution below which the circulation might not be resolved well, is typically 5 to 10 times the grid spacing (Bricaud et al., 2020). We empirically choose a cutoff at an area of 2500 km$^2$ (i.e. 6.25 $\Delta x$) to be in this range while keeping a sufficiently large number of ice shelves. This results into the 35 resolved ice shelves listed, e.g., in Fig. 2.

## 2.2 Basal melt parameterisations

We make the choice to assess parameterisations which have been developed for and applied on the circum-Antarctic scale. These are the simple linear (Beckmann and Goosse, 2003) and quadratic (DeConto and Pollard, 2016; Holland et al., 2008; Favier et al., 2019; Jourdain et al., 2020) parameterisations, the plume parameterisation (Lazeroms et al., 2018, 2019), the box parameterisation (Reese et al., 2018a) and the PICOP parameterisation (Pelle et al., 2019). There are also other parameterisations, but these have been only applied to single ice shelves. For example, Hoffman et al. (2019) propose a parameterisation combining an approximate solution of the plume equations to infer the temperature with the local/non-local formulation from Favier et al. (2019) and apply it to investigate variability in the melt of Thwaites ice shelf.

In all the parameterisations, the main driver of basal melt is the thermal forcing. However, the various parameterisations differ in their definitions of the far-field temperature and salinity, in the link between these and the thermal forcing at the ice-ocean interface, and in the complexity of the physical relationship between the thermal forcing and the melt rate. In the following, we present the different parameterisations. As a range of slightly different definitions of the variables are introduced, we provide two tables in Appendix C summarizing the main variables and different subscripts used in the following description. All implementations were done in Python (see Burgard, 2022), mainly with the numpy (Harris et al., 2020), xarray (Hoyer and Hamman, 2017) and dask (Dask Development Team, 2016) packages.

### 2.2.1 Simple parameterisations

Over the past decades, several simple parameterisations of the link between the ocean properties in front of the ice shelf and the melt at the base of ice shelves have been proposed. While some earlier simple parameterisations were based on sea-floor

or vertically integrated ocean properties, recent results point to the importance of the thermocline depth for melt rates (e.g. De Rydt et al., 2014; Favier et al., 2019). This is why we account for vertical profiles of the input properties in the following.

In the simple parameterisations, the water temperature and salinity at a given point of the ice-shelf draft are extrapolated from the same depth in the mean profiles in front of the ice shelf, which we call "far-field" temperature and salinity in the
190 following. If the ice-shelf draft is deeper than the deepest entrance point, i.e. the deepest point of the bed at the ice-shelf front (brown line in Fig. 2 and Fig. 3), we use the hydrographic properties from the mean profiles at the depth of the deepest entrance point. If both the ice-shelf draft and the deepest entrance are deeper than 1500 m (black line in Fig. 2 and Fig. 3), we use the hydrographic properties at the deepest point of the mean profiles. This extrapolation means that the thermal forcing is directly linked to the ocean properties in front of the shelf and the redistribution and transformation of water masses within the cavity
is not accounted for in most of them.

Several slightly different versions of the simple parameterisations have been formulated in past decades. To enable a consistent assessment and comparison, we start by revisiting these formulations in a common formalism.

Locally, the basal melt rate $m$ [in meters of ice per second] is determined by the heat flux across the turbulent boundary layer created by current shear against the ice base, and was formulated as follows in Jenkins et al. (2018) based on a linearisation of
200 the three equations (Jenkins et al., 2010):

$$m = \frac{\rho_{\text{oc}}}{\rho_{\text{i}}} \frac{C_d^{1/2} \Gamma_{TS}}{[L_i - c_i(T_i - T_{\text{f, loc}})]/c_{\text{oc}}} U(T_{\text{loc}} - T_{\text{f, loc}}) \tag{6}$$

where $\Gamma_{TS}$ is a transfer coefficient combining information about heat and salt transfer, $T_i$ and $T_{\text{loc}}$ are, respectively, the temperature of the ice and the far-field ocean temperature extrapolated to the local depth of the ice shelf draft $z_{\text{draft}}$, and $T_{\text{f, loc}}$ is the local freezing temperature computed using the $z_{\text{draft}}$ and $S_{\text{loc}}$, the far-field ocean salinity extrapolated to $z_{\text{draft}}$, in Eq. (3).
Given that ice temperature has a relatively small effect on melt rates (Dinniman et al., 2016; Arzeno et al., 2014), we can rewrite Eq. (6) as:

$$m = \frac{c_{\text{oc}}}{L_i} \frac{\rho_{\text{oc}}}{\rho_{\text{i}}} \gamma_{TS}(T_{\text{loc}} - T_{\text{f, loc}}) \tag{7}$$

with the turbulent exchange velocity:

$$\gamma_{TS} = C_d^{1/2} \Gamma_{TS} U \tag{8}$$

The simplest way to estimate melt rates from Eq. (7) is to assume a constant and uniform characteristic velocity ($U_{\text{Ant}}$) over all ice shelves, so that $\gamma_{\text{TS}}$ is a constant parameter that can be tuned to match observations or simulations:

$$\gamma_{TS,\text{loc, Ant}} = C_d^{1/2} \Gamma_{TS} U_{\text{Ant}} \tag{9}$$

The resulting parameterisation, referred to as the *linear-local* parameterisation, was initially proposed by Beckmann and Goosse (2003) and has been used in numerous ice-sheet simulations since then (e.g. Martin et al., 2011; Álvarez-Solas et al.,
2011).

However, assuming a constant $U_{\text{Ant}}$ over all Antarctica is not necessarily reasonable as velocities vary widely within an ice-shelf cavity and from one ice shelf to another (Jourdain et al., 2017). Since the introduction of the *linear-local* parameterisation, efforts have been made to include more information that could mimic the effect of the local geometry and of the vertical overturning circulation in the cavity on the melt rate. Jenkins et al. (2018), for example, propose to describe $U$ with other variables and parameters. This formulation assumes that the ocean circulation along the ice draft is mostly governed by the geostrophic balance:

$$U = \frac{g}{-2 \mid f \mid} \sin\theta \Delta\widetilde{\rho} \tag{10}$$

where $g$ is gravity, $f$ is the Coriolis parameter, $\theta$ is the slope of the ice shelf base relative to the horizontal, and $\Delta\widetilde{\rho}$ the dimensionless density deficit in the top boundary layer:

$$\Delta\widetilde{\rho} = P_0(\epsilon - 1) \mid T_{\text{loc}} - T_{\text{f, loc}} \mid \tag{11}$$

where the subscript *loc* denotes the far-field temperature and salinity extrapolated to the local ice-draft depth, $\epsilon$ can be related to the relative efficiency of mixing across the thermocline that separates the boundary current from the warmer water below and across the ice-ocean boundary layer, and $P_0$ is a constant defined as follows (Jenkins et al., 2018):

$$P_0 = \frac{\beta_S S_{\text{loc}} - \beta_T [(T_{\text{loc}} - T_{\text{f, loc}}) + (L_i - c_i(T_i - T_{\text{f, loc}}))/c_{\text{oc}}]}{(T_{\text{loc}} - T_{\text{f, loc}}) + (L_i - c_i(T_i - T_{\text{f, loc}}))/c_{\text{oc}} - \lambda_1 S_{\text{loc}}} \tag{12}$$

where $\beta_S$ and $\beta_T$ are the salt contraction and heat expansion coefficients respectively (see Table 2).

Combining Eq. (6), Eq. (10), and Eq. (11), this results in the following formulation for the melt rate:

$$m = M_0 \frac{g}{2 \mid f \mid} \sin\theta P_0 \epsilon(1 - \epsilon) \mid T_{\text{loc}} - T_{\text{f, loc}} \mid (T_{\text{loc}} - T_{\text{f, loc}}) \tag{13}$$

where $M_0$ is the term in braces in Eq. (6), which can be simplified, like in Eq. (7), to: $M_0 \approx C_d^{1/2} \Gamma_{TS} \frac{\rho_{\text{oc}}}{\rho_{\text{i}}} \frac{c_{\text{oc}}}{L_i}$. Note that we use the absolute value of the thermal forcing in the term describing $U$ because we consider the current speed, which has to be positive.

After a scale analysis based on the values in Table 2, a reasonable approximation of $P_0$ is: $P_0 \approx \frac{c_{\text{oc}}}{L_i} \beta_S S_{\text{loc}}$, which yields the following formulation for the melt rate:

$$m = C_d^{1/2} \Gamma_{TS} \frac{\rho_{\text{oc}}}{\rho_{\text{i}}} \left(\frac{c_{\text{oc}}}{L_i}\right)^2 \beta_S S_{\text{loc}} \frac{g}{2 \mid f \mid} \sin\theta \epsilon(1 - \epsilon) \mid T_{\text{loc}} - T_{\text{f, loc}} \mid (T_{\text{loc}} - T_{\text{f, loc}}) \tag{14}$$

We recognise the quadratic dependence of the melt rate to thermal forcing as used in previous formulations (Holland et al., 2008; DeConto and Pollard, 2016). We call this parameterisation the *quadratic-local* parameterisation in the following.

If we compare this new formulation with Eq. (7), this means that the turbulent exchange velocity now depends on the location through the inclusion of the slope $\theta$ and on the far-field temperature $T_{\text{loc}}$ and salinity $S_{\text{loc}}$:

$$\gamma_{TS,\text{loc},\theta} = C_d^{1/2} \Gamma_{TS} \frac{c_{\text{oc}}}{L_i} \beta_S S_{\text{loc}} \frac{g}{2 \mid f \mid} \sin\theta \epsilon(1 - \epsilon) \mid T_{\text{loc}} - T_{\text{f, loc}} \mid \tag{15}$$

**Table 2.** Constant parameters used in the melt parameterisations. Note that, due to different reference densities, $\beta_T$ and $\beta_S$ differ between the box and plume parameterisation.

| Symbol | Parameter | Value |
|---|---|---|
| **General** | | |
| $\rho_i$ | Ice density | 917 kg m$^{-3}$ |
| $\rho_{oc}$ | Seawater density | 1028 kg m$^{-3}$ |
| $g$ | Gravitational acceleration | 9.81 m s$^{-2}$ |
| $f$ | Coriolis parameter | -1.4 × 10$^{-4}$ s$^{-1}$ |
| $L_i$ | Latent heat of fusion | 3.34 × 10$^5$ J kg$^{-1}$ |
| $c_i$ | Specific heat capacity of ice | 2.0 × 10$^3$ J kg$^{-1}$ °C$^{-1}$ |
| $c_{oc}$ | Specific heat capacity of ocean | 3974 J kg$^{-1}$ °C$^{-1}$ |
| $\lambda_1$ | Liquidus slope | -5.75 × 10$^{-2}$ °C PSU$^{-1}$ |
| $\lambda_2$ | Liquidus intercept | 8.32 × 10$^{-2}$ °C |
| $\lambda_3$ | Liquidus pressure coefficient | 7.59 × 10$^{-4}$ °C m$^{-1}$ |
| **Box parameterisation** | | |
| $\beta_{S\star}$ | Salt contraction coefficient | 7.7 × 10$^{-4}$ PSU$^{-1}$ |
| $\beta_{T\star}$ | Thermal expansion coefficient | 7.5 × 10$^{-5}$ °C$^{-1}$ |
| $\rho_*$ | EOS reference density | 1033 kg m$^{-3}$ |
| **Plume parameterisation** | | |
| $\beta_S$ | Salt contraction coefficient | 7.86 × 10$^{-4}$ PSU$^{-1}$ |
| $\beta_T$ | Thermal expansion coefficient | 3.87 × 10$^{-5}$ °C$^{-1}$ |
| $C_\epsilon$ | Slope correction parameter | 0.6 |
| $C_d$ | Drag coefficient | 2.5 × 10$^{-3}$ |

The remaining unknown parameter to tune is then:

$$K = C_d^{1/2}\Gamma_{TS}\epsilon(1-\epsilon) \tag{16}$$

In formulations that do not explicitly include the slope, this means that a uniform slope $\sin\theta_{\text{Ant}}$ applicable to all Antarctica is assumed. Otherwise, either the *local* slope $\theta_{\text{loc}}$, for each point, or the *cavity* slope $\theta_{\text{cav}}$, on the ice-shelf level, can be used. We estimate the local slope between the neighbouring grid cells in x- and y- directions at each ice-shelf point. The cavity slope is estimated as the angle opened by the deepest grounding line point, the average ice draft depth at the ice shelf front, and the maximum distance between the grounding line and the ice shelf front.

So far, we assumed a local geostrophic balance due to a gradient between the ambient ocean properties and the local properties of the top boundary layer influenced by melting. However, the melt-induced circulation takes place at the scale of the cavity with non-local effects of melt rates (Jourdain et al., 2017). This is why Favier et al. (2019) proposed that the circulation should

be driven by the thermal forcing averaged over the whole ice shelf instead. In our reformulation, this results in the following:

$$\gamma_{TS,\text{semiloc},\theta} = C_d^{1/2} \Gamma_{TS} \frac{c_{\text{oc}}}{L_i} \beta_S \langle S_{\text{loc}} \rangle \frac{g}{2|f|} \sin\theta\epsilon(1-\epsilon) \,|\, \langle T_{\text{loc}} - T_{\text{f, loc}} \rangle \,| \tag{17}$$

and, when combined with Eq. (7) the following melt rate:

$$m = C_d^{1/2} \Gamma_{TS} \frac{\rho_{\text{oc}}}{\rho_{\text{i}}} \left( \frac{c_{\text{oc}}}{L_i} \right)^2 \beta_S \langle S_{\text{loc}} \rangle \frac{g}{2|f|} \sin\theta\epsilon(1-\epsilon) \,|\, \langle T_{\text{loc}} - T_{\text{f, loc}} \rangle \,| \, (T_{\text{loc}} - T_{\text{f, loc}}) \tag{18}$$

where the $\langle \cdot \rangle$ notation denotes a spatial average. We call this formulation the *quadratic-semilocal* parameterisation. If it is assumed that a constant $\theta_{\text{Ant}}$ can characterise all Antarctic ice shelves, this is analogous to the one proposed by Favier et al. (2019) and used as a standard parameterisation for ISMIP6 (Nowicki et al., 2020; Jourdain et al., 2020). If the local slope $\theta_{\text{loc}}$ is accounted for, this is analogous to the one used in Lipscomb et al. (2021) and discussed by Little et al. (2009), Jenkins et al. (2018) and Jourdain et al. (2020).

Note that we made the choice here to parameterise $U$ based on the assumption of a geostrophic circulation. We could also have parameterised it based on the assumption of a plume circulation (as described in the next subsection).

### 2.2.2 Plume parameterisation

More complex basal melting parameterisations aim to mimic the vertical overturning circulation in the cavity. The *plume parameterisation* describes the evolution of a 2-dimensional buoyant plume, originating at the grounding line with zero thickness and velocity. It evolves along the ice shelf base where it is affected by entrainment of ambient ocean water and melt at the ice-ocean interface. We implement it in two configurations. The first configuration of the two-dimensional plume parameterisation was initially proposed by Lazeroms et al. (2018). Here, we implement the revised, more physical, version described in Lazeroms et al. (2019). The second configuration of the plume parameterisation is a slightly modified version proposed for this particular study, separating the effects of the temperature and velocity on the thermal forcing.

In both configurations, the melt rate $m$ [in m ice per s] is computed as follows:

$$m = M_1 \cdot M_2 \cdot \frac{\rho_{\text{oc}}}{\rho_{\text{i}}} \tag{19}$$

The two versions mainly differ in the definition of the grounding line depth and of the input temperature $T$, salinity $S$ and slope $\theta$ used to compute $M_1$ and $M_2$. The grounding line can be the effective grounding line depth (subscript *gl*) or the deepest grounding line point (subscript *deepest gl*). The hydrographic input properties and the slope can be taken on the cavity scale (subscript *cav*), on the local scale (subscript *loc*), or on the upstream scale (subscript *ups*). The cavity scale means that the far-field temperature and salinity are extrapolated to the ice draft depth for each point and then averaged over the ice shelf area, and that one single slope is estimated for the whole cavity as described in Sec. 2.2.1. The local scale means that the far-field properties are extrapolated to the local ice draft depth and that we use the local slope as defined in Lazeroms et al. (2018). Note that this definition of local slope differs from the definition in the simple parameterisations (Sec. 2.2.1), so we add "laz" (for "Lazeroms") to the subscript. The effective grounding line depth and the local slope are computed as described in Lazeroms

et al. (2018), evaluating possible plume origins in 16 directions for each ice shelf point and averaging the local slope and grounding line depth, respectively, over the plausible plume origin directions. The upstream scale means that we average, for each point of the ice shelf, the portion of the far-field input profiles located between the local ice draft depth and local effective grounding line depth. For the upstream slope, we take, at each point, the angle opened by the effective grounding line depth, the local ice draft depth, and the shortest distance between the grounding line and the given point. In the following, we highlight the subscripts in bold when they differ between the formulations.

$M_1$ is computed as follows in the *Lazeroms* version:

$$M_{1,\text{Lazeroms}} = \left[ \frac{\beta_S \mathbf{S_{cav}} g}{\lambda_3 (L_i/c_{oc})^3} \right]^{1/2} \left[ \frac{1 - \mathbf{c_{\rho 1,cav}} C_d^{1/2} \Gamma_{TS}}{C_d + E_0 \sin\theta_{\mathbf{loc,laz}}} \right]^{1/2} \left[ \frac{C_d^{1/2} \Gamma_{TS} E_0 \sin\theta_{\mathbf{loc,laz}}}{C_d^{1/2} \Gamma_{TS} + \mathbf{c_{\tau,cav}} + E_0 \sin\theta_{\mathbf{loc,laz}}} \right]^{3/2} \cdot (\mathbf{T_{cav}} - \mathbf{T_{f,gl}})^2 \quad (20)$$

In the *modified* version, the last factor of $M_1$ is divided into the part driven by the velocity scale (on the local scale) and the temperature scale (upstream mean):

$$M_{1,\text{modified}} = \left[ \frac{\beta_S \mathbf{S_{loc}} g}{\lambda_3 (L_i/c_{oc})^3} \right]^{1/2} \left[ \frac{1 - \mathbf{c_{\rho 1,loc}} C_d^{1/2} \Gamma_{TS}}{C_d + E_0 \sin\theta_{\mathbf{loc}}} \right]^{1/2} \left[ \frac{C_d^{1/2} \Gamma_{TS} E_0 \sin\theta_{\mathbf{loc}}}{C_d^{1/2} \Gamma_{TS} + \mathbf{c_{\tau,loc}} + E_0 \sin\theta_{\mathbf{loc}}} \right]^{1/2} \cdot (\mathbf{T_{loc}} - \mathbf{T_{f,loc}})$$
$$\cdot \left[ \frac{C_d^{1/2} \Gamma_{TS} E_0 \sin\theta_{\mathbf{ups}}}{C_d^{1/2} \Gamma_{TS} + \mathbf{c_{\tau,ups}} + E_0 \sin\theta_{\mathbf{ups}}} \right] \cdot (\mathbf{T_{ups}} - \mathbf{T_{f,ups,gl}}) \quad (21)$$

$E_0$ is the entrainment coefficient. The parameters $c_{\rho 1}$ and $c_\tau$, presented in in Lazeroms et al. (2019), can be defined on the cavity, the local or the upstream scale:

$$\mathbf{c_{\rho 1,loc/cav/ups}} = \frac{L_i/c_{oc}}{C_d^{1/2} \Gamma_{TS}} \frac{\beta_T}{\beta_S \mathbf{S_{loc/cav/ups}}} \quad (22)$$

$$\mathbf{c_{\tau,loc/cav/ups}} = \frac{-\lambda_1 \beta_T/\beta_S}{\mathbf{c_{\rho 1,loc/cav/ups}}} \quad (23)$$

The formulation of $M_2$ is the same in both versions:

$$M_2 = \frac{1}{2\sqrt{2}} [3(1-x)^{4/3} - 1] \sqrt{1 - (1-x)^{4/3}} \quad (24)$$

but based on two different characteristic length scales:

$$x_{\text{Lazeroms}} = \lambda_3 \frac{z_{\text{draft}} - \mathbf{z_{gl}}}{\mathbf{T_{loc}} - \mathbf{T_{f,gl}}} \left[ 1 + C_\epsilon \left( \frac{E_0 \sin\theta_{\mathbf{loc, laz}}}{C_d^{1/2} \Gamma_{TS} + \mathbf{c_{\tau,loc}} + E_0 \sin\theta_{\mathbf{loc, laz}}} \right)^{3/4} \right]^{-1} \quad (25)$$

$$x_{\text{modified}} = \lambda_3 \frac{z_{\text{draft}} - \mathbf{z_{deepest\ gl}}}{\mathbf{T_{cav}} - \mathbf{T_{f,\ deepest\ gl}}} \left[ 1 + C_\epsilon \left( \frac{E_0 \sin\theta_{\mathbf{cav}}}{C_d^{1/2} \Gamma_{TS} + \mathbf{c_{\tau,cav}} + E_0 \sin\theta_{\mathbf{cav}}} \right)^{3/4} \right]^{-1} \quad (26)$$

where $C_\epsilon$ is a slope correction parameter (see Table 2).

In the plume parameterisation, two parameters can be tuned: the effective thermal Stanton number $C_d^{1/2} \Gamma_{TS}$ and the entrainment coefficient $E_0$.

### 2.2.3 Box parameterisation

The *box parameterisation* was originally proposed by Olbers and Hellmer (2010) and further developed as *PICO* by Reese et al. (2018a). It simulates the overturning transport of heat and salt from the far field to the grounding line, and then along the ice-ocean interface up to the front. We divide each ice-shelf domain into several boxes, which are defined based on the relative distance between the closest ice-shelf front and the closest grounding line point. The division into boxes for each ice shelf is done following the criteria given in Reese et al. (2018a), depending on $r$, the non-dimensional relative distance to the grounding line at each point:

$$r = d_{\text{GL}}/(d_{\text{GL}} + d_{\text{IF}}) \tag{27}$$

and a grid point belongs to box $k$ if:

$$1 - \sqrt{(n_D - k + 1)/n_D} \leq r \leq 1 - \sqrt{(n_D - k)/n_D} \tag{28}$$

where $d_{\text{GL}}$ and $d_{\text{IF}}$ are the distance between the ice shelf point and the nearest grounding line point on the one hand and between the ice shelf point and the nearest ice shelf front point on the other hand, $n_D$ is the total number of boxes, and $k \in [1, n_D]$.

We use the criterion given by Reese et al. (2018a) to define the number of boxes for each ice shelf, resulting in an number of boxes between 2 and 5 (see Table D1, last column), similar to the configuration used in PICO, and called *PICO boxes* in the following. Favier et al. (2019) showed that melt rates do not necessarily converge above five boxes. Therefore, we investigate three additional box setups: one with two boxes, one with five boxes, and one with ten boxes. If one of the boxes has an area of zero because the resolution of the ice shelf is too low, or if the slope between two boxes is negative in the direction from the grounding line to the ice shelf front, we apply a smaller number of boxes in the given setup until all boxes have a non-zero area and the slope between all boxes is positive. When such correction is needed, we enforce that $n_{\text{D, 10 boxes}} > n_{\text{D, 5 boxes}} > n_{\text{D, 2 boxes}}$, with 1 box being the lowest number possible. The resulting number of boxes in each setup is shown in Table D1.

In contrast to the simple and plume parameterisations, the box parametrisation does not use the vertical profile as input. Instead, only the far-field properties at the average entrance depth of each ice-shelf cavity $T_0$ and $S_0$ are advected to the grounding line. This is slightly different from Reese et al. (2018a) who consider the sea-floor temperature on the scale of larger basins. Then, the far-field water mixes with meltwater and rises up along the ice-shelf base due to buoyancy. In contrast to plume models (e.g. Jenkins, 1991), entrainment of deeper water is neglected. Assuming steady state, the temperature $T_k$ and salinity $S_k$ for the box $k$ ($k$ increasing from the grounding line to the ice shelf front) depend on the temperature $T_{k-1}$ and the salinity $S_{k-1}$ in the previous box, the area $A_k$ of the current box, the melt rate $m_k$ in the current box and the overturning flux $q$:

$$
\begin{aligned}
0 &= q(T_{k-1} - T_k) - A_k m_k \left( \frac{\rho_{\text{oc}} c_{\text{oc}}}{\rho_{\text{i}} L_{\text{i}}} \right) \\
0 &= q(S_{k-1} - S_k) - A_k m_k S_k
\end{aligned}
\tag{29}
$$

with the overturning flux calculated as:

$$q = C\rho_\star[\beta_{S\star}(S_0 - S_1) - \beta_{T\star}(T_0 - T_1)] \tag{30}$$

where $C$ is the overturning strength, $\rho_\star$ is the reference density for the haline contraction coefficient $\beta_{S\star}$ and the thermal expansion coefficient $\beta_{T\star}$ (see Table 2).

Finally, the melt rate [in m ice per s] is computed similarly to Eq. (7) for each box:

$$m_k = \gamma_T^\star \left( \frac{\rho_{oc} c_{oc}}{\rho_i L_i} \right) (T_k - T_{f,k}) \tag{31}$$

where $\gamma_T^\star$ is the effective turbulent temperature exchange velocity, assumed to be constant and uniform, like in the *linear-local* parameterisation, which also contrasts with most plume models (e.g. Jenkins, 1991). $T_{f,k}$ is the freezing point in box $k$. $T_{f,k}$ can either be assumed as constant, computed based on the mean depth of the box (homogeneous boxes in the following), or can be assumed as depth-dependent at each ice-draft point, computed based on the local ice-draft depth (heterogeneous boxes in the following). A more detailed description of the equations underlying the box parameterisation can be found in Reese et al. (2018a). In the box parameterisation, there are two parameters to be tuned: the overturning coefficient $C$ and the effective turbulent temperature exchange velocity $\gamma_T^\star$.

### 2.2.4 PICOP parameterisation

The *PICOP parameterisation* is a combination of the box and plume parameterisation (Pelle et al., 2019). The temperature and salinity in the ice shelf cavity are computed using PICO (Reese et al., 2018a). This temperature and salinity are then used as input for the plume parameterisation as described in Lazeroms et al. (2018). In that plume formulation, $M_2$ is not described by an analytical function like in Eq. (24) but by a polynomial (see Eq. (A13), Eq. (A10), and Eq. (A11) in Lazeroms et al. (2018)). We use all parameters as defined in Lazeroms et al. (2018) and Pelle et al. (2019), except $C_d^{1/2}\Gamma_T$ which we change to $7 \times 10^{-5}$ following personal communication with T. Pelle. Unlike in Pelle et al. (2019), we use the effective grounding line depth as in Lazeroms et al. (2018), while Pelle et al. (2019) computed it through a pathway following the ice advection.

In Lazeroms et al. (2018), $C_d^{1/2}\Gamma_{TS}$ was computed based on other fixed parameters. For our re-tuning, we therefore use the plume implementation from Lazeroms et al. (2019). This way, all four parameters $\gamma_{T\star}$, $C$, $C_d^{1/2}\Gamma_{TS}$, and $E_0$ can in principle be tuned here. To reduce complexity, we choose to use the $\gamma_{T\star}$ and $C$ tuned for the corresponding setup of the box parameterisation, and only re-tune $C_d^{1/2}\Gamma_{TS}$, and $E_0$.

### 2.3 Input profiles to the parameterisations

We compute a mean potential temperature and a mean practical salinity profile in front of each ice shelf to be used as "far-field" input to the different parameterisations. We use yearly mean profiles as the residence time of water in ice shelf cavities might be longer than a month for some cavities. Note that this also means that we assume that the advection between the shelf front and the grounding line takes less than a year and we ignore possible longer advection time (see Sec. 4.1.1 for further discussion). For each ice shelf, we sample all grid points within a given distance of the ice shelf front and compute the mean over all

370 such grid points. To study the sensitivity of the tuning to the size of the chosen domain, we define four different domain sizes encompassing all grid points on the continental shelf, defined as points where the depth of the bathymetry is shallower than 1500 m, within 10, 25, 50 and 100 km of the ice shelf front respectively. Additionally, to mimic the coarse resolution of some global climate models, e.g. some of the CMIP or PMIP models (Paleoclimate Modelling Intercomparison Project, Kageyama et al., 2018), that do not properly resolve the continental shelf or associated processes, we define an additional domain, which

we call "offshore" domain. This domain is defined as all ocean points within 10° of longitude and 3.5° of latitude from the ice shelf front and with a bathymetry deeper than 1500m, i.e. we exclude the points on the continental shelf. The offshore domain size is two times the effective resolution of a typical climate model, which we assume to be 5.0° x 1.75° at 70°S, i.e. $\sim 5\Delta x$ for a model of 1° resolution in longitude.

In most cases, there is no clear visible difference between the profiles averaged over 10, 25, 50 and 100 km (REALISTIC

simulation shown as example in Fig. 3) but there is a clear difference between these domains and the offshore domains. As a consequence, for the further analysis, we will keep the diversity introduced by the four simulations of the ensemble (Fig. 2) to introduce variability in our forcing but we will only focus on one domain over the continental shelf and one over the offshore domain. We choose the 50 km domain as most CMIP-type global ocean models have resolutions around 1° (Heuzé, 2021) and this corresponds to a distance of between 38 km (70°S) and 56 km (60°S) in longitude.

## 2.4 Tuning and evaluation approach

### 2.4.1 Evaluation statistics

Our motivation for re-tuning the parameterisations is to minimise the difference between the melt rates simulated by NEMO and the parameterised melt rates for ice shelves all around Antarctica (Fig. 1). A common way to conduct this minimisation is to tune towards and evaluate this difference at the grid-cell level by computing an area-weighted root-mean-squared error:

$$390 \quad RMSE_{local} = \sqrt{\frac{\sum\limits_{j=1}^{N_{grid\ cells}} (m_{param}[j] - m_{ref}[j])^2 a_j}{\sum\limits_{j=1}^{N_{grid\ cells}} a_j}} \tag{32}$$

where $m_{param}[j]$ and $m_{ref}[j]$ are the parameterised and reference melt rates, respectively, in m ice per year in the grid cell $i$, $N_{grid\ cells}$ is the total number of grid cells covered by ice shelves, and $a_i$ is the ice-covered area of the grid cell. However, for this assessment, we argue that minimising Eq. (32) would not yield the best compromise for the tuning and for the conclusion drawn from the evaluation for three reasons. First, this RMSE would be highly biased towards the Filchner-Ronne and Ross

ice shelves, which cover a much larger area (i.e. many more grid cells) than the others, while they are not necessarily the ice shelves that (1) affect most near-future ice dynamics (Seroussi et al., 2020) or (2) contribute the most to the present-day meltwater release into the ocean (Rignot et al., 2013). Second, this RMSE gives the same importance to all grid cells although we know that there are regions which are more or less important for buttressing and therefore for the influence of melt on the ice-sheet evolution (Reese et al., 2018b). Many grid cells of Ross and Filchner-Ronne as well as other smaller ice shelves are

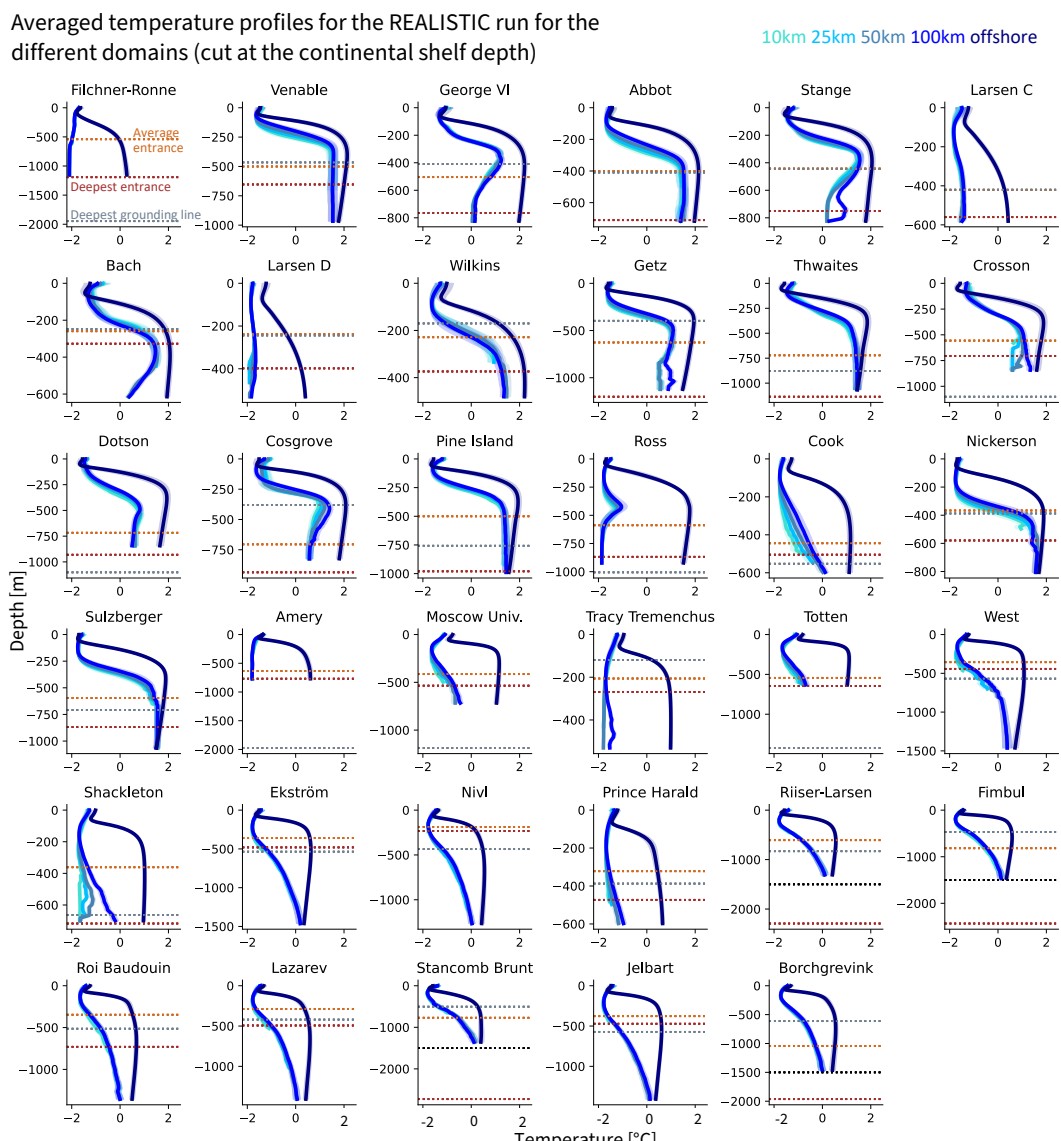

**Figure 3.** Comparison of mean input temperature profiles between the five different domains which were averaged (10, 25, 50, 100 km and offshore) in one given simulation (REALISTIC). The shading represents the interannual variability (one standard deviation over time). The horizontal dotted lines show the average depth of the bed at the ice front (light brown), the largest depth of the bed at the ice front (dark brown) and the largest depth of the grounding line (grey). For a few ice shelves, the deepest entrance is deeper than 1500 m depth but the profiles are defined over the continental shelf (depth shallower than -1500 m). A black dotted line denotes the 1500 m limit in this case. For Stange and Larsen C ice shelves, the average entrance depth and the deepest grounding line overlap.

"passive" and can therefore suffer from biases that will not significantly affect the ice dynamics. Third, we consider that the

melt parameterisations we tune and evaluate are too simple to reproduce all the details of the spatial melt patterns. If they can reproduce the main pattern (e.g. more melt near the grounding line) but the pattern is shifted a little in space, this will result in a high RMSE, penalising a parameterisation that could reproduce some of the complexities of the melt patterns but not at the exact correct location.

Instead, we use the RMSE between the simulated and parameterised yearly integrated melt ($M$) of the individual ice shelves [in Gt/yr] as follows:

$$
RMSE_{\text{int}} = \sqrt{\frac{\sum_{k}^{N_{\text{isf}}} \sum_{t}^{N_{\text{years}}} (M_{\text{param}}[k,t] - M_{\text{ref}}[k,t])^2}{N_{\text{isf}} N_{\text{years}}}}
\tag{33}
$$

where $N_{\text{isf}}$ is the number of ice shelves and $N_{\text{years}}$ the number of simulated years, and the integrated melt $M$ of ice shelf $k$ [in Gt/yr] is:

$$
\quad M[k] = \rho_i \times 10^{-12} \sum_{j}^{N_{\text{grid cells in k}}} m_j a_j
\tag{34}
$$

$RMSE_{\text{int}}$ gives more importance to ice shelves with higher integrated melt and gives the same importance for two ice shelves with the same integrated melt irrespectively of their size, buttressing effect on ice dynamics, or effect on ocean convection. Note that we only consider relatively large ice shelves (those that are well resolved by NEMO) so there is no issue with the number of very small ice shelves that matter for neither ice nor ocean dynamics. In summary, $RMSE_{\text{int}}$ is a careful choice

following our motivation to make conclusions useful for both ocean and ice-sheet modelling on a circum-Antarctic scale. In terms of impact on the ice sheet, this metric will give more importance to ice shelves with a higher integrated melt, which are the most important for the near-future ice-sheet evolution. In terms of impacts on the ocean circulation, we believe that getting a correct freshwater budget around Antarctica (i.e. cavity integrated melt) is a priority before getting the correct depth distribution of the freshwater release at a given location.

To evaluate the performance of the different parameterisations and the robustness of their tuning, we conduct two variations of leave-one-block-out cross-validation on the minimisation of $RMSE_{\text{int}}$, one on the ice shelf dimension and one on the time dimension. This approach consists of dividing the dataset into $N$ blocks, tuning the parameterisation by minimising the evaluation metric on $N-1$ blocks and applying the tuned parameter(s) on the left-out block (Wilks, 2006; Roberts et al., 2017). The procedure is re-iterated $N$ times, leaving out each of the $N$ blocks successively, so that, in the end, each $N$th

block has been left out once. All left-out blocks, using the separately tuned parameters, can then be concatenated to form a "synthetically independent" evaluation dataset. Applying the evaluation metric on this evaluation dataset, we can assess how well the parameterisation generalises to blocks not "seen" during tuning. We apply the cross-validations to each input domain (i.e. the 50 km and the offshore domain) separately, using $RMSE_{\text{int}}$ as an evaluation metric. On the ice-shelf dimension, we use $N$=35 for the cross-validation over ice shelves. On the time dimension, we divide the years into blocks of approximately

10 years (ten 10-year blocks and three 9-year blocks) to reduce the effect of autocorrelation, which is typically 2 to 3 years in our input temperatures. This results in $N$=13 for the cross-validation over time.

Finally, to provide a recommendation for the "best-estimate" parameters to use, we conduct one additional tuning, in which we use all ice shelves and time blocks. To further estimate the uncertainty around the best-estimate parameters (see Sec. 4.1), we turn to block-bootstrapping, as cross-validation per definition provides an overview of the generalisation capabilities of the parameterisations but is not the most robust way to estimate the uncertainty in the parameters themselves (Wilks, 2006). Block-bootstrapping consists of iterating the tuning on different resampled datasets of the *same* size as the original one, here 35 ice shelves x 13 time blocks (Wilks, 2006). To achieve a variety of such samples, we randomly draw an ice shelf and a time block from our data, replace them in the selection pool and repeat the drawing 35 times for the ice shelves and 13 times for the time blocks. This creates a "synthetic" sample of our data with the same sample size as the original sample, which is essential to evaluate uncertainty via bootstrapping. A very large number of synthetic samples, ideally of the order of $10^4$ or higher, can be created this way. The tuning is applied to each of them, resulting in a large distribution of the parameters.

### 2.4.2 Tuning algorithms

The simple parameterisations are based on a linear relationship between the far-field properties and the basal melt. Therefore, for each of them, we compute a thermal forcing factor containing all information that is multiplied with the tuneable parameter, and fit it to the simulated yearly integrated melt via a least-squares regression. The resulting slope is the tuned parameter $\gamma_{TS,\text{loc, Ant}}$ for the *linear-local* parameterisations and $K$ for the other simple parameterisations (see Eq. (16)).

The plume and the box parameterisation are more complex and each have two parameters to be tuned: $(C_d^{1/2}\Gamma_{TS}, E_0)$ and $(\gamma_T^\star, C)$ respectively. The PICOP parameterisation has all these four parameters to be tuned. For simplification, we take the newly tuned "best-estimate" box parameters $(\gamma_T^\star, C)$ and only tune the plume parameters $(C_d^{1/2}\Gamma_{TS}, E_0)$. For the plume and PICOP parameterisation, we use a Trust Region Reflective algorithm (Branch et al., 1999), which loops over different parameter choices within given bounds (here we aim for positive parameters) and minimises $RMSE_\text{int}$ step by step. For the parameters of the box parameterisation, two additional constraints are needed, namely that the melt rate in box 1 (near the grounding line) should always be positive and that the melt rate in box 1 should always be more than the melt rate in box 2 (Reese et al., 2018a). This is why we apply a Sequential Least Squares Programming algorithm (Kraft, 1988), which allows the definition of such constraints, to minimise $RMSE_\text{int}$. Both algorithms are implemented in the python scipy package (Virtanen et al., 2020). To ensure that the algorithms search for physically sensible parameters, we provide physically-informed bounds and use the parameters given by the previous literature as initial guesses.

### 3 Results

We assess the re-tuned parameterisations in two steps. First, we evaluate the performance of the parameterisations in representing the integrated ice-shelf melt compared to our reference, the NEMO simulation, using (1) the parameters suggested in previous work and (2) the cross-validations over time and ice shelves. We also provide a "best estimate" of the re-tuned parameters. Second, we examine the performance of the tuned parameterisations in regard to the spatial distribution of the melt

rates. Note that we only discuss circum-Antarctic results. For a brief overview on the average generalising performance to the individual ice shelves as inferred from the cross-validation over ice shelves, refer to Appendix E.

## 3.1 Evaluation of the parameterised integrated ice-shelf melt and "best-estimate" parameters

### 3.1.1 Overview

The cross-validations over time and ice shelves provide an estimate of how well the parameterisations perform on a time block or an ice shelf that has not been "seen" during tuning. To provide an intuition about what this means: (1) an "unseen" time block represents variations in hydrographic properties driven by climatic variations or ocean model parameters and (2) an "unseen" ice
shelf represents both variations in ice-shelf geometry and variations in hydrographic properties driven by different geographical configurations. We apply Eq. (33) on the synthetically independent evaluation dataset, which is the concatenation of the left-out blocks used for evaluation in the cross-validation. This results in a $RMSE_{int}$ of the parameterisation diagnosed on samples "unseen" during tuning. We show the results of the cross-validation $RMSE_{int}$ when taking the input profiles averaged over the continental shelf within 50 km of the ice-shelf front (Fig. 4, left), and when taking the input profiles averaged over the offshore
domain (Fig. 4, right). For context to these values, the mean reference integrated melt on the ice-shelf level is 39 Gt/yr. The reference integrated melt for the individual ice shelves is shown in the left panel of Fig. E1.

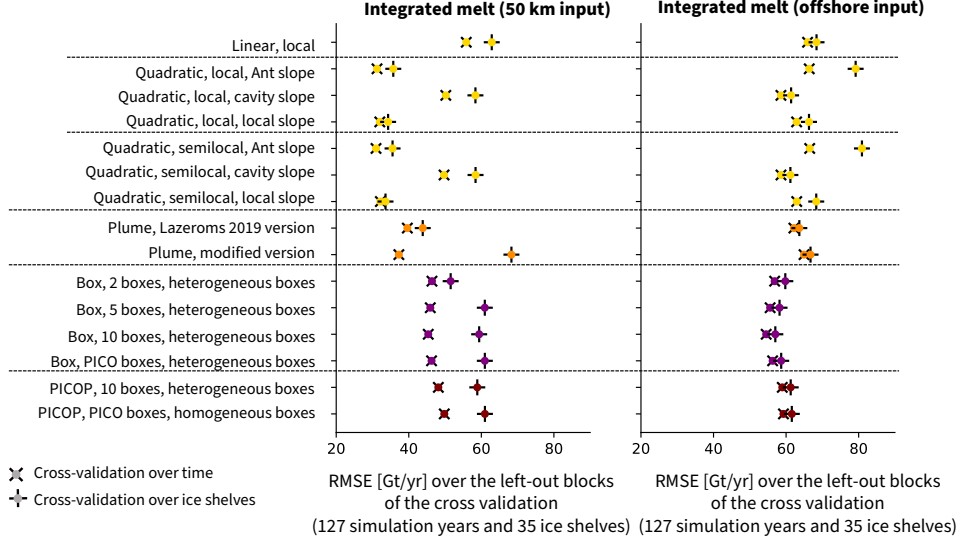

**Figure 4.** Summary of the RMSE of the integrated melt ($RMSE_{int}$) [in Gt/yr] for the cross-validation over time ($\times$) and for the cross-validation over ice shelves (+) for a selection of parameterisations, using the 50 km domain (left) and the offshore domain (right). The colors represent the different parameterisation approaches: simple (yellow), plume (orange), box (purple), PICOP (brown). The RMSE is computed following Eq. (33) on the synthetically independent evaluation dataset (35 left-out ice shelves and 13 left-out time blocks).

For all parameterisations, the cross-validation over time yields a lower RMSE than the cross-validation over ice shelves, which implies that the parameterisation can generalise better to a time block "unseen" during tuning than to an ice shelf "unseen" during tuning. This implies that, in the current formulations of the parameterisations, it is important to take into account as many ice shelves as possible when tuning the parameters to be applicable on the circum-Antarctic scale.

In regard to the generalisation over time, note that our time blocks are taken over an historic sample, where the conditions only vary in a limited way over time although a slightly larger range of variations is introduced through different sets of ocean model parameters (see Sec. 2.1.2). We can therefore not necessarily draw the conclusion that the tuned parameterisations would generalise well under future climate change. Also, climate change will affect the geometry of the ice shelves and the generalisation over ice shelves is challenging for most of the parameterisations.

For the 50 km domain, the lowest RMSE in the integrated ice-shelf melt, on the order of 30 to 35 Gt/yr, is found for the simple quadratic parameterisations using a constant Antarctic slope or the local slope. The RMSE of the "Lazeroms" version of the plume parameterisation is also comparatively low, on the order of 37 to 44 Gt/yr, while the "modified" version struggles with the generalisation over ice shelves.

Using offshore properties substantially increases the RMSE, now reaching 54 to 81 Gt/yr. In this combination, the lowest RMSE is found for the parameterisations performing less well in the 50 km domain, such as the box parameterisation, the simple quadratic formulation using the cavity slope, and the PICOP parameterisation. The increase in RMSE for the offshore domain confirms the importance of using hydrographic properties from the continental shelf to reduce uncertainties, as recommended by Dinniman et al. (2016) and Asay-Davis et al. (2017).

In the following, we further evaluate the performance of each parameterisation type and provide "best-estimate" parameters, tuned over the full original sample.

### 3.1.2 Simple parameterisations

The $RMSE_{\text{int}}$ between the parameterised and reference integrated melt for the simple parameterisations with parameters from previous literature and resulting from the two cross-validations is shown in Table 3. The parameters tuned in our cross-validations, using hydrographic input from the 50 km domain, clearly improve the representation of the integrated sub-shelf melt compared to the use of parameters suggested by Favier et al. (2019) and the "PIGL" recommendation for ISMIP6 by Jourdain et al. (2020), as shown in Table 3 (1st and 3rd column). While the original parameters result in a RMSE between 177 and 1005 Gt/yr, the cross-validations lead to a RMSE between 31 and 63 Gt/yr. Note that the "PIGL" recommendation goes hand-in-hand with local temperature corrections, which are negative for the majority of basins (Jourdain et al., 2020), so the high RMSE here is not necessarily surprising in the absence of temperature corrections. In contrast, the parameters proposed by Jourdain et al. (2020) for the "MeanAnt" case in ISMIP6 considerably reduce the difference between the parameterised and the reference melt (Table 3, 2nd column), especially for the quadratic-semilocal formulation including the local slope. The new tuning achieves only a slight further reduction in the RMSE.

The comparably lowest RMSE, on the order of 30 to 35 Gt/yr, are obtained for the quadratic versions of the parameterisation when using a constant Antarctic slope or including the local slope (Table 3, 4th and 5th column). The difference between the

**Table 3.** Root-mean-squared error ($RMSE_{\text{int}}$) between the reference (NEMO) and parameterised (simple parameterisations) yearly integrated ice-shelf melt [Gt/yr] over the 35 individual ice shelves and 127 simulation years using the parameters from Favier et al. (2019) and ISMIP6 (Jourdain et al., 2020). $RMSE_{\text{int}}$ over the synthetically independent evaluation dataset resulting from the cross-validation (CV) over time and the cross-validation over ice shelves. The 1st to 5th column are computed with input properties from the 50 km domain. The 6th and 7th column show the cross-validation results using the input from the offshore domain. The $RMSE_{\text{int}}$ combining offshore properties and parameters from previous studies are not shown because they are one to three orders of magnitude higher.

| Parameterisation | Favier 2019 | ISMIP6 MeanAnt | ISMIP6 PIGL | CV time | CV ice shelves | CV time | CV ice shelves |
| --- | --- | --- | --- | --- | --- | --- | --- |
| | (50 km) | (50 km) | (50 km) | (50 km) | (50 km) | (offshore) | (offshore) |
| Linear-local | 364.3 | | | 55.8 | 62.9 | 65.9 | 68.4 |
| Quadratic-local Ant slope | 177.0 | 34.5 | 317.4 | 31.2 | 35.7 | 66.4 | 79.2 |
| Quadratic-local cavity slope | | | | 50.2 | 58.4 | 58.6 | 61.4 |
| Quadratic-local local slope | | | | 32.0 | 34.3 | 62.9 | 66.3 |
| Quadratic-semilocal Ant slope | 215.4 | 40.5 | 1004.3 | 31.0 | 35.5 | 66.5 | 80.9 |
| Quadratic-semilocal cavity slope | | | | 49.7 | 58.4 | 58.6 | 61.2 |
| Quadratic-semilocal local slope | | 34.2 | 85.2 | 32.2 | 33.6 | 62.9 | 68.3 |

two values of RMSE resulting from the different cross-validations varies between 1.4 Gt/yr and 4.5 Gt/yr, showing that the parameterisations generalise well on "unseen" samples during tuning for both time and ice shelves. Using the cavity slope as slope information or the linear-local parameterisation leads to comparably higher RMSE, from 49 to more than 60 Gt/yr.

Using offshore properties as input to the parameterisations leads to a RMSE between the parameterised melt and the reference melt from 3 to 40 Gt/yr higher than when using input from the 50 km domain (Table 3, 6th and 7th column). In this case, the RMSE of both cross-validations is lowest when applying quadratic formulations using the cavity slope.

As a recommendation for future users of the simple parameterisations on the circum-Antarctic scale, we provide "best-estimate" parameters obtained by tuning the simple parameterisations over the original sample (all 35 ice shelves and 127 years at once) in Table 4, for the 50 km and the offshore domain respectively. Uncertainty ranges around these parameters are discussed in Sec. 4.1.3.

### 3.1.3 Plume parameterisation

Using 50 km domain input properties, the cross-validation over time yields the same $RMSE_{\text{int}}$ (37.3 Gt/yr, Table 5, 2nd column) for both formulations of the plume parameterisation, lower than using the original parameters (Table 5, 1st column; original parameters shown in Table 6, 1st and 2nd column). The RMSE of the cross-validation over ice shelves is 1.5 times higher for the modified version than for the Lazeroms version, showing that the modified version struggles to generalise over a large variety of ice shelves (Table 5, 3rd column). Using the offshore input properties, the RMSE is higher and varies between 62 and 67 Gt/yr depending on the formulation and the cross-validation (Table 5, 4th and 5th column).

**Table 4.** Summary of the "best-estimate" parameters ($\gamma_{TS,\text{loc, Ant}}$ and $K$, see Eq. (9) and Eq. (16)) tuned over the full 35 ice shelves and 127 years for the simple parameterisations. For the Antarctic slope parameterisations, $K$ is inferred by assuming that $\sin\theta_{\text{Ant}} = 2.9\times10^{-3}$, which is the average over all local slopes in our virtual reality.

| Parameterisation | $\gamma_{TS,\text{loc, Ant}}$ or $K$ tuned (50 km) | $\gamma_{TS,\text{loc, Ant}}$ or $K$ tuned (offshore) |
|---|---|---|
| Linear-local | $2.6\times10^{-6}$ | $0.29\times10^{-6}$ |
| Quadratic-local Ant slope | $11.6\times10^{-5}$ | $0.25\times10^{-5}$ |
| Quadratic-local cavity slope | $5.7\times10^{-5}$ | $0.59\times10^{-5}$ |
| Quadratic-local local slope | $7.9\times10^{-5}$ | $0.34\times10^{-5}$ |
| Quadratic-semilocal Ant slope | $13.4\times10^{-5}$ | $0.26\times10^{-5}$ |
| Quadratic-semilocal cavity slope | $6.3\times10^{-5}$ | $0.61\times10^{-5}$ |
| Quadratic-semilocal local slope | $9.4\times10^{-5}$ | $0.36\times10^{-5}$ |

**Table 5.** Root-mean-squared error ($RMSE_{\text{int}}$) between the reference (NEMO) and parameterised (plume parameterisation) yearly integrated ice-shelf melt [Gt/yr] over the 35 individual ice shelves and 127 simulation years using the original parameters (Lazeroms et al., 2019). $RMSE_{\text{int}}$ over the synthetically independent evaluation dataset resulting from the cross-validation (CV) over time and the cross-validation over ice shelves. It is given for the Lazeroms formulation (Lazeroms et al., 2019) and the modified version.

| Version | Original (50 km) | CV time (50 km) | CV ice shelves (50 km) | CV time (offshore) | CV ice shelves (offshore) |
|---|---|---|---|---|---|
| Lazeroms version | 44.3 | 37.3 | 43.9 | 62.2 | 63.7 |
| Modified version | 104.3 | 37.3 | 68.3 | 64.9 | 66.7 |

As a recommendation for future users of the plume parameterisation on the circum-Antarctic scale, we provide "best-estimate" parameters obtained by tuning the plume parameterisation over the original sample (all 35 ice shelves and 127 years at once) in Table 6, for the 50 km and the offshore domain respectively. Uncertainty ranges around these parameters are discussed in Sec. 4.1.4. The tuned parameters using 50 km domain input properties are of similar order of magnitude as the parameters used in Lazeroms et al. (2019) but the Stanton Number is lower while the entrainment coefficient is higher. Using the offshore input properties for the Lazeroms version, the tuned Stanton Number is 20 times higher than for the 50 km domain, while the entrainment coefficient is one order of magnitude lower. For the modified version, however, the Stanton number is 1000, the upper boundary of our tuning algorithm, instead of $10^{-4}$. After several experiments, we still cannot pinpoint to the exact reason for this large difference in order of magnitude but conjecture that it is related to the behaviour of the modified version on large ice shelves in conjunction with larger thermal forcing from the offshore domain compared to the 50 km domain. We do not recommend using the modified version with offshore properties as the parameters are not physically plausible.

**Table 6.** Comparison between original (from Lazeroms et al., 2019) and "best-estimate" parameters tuned over the full 35 ice shelves and 127 years for the plume parameterisation. "Lazeroms" refers to the version from Lazeroms et al. (2019) and "modified" to the modified version, both presented in Sec. 2.2.2. The values in italic are subject to caution (see text).

| Parameterisation | $C_d^{1/2}\Gamma_{TS}$ Lazeroms | $E_0$ Lazeroms | $C_d^{1/2}\Gamma_{TS}$ tuned (50 km) | $E_0$ tuned (50 km) | $C_d^{1/2}\Gamma_{TS}$ tuned (offshore) | $E_0$ tuned (offshore) |
|---|---|---|---|---|---|---|
| Lazeroms formulation | $5.9\times10^{-4}$ | $3.6\times10^{-2}$ | $2.8\times10^{-4}$ | $4.2\times10^{-2}$ | $42.2\times10^{-4}$ | $0.34\times10^{-2}$ |
| Modified version | $5.9\times10^{-4}$ | $3.6\times10^{-2}$ | $1.3\times10^{-4}$ | $7.6\times10^{-2}$ | *$10.0\times10^{2}$* | *$0.14\times10^{-2}$* |

**Table 7.** Root-mean-squared error ($RMSE_{\text{int}}$) between the reference (NEMO) and parameterised (box parameterisation) yearly integrated ice-shelf melt [Gt/yr] over the 35 individual ice shelves and 127 simulation years using the original parameters (Reese et al., 2018a). $RMSE_{\text{int}}$ over the synthetically independent evaluation dataset resulting from the cross-validation (CV) over time and the cross-validation over ice shelves. RMSE is given for the version with heterogeneous boxes, and for the use of input from the 50 km and the offshore domains.

| Maximum number of boxes | Original (50 km) | CV time (50 km) | CV ice shelves (50 km) | CV time (offshore) | CV ice shelves (offshore) |
|---|---|---|---|---|---|
| 2 boxes | 81.4 | 46.4 | 51.5 | 56.9 | 59.8 |
| 5 boxes | 68.2 | 45.9 | 61.0 | 55.6 | 58.2 |
| 10 boxes | 59.8 | 45.4 | 59.4 | 54.5 | 57.0 |
| PICO boxes | 74.7 | 46.3 | 61.0 | 56.3 | 58.7 |

### 3.1.4 Box parameterisation

With values varying only slightly between 45.4 and 46.4 Gt/yr, the $RMSE_{\text{int}}$ of the cross-validation over time using the 50 km domain input (Table 7, 2nd column) is considerably reduced compared to the RMSE using the original parameters from Reese et al. (2018a), as shown in Table 7 (1st column; original parameters shown in Table 8, 1st and 2nd column). Note that the results do not differ significantly between the homogeneous and heterogeneous boxes approach. We therefore only show results for heterogeneous boxes when we discuss the box parameterisation hereafter. Increasing the number of boxes slightly improves the parameterised melt but the variations between the different setups remains small. This might be explained by the fact that we tune the parameters for each setup separately, resulting in the optimal parameters for each setup, adapting to the difference in the number of boxes. Being 10 to 15 Gt/yr higher, the RMSE of the cross-validation over ice shelves shows that the box parameterisation struggles to generalise to ice shelves "unseen" during tuning (Table 7, 3rd column).

The RMSE of the cross-validation over time is about 10 Gt/yr higher when using offshore properties than when using 50 km domain input and the cross-validation over ice shelves yields a RMSE about 3 Gt/yr higher than the cross-validation over time (Table 7, 4th and 5th column). This suggests that the box parameterisation yields a higher error when using offshore input but the tuning depends less on the sample chosen for tuning than when using 50 km input.

As a recommendation for future users of the box parameterisation on the circum-Antarctic scale, we provide "best-estimate" parameters obtained by tuning the box parameterisation over the original sample (all 35 ice shelves and 127 years at once)

**Table 8.** Comparison between original (from Reese et al., 2018a) and "best-estimate" parameters tuned over the full 35 ice shelves and 127 years for the box parameterisation for the different setups, using heterogeneous boxes.

| Maximum number of boxes | $\gamma_T^\star$ original | $C$ original | $\gamma_T^\star$ tuned (50 km) | $C$ tuned (50 km) | $\gamma_T^\star$ tuned (offshore) | $C$ tuned (offshore) |
|---|---|---|---|---|---|---|
| 2 boxes | $2\times10^{-5}$ | $1\times10^6$ | $0.39\times10^{-5}$ | $16.1\times10^6$ | $0.51\times10^{-5}$ | $0.14\times10^6$ |
| 5 boxes | $2\times10^{-5}$ | $1\times10^6$ | $0.41\times10^{-5}$ | $17.8\times10^6$ | $0.73\times10^{-5}$ | $0.14\times10^6$ |
| 10 boxes | $2\times10^{-5}$ | $1\times10^6$ | $0.44\times10^{-5}$ | $20.5\times10^6$ | $0.92\times10^{-5}$ | $0.14\times10^6$ |
| PICO boxes | $2\times10^{-5}$ | $1\times10^6$ | $0.39\times10^{-5}$ | $20.5\times10^6$ | $0.63\times10^{-5}$ | $0.13\times10^6$ |

in Table 8, for the 50 km and the offshore domain respectively. Uncertainty ranges around these parameters are discussed in Sec. 4.1.4. For the 50 km domain, the newly tuned $\gamma_T^\star$ is five times lower than in Reese et al. (2018a), as shown in Table 8 (3rd column), and of a similar order of magnitude as our tuned $\gamma_{TS,\text{loc},Ant}$ for the linear-local parameterisation (see Table 4). The newly tuned overturning coefficient $C$ is 15 to 20 times higher than the original value (Table 8, 4td column). In Reese et al. (2018a), $C$ was bound at the higher end through the constraint that the mean melt rate in box 2 has to be lower than the mean melt rate in box 1. In our case, this constraint did not lead to an upper bound for $C$, which might be a consequence of different input temperatures compared to Reese et al. (2018a).

The tuned parameters using the input from the offshore domain are distributed differently. $\gamma_T^\star$ is a little higher than the tuned $\gamma_T^\star$ for the 50 km domain, while $C$ is now one order of magnitude lower than the original $C$, i.e. two orders of magnitude lower than the $C$ tuned for the 50 km domain (Table 8, 5th and 6th column).

### 3.1.5 PICOP parameterisation

For the PICOP parameterisation, we only vary the Stanton number and entrainment coefficient and use the tuned "best-estimate" box parameters (see Table 8). We showed in Sec. 3.1.4 that the RMSE remains similar between the different number of boxes and when using the homogeneous and heterogeneous boxes. Therefore, we reduce the diversity of setups for the PICOP tuning. We keep the version using the PICO number of boxes and the homogeneous boxes, as this is the original implementation in Pelle et al. (2019), and we keep a version using the 10-box setup and heterogeneous boxes, as this setup results in the lowest RMSE for the box parameterisation (see Table 7).

For the cross-validations using 50 km domain input, as shown in Table 9, the RMSE is considerably reduced compared to using the original plume parameters from Lazeroms et al. (2019). The RMSE of the cross-validation over ice shelves using the 50 km domain input, on the order of 60 Gt/yr (Table 9, 3rd column) is 10 Gt/yr higher than the RMSE of the cross-validation over time, on the order of 50 Gt/yr for both PICOP variations (Table 9, 2nd column). Using offshore properties, the RMSE of the cross-validation over time is about 10 Gt/yr higher than using 50 km domain input (Table 9, 4th column) and the cross-validation over ice shelves yields a comparable RMSE.

As a recommendation for future users of the PICOP parameterisation on the circum-Antarctic scale, we provide "best-estimate" parameters obtained by tuning the PICOP parameterisation over the original sample (all 35 ice shelves and 127

**Table 9.** Root-mean-squared error ($RMSE_{\text{int}}$) between the reference (NEMO) and parameterised (PICOP parameterisation) yearly integrated ice-shelf melt [Gt/yr] over the 35 individual ice shelves and 127 simulation years using the original parameters (Reese et al., 2018a; Lazeroms et al., 2019). $RMSE_{\text{int}}$ over the synthetically independent evaluation dataset resulting from the cross-validation (CV) over time and the cross-validation over ice shelves.

| PICOP setup | Original (50 km) | CV time (50 km) | CV ice shelves (50 km) | CV time (offshore) | CV ice shelves (offshore) |
|---|---|---|---|---|---|
| 10 boxes, heterogeneous | 67.9 | 48.1 | 59.0 | 59.0 | 61.3 |
| PICO boxes, homogeneous | 68.2 | 49.8 | 61.0 | 59.3 | 61.6 |

**Table 10.** Comparison between original and "best-estimate" parameters tuned over the full 35 ice shelves and 127 years for the PICOP parameterisation. These are the plume parameters, the box parameters are the tuned parameters, for the 10-box and PICO boxes respectively, shown in Table 8.

| PICOP setup | $C_d^{1/2}\Gamma_{TS}$ Lazeroms | $E_0$ Lazeroms | $C_d^{1/2}\Gamma_{TS}$ tuned (50 km) | $E_0$ tuned (50 km) | $C_d^{1/2}\Gamma_{TS}$ tuned (offshore) | $E_0$ tuned (offshore) |
|---|---|---|---|---|---|---|
| 10 boxes, heterogeneous | $5.9\times10^{-4}$ | $3.6\times10^{-2}$ | $0.94\times10^{-4}$ | $30.7\times10^{-2}$ | $2.0\times10^{-4}$ | $95.9\times10^{-2}$ |
| PICO boxes, homogeneous | $5.9\times10^{-4}$ | $3.6\times10^{-2}$ | $0.85\times10^{-4}$ | $34.5\times10^{-2}$ | $1.5\times10^{-4}$ | $136\times10^{-2}$ |

years at once) in Table 10, for the 50 km and the offshore domain respectively. Uncertainty ranges around these parameters are discussed in Sec. 4.1.4. The best-estimate Stanton numbers are lower than the original ones (Table 10).The retuned entrainment coefficients, however, are about 10 times higher than the original ones for the 50 km domain and more than 30 times higher for the offshore domain. These high entrainment coefficients are not necessarily physically plausible and we conjecture that the combination of the box and plume parameterisation within PICOP might violate one or more assumptions taken in the derivation of the individual box and plume parameterisations. In addition, we use the plume formulation by Lazeroms et al. (2019) for the tuning version of PICOP while Pelle et al. (2019) use the formulation of Lazeroms et al. (2018). This might also influence the connection between the temperature and salinity from the box parameterisation and the melt computed through the plume parameterisation.

## 3.2 Evaluation of the spatial melt patterns

While Joughin et al. (2021) suggest that the integrated melt is the main driver for grounding line retreat, other studies suggest that ice-sheet models are more sensitive to melt rates near the grounding line than to cavity-integrated melt rates (e.g. Gagliardini et al., 2010; Reese et al., 2018b; Morlighem et al., 2021). Therefore, simulating realistic melt patterns, especially near the grounding line, might be at least as important as simulating a realistic integrated melt. To assess the parameterisations from another perspective, we investigate their ability to represent time-averaged melt patterns. First, we visually assess the difference between the parameterised and the reference melt pattern. Then, we use the cross-validation results to quantify differences in these time-averaged melt rates near the grounding line between parameterisations and reference through a RMSE. As the

parameterisations clearly perform better when using 50 km domain input, we only concentrate further on the 50 km domain in the following.

### 3.2.1 Visual evaluations

The circum-Antarctic time-averaged pattern for a subset of parameterisations applied to the REALISTIC simulation with the "best-estimate" parameters is shown in Fig. 5. The subset represents the respective configurations of the parameterisations that yield a comparatively lower RMSE in the integrated melt (see Sec. 3.1).

At first sight, on the circum-Antarctic scale, all parameterisations lead to reasonable results compared to the reference. Differences can mainly be seen in terms of refreezing. While the simple parameterisations do not exhibit any refreezing, the plume parameterisation leads to some refreezing under the Filchner-Ronne ice shelf, and the boxes and PICOP lead to large refreezing areas under both the Filchner-Ronne and Ross ice shelves. Also, in the box and PICOP parameterisations, the melt rates in the Amundsen Sea are homogeneous for the different ice shelves, while the other parameterisations better represent strong melt rates for the Pine Island, Thwaites, and Getz ice shelves, and lower melt rates for the others. We suggest that this is because the thermocline depth is particularly important for these ice shelves while only one temperature and salinity value, the one at the average entrance depth, is used as input for the box and PICOP parameterisations and the properties at the entrance depth are similar for all ice shelves in this region (see e.g. Fig. 2).

For a more detailed overview on the ice-shelf level, the time-averaged patterns of a subset of ice shelves, representative for different regions, are shown in Fig. 6. These include the three largest ice shelves Filchner-Ronne, Ross, and Amery; the ice shelf in front of Pine Island Glacier in the Amundsen Sea; the Fimbul ice shelf located at the outskirts of Dronning Maud Land; and Totten ice shelf, located in the East. It becomes clear that the melt patterns are very different on the ice-shelf level depending on the parameterisation. For example, the quadratic local parameterisation using a constant Antarctic slope tends to have a smoother pattern than the quadratic semilocal parameterisation using a local slope. The latter leads to locally very high melt rates, substantially higher than the reference (e.g. for Pine Island or Totten), which could explain the high RMSE for the yearly mean melt rate near the grounding line (see Fig. 7). For this given subset of ice shelves, the Lazeroms version of the plume parameterisation captures some patterns better than the modified version (e.g. Filchner-Ronne, Pine Island, Amery). Both plume parameterisations overestimate the melt rates for Totten Ice Shelf, with the Lazeroms version exhibiting maximum melt rates twice as high as the modified version. The box parameterisation exhibits its signature pattern, i.e. decreasing melt rates from the grounding line to the front for all ice shelves, however the melt rates remains generally lower than the reference, except for the Fimbul Ice Shelf. Finally, PICOP exhibits a pattern very close to the box parameterisation, but with slightly higher melt rates overall and faintly recognisable spatial heterogeneities.

For the Ross ice shelf, all parameterisations exhibit melting over too large areas compared to the reference. Finally, for the Fimbul Ice Shelf, the reference shows high melt rates near the ice front. This is a result of a seasonal melt cycle, where surface water heated by the atmosphere in summer is transported to the ice-shelf front by tides, eddies, and Ekman transport, leading to seasonally high melt rates near the front (third mode of melting described by Jacobs et al., 1992; Silvano et al., 2016). None

of the parameterisations matches the reference pattern, likely because they are all forced by yearly ocean temperatures and because, in the box and PICOP parameterisations, shallow input properties are not taken into account.

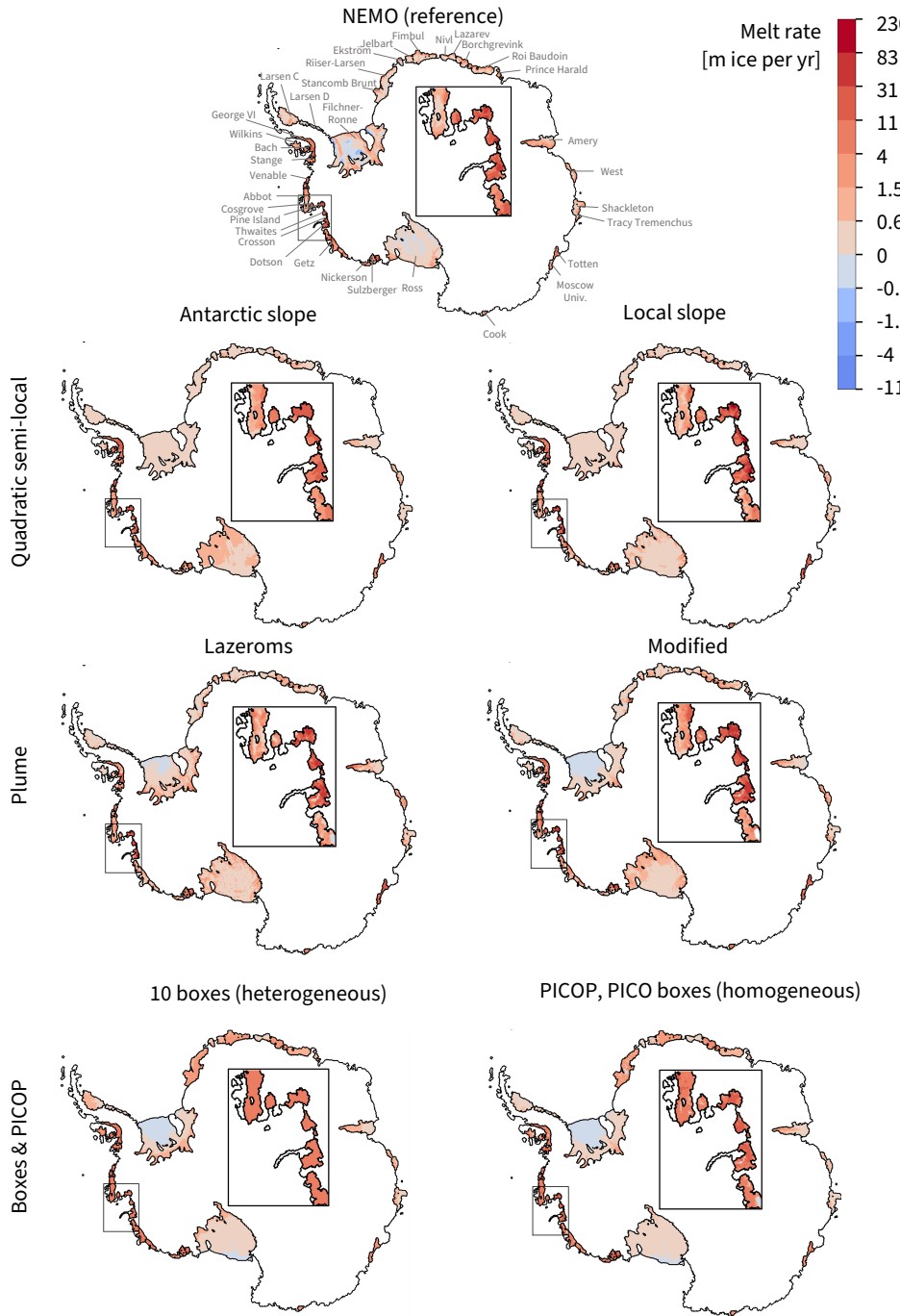

**Figure 5.** Spatial distribution of the yearly mean sub-shelf melt rates for a subset of the tuned parameterisations and for the reference for comparison. The parameterised melt rates are computed using the "best-estimate" parameters given in Tables 4, 6, 8 and 10. This is the time average for the REALISTIC run (39 years). Note that the land tongue in the Amundsen Sea was introduced to mimic the effect of grounded icebergs present in that region on the sea-ice circulation (see Sec. 2.1.2).

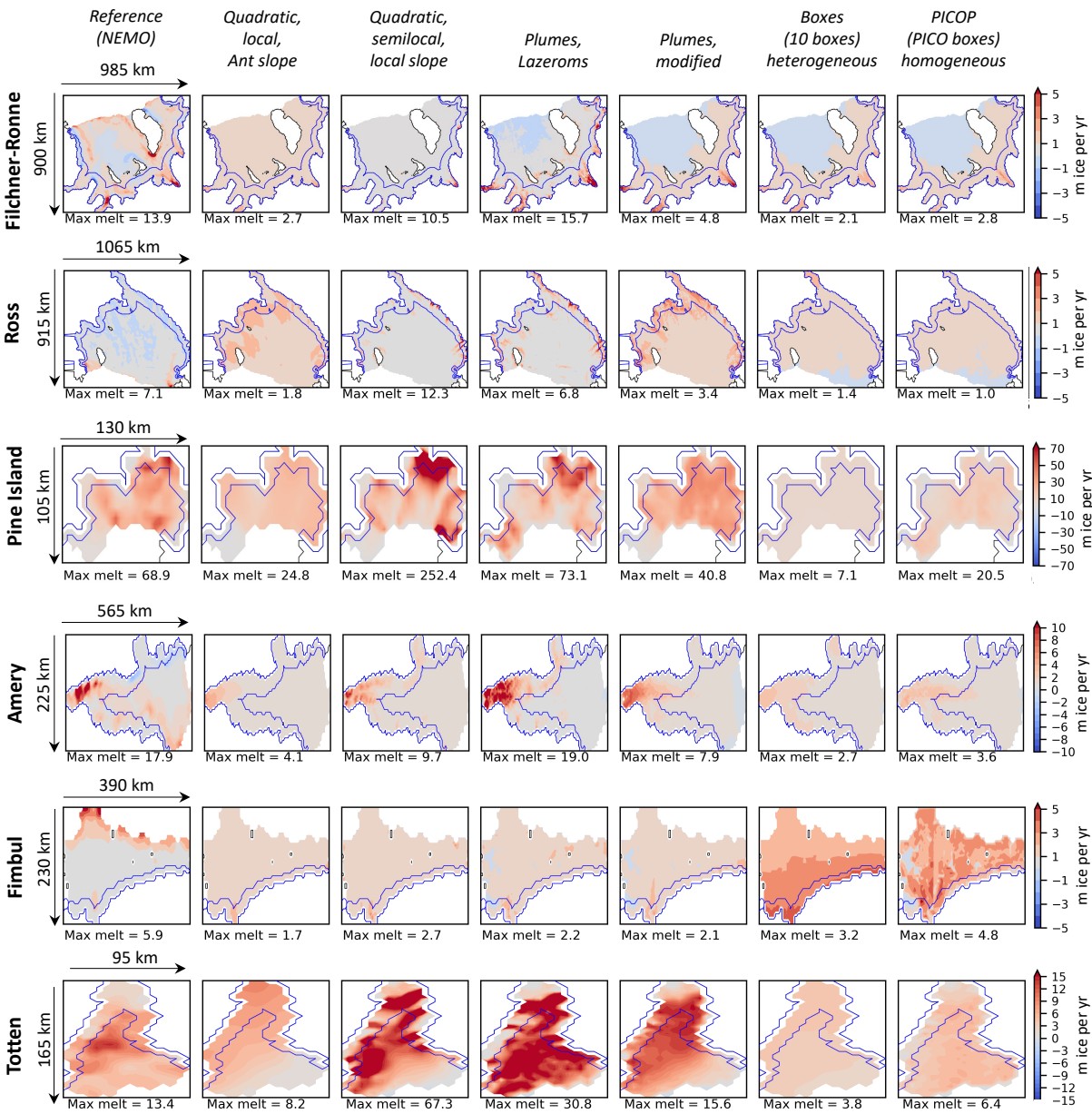

**Figure 6.** Subset of ice shelves for a visual evaluation of the melt patterns. The parameterised melt rates are computed using the "best-estimate" parameters given in Tables 4, 6, 8 and 10. This is the time average for the REALISTIC run (39 years). The blue line indicates the region used to evaluate the melt rate near the grounding line (which is defined as the first box in the 5-box setup of the box parameterisation).

### 3.2.2 Statistical evaluation

To quantify the performance of the different parameterisations, in addition to the visual evaluation, we conduct a statistical
evaluation of the melt rate near the grounding line (GL). To do so, we expand the evaluation of the cross-validations conducted
in Sec. 3.1. Instead of inferring the integrated melt, we compute the mean over time and space of the melt rates in the region
defined through the first box of the box parameterisation in the 5-box setup, which represents $\approx 10\%$ of the ice-shelf area.
We use average melt rates $m$ (in m ice/yr) rather than integrated melt $M$ (in Gt/yr) used in the previous section to have a
complementary metric. With the integrated melt, we focused on an ice-shelf-wide metric, which implicitly contains information
about the size of the ice shelf, and its variability with time. By evaluating the average melt rate over time and space near the
grounding line, we evaluate if, on average, the right melt rate is occurring near the grounding line, independently of the size of
the ice shelf. Again, we compute a RMSE over the synthetically independent evaluation dataset:

$$RMSE_{\text{GL}} = \sqrt{\frac{\sum\limits_{k}^{N_{\text{isf}}} \sum\limits_{n}^{N_{\text{simu}}} (m_{\text{GL,param}}[k,n] - m_{\text{GL, ref}}[k,n])^2}{N_{\text{isf}} N_{\text{simu}}}} \tag{35}$$

where $N_{\text{simu}}$ is the number of simulations in the ensemble and where $m_{\text{GL}}$ for ice shelf $k$ and simulation $n$ is:

$$m_{\text{GL}}[k,n] = \frac{1}{N_{\text{years in n}}} \sum\limits_{t}^{N_{\text{years in n}}} \frac{\sum\limits_{j}^{N_{\text{grid cells near GL in k}}} (m_j a_j)}{\sum\limits_{j}^{N_{\text{grid cells near GL in k}}} a_j} \tag{36}$$

Note that we do not take the average over all 127 years at once but average over the individual time periods covered by the four
different simulations in our ensemble. This is to avoid taking one single average over four different oceanic states that are not
necessarily consistent with each other.

Similarly to the RMSE of the integrated melt, our choice to evaluate our RMSE on the ice-shelf level, i.e. one average
per ice-shelf and not on the grid-cell level, is motivated mainly by two aspects. First, an RMSE evaluating on the grid-cell
level might be biased towards Filchner-Ronne and Ross ice shelves, which have the longest grounding lines. Second, such an
RMSE would also penalise small shifts in the spatial patterns, resulting in possibly higher RMSE for a parameterisation that
could reproduce some of the complexities of the melt patterns near the grounding line but not at the exact correct location. For
completeness, we show the results for an RMSE of the melt rate near the grounding line evaluated on the grid-cell level and
discuss it in Appendix F.

The results for $RMSE_{\text{GL}}$ are shown, alongside the previously presented $RMSE_{\text{int}}$ for the 50 km domain input, in Fig. 7.
For context for these values, the mean reference melt rate near the grounding line is 0.45 m ice/yr and the values for the
individual ice shelves are shown in the right panel of Fig. E1. The $RMSE_{\text{GL}}$ for the cross-validation over time and ice shelves
nearly overlap for the majority of the parameterisations, which means that the average melt rate over time and space near the
grounding line is less sensitive to changing parameters, at least on the scale of values shown here. It also means that the choice
of parameterisation has a much larger influence on the resulting melt rate near the grounding line than the choice of parameters
for the individual parameterisations.

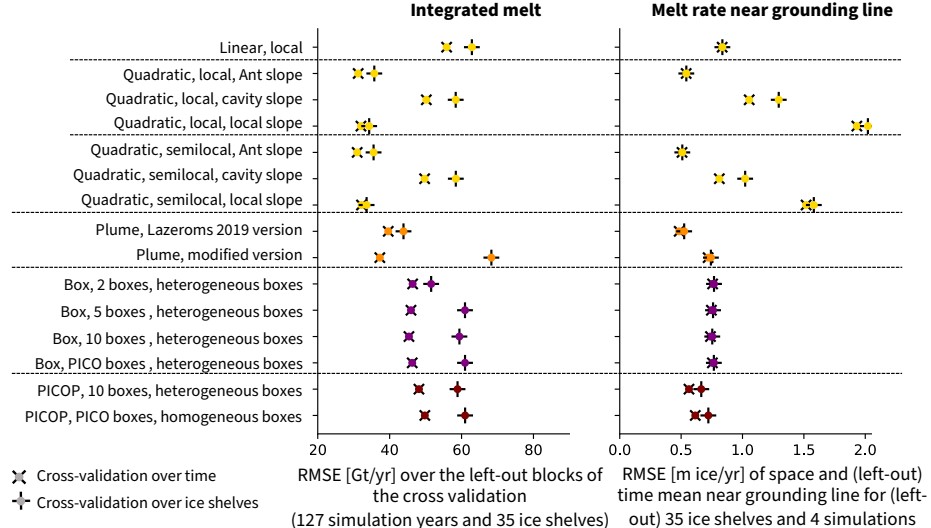

**Figure 7.** Summary of the RMSE of the integrated melt ($RMSE_{\text{int}}$) for the cross-validation over time ($\times$) and for the cross-validation over ice shelves (+) for a selection of parameterisations, using the 50 km domain, [in Gt/yr] (left, same as Fig. 4 left) and summary of the RMSE of the melt rate averaged over time and space near the grounding line (average in the first box of the 5-box setup, $RMSE_{\text{GL}}$) [in m ice/yr] (right). The colors represent the different parameterisation approaches: simple (yellow), plume (orange), box (purple), PICOP (brown). The RMSE is computed following Eq. (33, left panel) and Eq. (35, right panel) on the synthetically independent evaluation dataset.

Similarly to the integrated melt, the simple quadratic parameterisations using a constant Antarctic slope and the plume parameterisation lead to the lowest $RMSE_{\text{GL}}$. However, using a local slope leads to a high $RMSE_{\text{GL}}$ in both cross-validations.

This can be explained by locally very high (and too high) melt rates near the grounding line, as seen in Fig. 6 for Ross, Pine Island, and Totten for example. While the local slope leads to reasonable RMSE in the integrated melt, it is therefore important to keep in mind that it can induce an overestimation of the melt rates on a local scale. In contrast, PICOP performs comparably less well for the integrated melt but very well for the melt rate near the grounding line, with an RMSE comparable to the quadratic parameterisation assuming a constant Antarctic slope and to the Lazeroms version of the plume parameterisation.

This suggests that the combination of box and plume parameterisation is useful for parameterising the melt rate near the grounding line.

## 4 Discussion

### 4.1 Uncertainties in the tuning

The tuning of the various parameters in this study was done in a more consistent way and with larger amounts of data than would be currently possible with observations, making the newly tuned parameters more representative in time and space. However, uncertainties remain, which we discuss and estimate in the following.

#### 4.1.1 The perfect-model approach and the use of NEMO

Our perfect-model approach relies on the assumption that NEMO results in a realistic approximation of the circulation in the ice-shelf cavity and melt behaviour at the ice-ocean interface. While it is clear that NEMO does not replicate reality exactly, a part of the melt biases found in NEMO (see Appendix B) are a result of problems unrelated to the representation of the ice-shelf cavities (e.g. Southern Ocean biases related to the atmospheric forcing, the representation of sea ice, or unresolved or poorly parameterised ocean processes). Such biases do not alter the physical consistency of the relationship between ocean properties in front of ice shelves and basal melting, and are therefore not a problem for our perfect-model approach.

It is nonetheless obvious that the representation of ice-ocean exchange in NEMO is far from perfect. As presented in Sec. 2.1.1, the melt still relies on a parameterisation, which is, however, more advanced than the ones we assess. The resolution of NEMO is of several kilometers, which might hinder an accurate representation of the small-scale geometry and the small-scale processes occurring near the grounding line. As we assess the parameterisations based on NEMO's topography here, which does not include the thinnest part of the cavity near the grounding line (NEMO needs at least two vertical cells), this does not affect our conclusions whether a given parameterisation emulates NEMO well but might be a limitation of our assessment concerning applications on smaller scales. Still, some uncertainty remains because not resolving small-scale geometric features, such as thin bathymetric ridges and basal channels, and eddies may also affect the ocean circulation in the whole cavity, which could affect both local and integrated melt.

To gain a first-order insight into the importance of the resolution in the assessment and application of the different parameterisations, we conduct a quick experiment focused on the Pine Island Glacier ice shelf, for which multiannual observational estimates of input temperature and salinity (Dutrieux et al., 2014), high-resolution topography (500 m resolution, Morlighem, 2020; Morlighem et al., 2020) and high-resolution observational estimates of the basal melt rates (32-256 m resolution, Shean et al., 2019) are available. We use these temperature and salinity profiles and high-resolution topography as input to our parameterisations, and apply our "best-estimate" parameters. We compare the resulting melt patterns (Fig. 8b) to the high-resolution observational melt rate estimates (Fig. 8a). We find that most of the parameterisations clearly underestimate the melt rates, which does not necessarily come as a surprise as it was already visible at lower resolution (Fig. 6) and therefore rather is a result of the circum-Antarctic tuning than of the resolution. Taking into account the local slope, in contrast, leads to a large overestimation over large parts of the ice shelf. We suggest that this is a result of small-scale irregularities in the draft geometry that are either introduced by the high resolution and therefore less smoothed out or introduced by imprecisions in the observa-

tional estimate. The resulting higher local slopes lead to higher melt rates. The overestimation implies that the local slope is
not necessarily a good feature to locally adjust the melt rate.

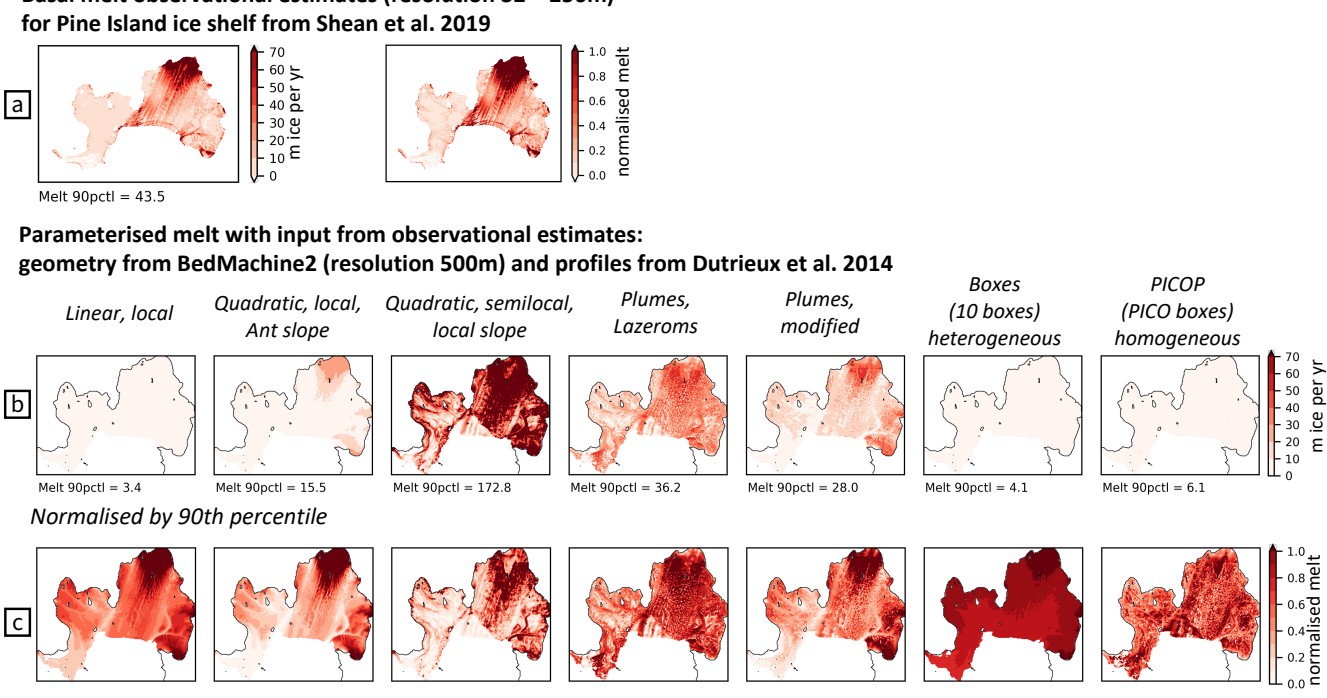

**Figure 8.** Comparison between (a) observational estimates inferred from remote sensing and (b) parameterised melt rates inferred for the Pine Island Glacier ice shelf based on observational estimates of input properties and topography and the "best-estimate" tuned parameters. (c) Parameterised melt rates normalised by dividing by the 90th percentile. The observational estimates of basal melt rates (mean over 2008-2015) were inferred by Shean et al. (2019) using satellite and airborne altimetry with an "initial-pixel method" (see Fig. 7b in Shean et al., 2019). The input temperature and salinity are from Dutrieux et al. (2014) and cover 6 years (1994, 2000, 2007, 2009, 2010, 2012). The Bedmachine Antarctica version 2 dataset (Morlighem, 2020; Morlighem et al., 2020) is used for the geometric information. We averaged the parameterised basal melt patterns over the 6 years.

Normalising the melt rate by dividing by the 90th percentile (Fig. 8c) provides an additional perspective on the pattern, more independent from biases induced by the tuning. We see that taking a constant Antarctic slope and the modified version of the plume parameterisation result in patterns closest to the observational estimates (Fig. 8a), while the highest normalised melt rate is more homogeneously distributed across the ice shelf in the Lazeroms version of the plume parameterisation, in the box
parameterisation and in the PICOP parameterisation. This result, based on one single ice shelf, implies that the modified plume might adapt better to increasing resolution.

Nevertheless, this quick check remains focused on one ice shelf and might not provide robust conclusions applicable on the circum-Antarctic scale as we saw that most parameterisations struggle to adapt to ice shelves "not seen" during tuning. Further

work with alternative models (e.g. sigma-coordinate models and higher resolution) is needed to assess the uncertainty related to the use of our 1/4° NEMO simulations.

In addition, NEMO is advantageous to use for this study as it resolves the ice-shelf cavities. However, it is run in an uncoupled mode, which means that basal melt does not affect the ice-shelf geometry. While this is unrealistic at first glance, the aim of this study was to assess the physical link between the hydrographic properties in front of the ice shelf and the melt rates. A change in geometry of the ice shelf would affect the melt patterns as such but should in principle not affect the physical link between hydrographic properties and the melt.

As we have seen with the Fimbul ice shelf, the use of yearly average input temperature and salinity does not allow the representation of seasonal melt, which takes place at the front of a few ice shelves due to ocean surface warming in summer (Silvano et al., 2016). If we used monthly temperature and salinity profiles as input, it should be possible to represent this seasonal melt ("mode 3" melt), except for the box and PICOP parameterisations. This would be possible both for relatively small cavities, for which the residence time of the water in the cavity is significantly shorter than the seasonal period, and for larger cavities, as seasonal melt usually occurs relatively close to the ice-shelf front. However, while going to shorter time scales would improve the representation of mode 3 melt close to the ice shelf front, it would probably require accounting for the possible time lag between the input forcing entering the cavity and the occurrence of the melt near the grounding line (Holland, 2017), which has been ignored in this study. Even with the yearly data used in our study, not accounting for the lag is a limitation for the largest ice-shelf cavities in which the residence time of water reaches several years (Michel et al., 1979; Nicholls and Østerhus, 2004).

### 4.1.2   The choice of the statistical metrics

The tuning and evaluation of the parameterisations relies heavily on the statistical metrics used. We decided to tune to the integrated shelf melt of the 35 largest ice shelves and evaluated against that metric and the time- and space-averaged melt rates near the grounding line. As already mentioned in Sec. 2.4.1, this was a careful choice following our motivation to make conclusions useful for both ocean and ice-sheet modelling on a circum-Antarctic scale. For ice-sheet modelling, it is important that ice shelves with a higher integrated melt are more important during tuning, because they are currently the most important for the ice-sheet evolution. For ocean modelling, it is important to get a correct freshwater budget around Antarctica, in the form of integrated melt. However, we acknowledge that the tuning and evaluation can be done differently, depending on the goal of the tuning.

If the goal is to match the melt of all ice shelves better, one possibility is to use region-dependent parameters. As our study showed that one constant parameter on the circum-Antarctic scale leads to high RMSE, the parameters could be tuned separately for each region or basin. While this is a practical way to reduce regional biases introduced by the parameters, we argue that the parameters would lose some of their physical meaning in that case and would compensate even more biases that are not directly related to melt physics than when tuned on the circum-Antarctic scale.

If the goal is to evaluate the performance of the parameterisation for each grid cell, regardless of where it is situated, one possibility is to use the RMSE on the grid-cell level described in Eq. (32). This would be the most objective and universal

evaluation method but we argue that the parameterisations are too simple yet to correctly reproduce the melt patterns at the grid-cell level.

Finally, if the goal is to evaluate on the grid-cell level but giving more importance to some cells and less importance to others, one possibility is to use a RMSE weighted with the buttressing flux response numbers presented in Reese et al. (2018b). This would give more importance to points that are more important for the buttressing of the ice sheet. As most of the points with high buttressing flux response numbers are situated near the grounding line, we argue that this is close to evaluating the melt rates near the grounding line.

### 4.1.3   Simple parameterisations

To examine the uncertainty in the tuned parameters for the simple parameterisations, we conduct a range of bootstrapping experiments, as explained in Sec. 2.4.1. We replicate our tuning procedure by applying a least-squares linear fit to 15,000 different synthetic samples chosen via bootstrapping. Each tuning sample has the same number of data blocks (35 ice shelves x 13 time blocks), which are randomly drawn with replacement. The 5th, 10th, 33rd, 50th, 66th, 90th, and 95th percentiles describing

the resulting distribution of 15,000 parameters for the simple parameterisations are shown in Table 11.

    The medians of the distributions are close to the best-estimate parameters shown in Table 4. The distributions of the parameters of the different quadratic formulations slightly overlap. The distribution of the formulation taking into account the local slope overlaps with the lower tail of the distribution from the formulation assuming a constant Antarctic slope and the upper tail of the distribution from the formulation taking into account the cavity slope. The parameters of the semilocal formulation

are slightly higher than the parameters of the local formulation.

**Table 11.** Percentiles describing the uncertainty range of the $\gamma_{TS,\mathrm{loc,\,Ant}}$ for the linear local parameterisation and $K$ for the other simple parameterisations after 15,000 bootstrap experiments with replacement. For the Antarctic slope parameterisations, $K$ is inferred assuming that the mean local slope is $\sin\theta_{\mathrm{Ant}} = 2.9 \times 10^{-3}$.

| Parameterisation | 5th | 10th | 33rd | Median | 66th | 90th | 95th |
|---|---|---|---|---|---|---|---|
| Linear-local ($\times 10^{-6}$) | 1.8 | 2.0 | 2.4 | 2.7 | 3.0 | 7.0 | 9.0 |
| Quadratic-local Ant slope ($\times 10^{-5}$) | 8.7 | 9.4 | 11.1 | 12.0 | 12.8 | 14.2 | 14.5 |
| Quadratic-local cavity slope ($\times 10^{-5}$) | 3.2 | 3.5 | 4.7 | 5.5 | 6.2 | 7.7 | 8.2 |
| Quadratic-local local slope ($\times 10^{-5}$) | 6.0 | 6.4 | 7.4 | 7.9 | 8.2 | 8.9 | 9.2 |
| Quadratic-semilocal Ant slope ($\times 10^{-5}$) | 10.1 | 10.9 | 12.9 | 14.0 | 15.1 | 16.3 | 16.7 |
| Quadratic-semilocal cavity slope ($\times 10^{-5}$) | 3.7 | 4.2 | 5.6 | 6.5 | 7.4 | 9.1 | 9.6 |
| Quadratic-semilocal local slope ($\times 10^{-5}$) | 7.3 | 7.8 | 8.9 | 9.3 | 9.7 | 10.5 | 11.0 |

### 4.1.4   Complex parameterisations

As the tuning algorithms are computationally more expensive for the complex parameterisations, we cannot run 15,000 bootstrapping experiments and cannot investigate the uncertainty for each variation of the complex parameterisations. Instead, we

concentrate on a subset of parameterisations. For the Lazeroms version of the plume parameterisation, we apply the tuning to 500 "synthetic" samples generated via bootstrapping. We also conduct the tuning on $2 \times 250$ "synthetic" samples for the box parameterisation with 10 heterogeneous boxes, and PICO heterogeneous boxes, respectively, as well as for $2 \times 250$ "synthetic" samples for the two PICOP configurations presented earlier. The resulting parameters are shown in Fig. 9 and Table 12.

For the plume parameterisation, the entrainment coefficient $E_0$ is mainly clustered between 2 and $10 \times 10^{-2}$ (Fig. 9, left). The Stanton number $C_d^{1/2}\Gamma_{TS}$ varies between 1 and $10 \times 10^{-4}$. For low Stanton numbers, $E_0$ can reach very high values ($\approx 6\%$ of the values above $40 \times 10^{-2}$, not shown). This means that turbulent exchange and entrainment compensate each other to result in the appropriate heat needed to match the reference melting. An inverse quadratic function can be empirically fitted to describe the relationship between the two parameters. The resulting relationship, ignoring $E_0$ above $50 \times 10^{-2}$ (Fig. 9a, grey line), is:

$$E_0 = \left[\frac{15.9}{(C_d^{1/2}\Gamma_{TS} \times 10^4)^2} + 2.2\right] \times 10^{-2} \tag{37}$$

If only fitted to the main cluster (for $E_0 < 15 \times 10^{-2}$, Fig. 9a, black line), the fit results in the following relationship:

$$E_0 = \left[\frac{8.9}{(C_d^{1/2}\Gamma_{TS} \times 10^4)^2} + 2.9\right] \times 10^{-2} \tag{38}$$

Our best-estimate tuned parameters are situated in the middle of the scatter cloud, and are close to the median of the distribution. The parameters from Lazeroms et al. (2019) are located in the uncertainty range, closer to the upper tail of the distribution.

**Table 12.** Percentiles describing the uncertainty range of the $\gamma_T^\star$ for the box parameterisation and $C_d^{1/2}\Gamma_{TS}$ for the plume and PICOP parameterisation for the other simple parameterisations after bootstrap experiments with replacement as shown in Fig. 9.

| Parameterisation | 5th | 10th | 33rd | Median | 66th | 90th | 95th |
|---|---|---|---|---|---|---|---|
| Plume, Lazeroms $C_d^{1/2}\Gamma_{TS} \times 10^{-4}$, 500 bootstrap samples) | 1.4 | 1.6 | 2.2 | 2.6 | 3.1 | 4.9 | 6.0 |
| Boxes, 10-box setup ($\gamma_T^\star \times 10^{-5}$, 250 bootstrap samples) | 0.23 | 0.25 | 0.39 | 0.48 | 0.76 | 1.64 | 1.99 |
| Boxes, PICO boxes ($\gamma_T^\star \times 10^{-5}$, 250 bootstrap samples) | 0.22 | 0.24 | 0.36 | 0.44 | 0.54 | 1.41 | 2.14 |
| PICOP, 10-box setup ($C_d^{1/2}\Gamma_{TS} \times 10^{-4}$, 250 bootstrap samples) | 0.99 | 1.1 | 1.5 | 2.7 | 5.6 | 26.2 | $5.6 \times 10^6$ |
| PICOP, PICO boxes ($C_d^{1/2}\Gamma_{TS} \times 10^{-4}$, 250 bootstrap samples) | 0.96 | 1.0 | 1.5 | 3.2 | 6.0 | 72.5 | $1.0 \times 10^7$ |

For the box parameterisation, the distribution of parameters resulting from the bootstrap experiments also hints to an inverse quadratic relationship between the overturning coefficient $C$ and the effective turbulent temperature exchange velocity $\gamma_T^\star$, which seems consistent across the two different configurations presented here (Fig. 9b). The lower the former, the higher the latter, and the other way. The empirical fit resulting from this distribution is:

$$C = \left[\frac{4.8}{(\gamma_T^\star \times 10^5)^2} - 2.4\right] \times 10^6 \tag{39}$$

if $C < 0.1 \times 10^6$, $C$ is set to $0.1 \times 10^6$ \tag{40}

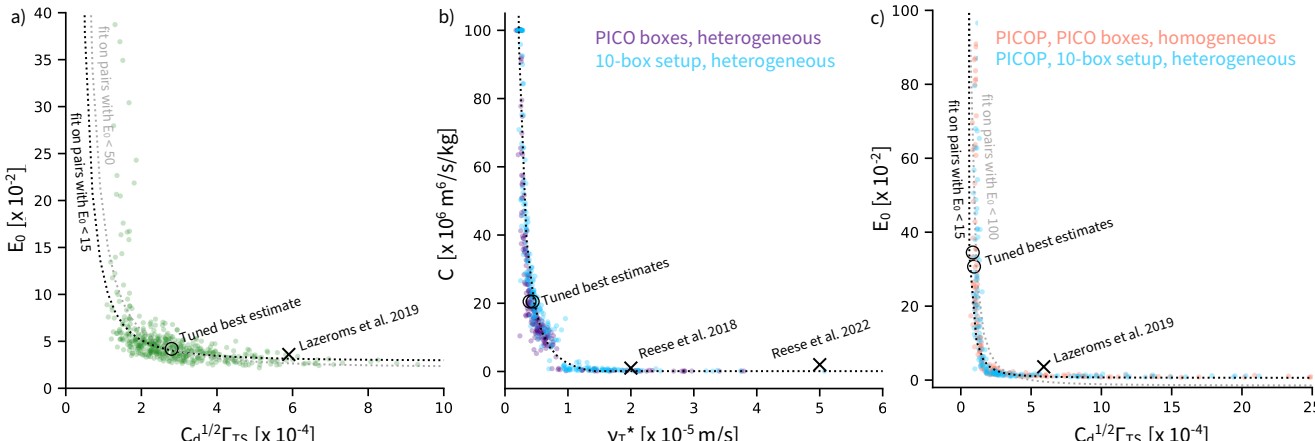

**Figure 9.** Distribution of the parameters resulting from the bootstrap experiments applied to the subset of complex parameterisations. (a) Distribution of $C_d^{1/2}\Gamma_{TS}$ and $E_0$ in 500 tuning experiments for the Lazeroms version of the plume parameterisation. (b) Distribution of $C$ and $\gamma_T^\star$ in 2×250 tuning experiments for 2 different configurations, respectively. (c) Distribution of $C_d^{1/2}\Gamma_{TS}$ and $E_0$ in 2×250 tuning experiments for 2 different PICOP configurations, respectively. An inverse quadratic function has been fit empirically for the different pairs of parameters. Note that we cut the axes for better visibility. For the plumes, 6% of the $E_0$ are larger than $40\times10^{-2}$ and 2% of the $C_d^{1/2}\Gamma_{TS}$ are larger than $10\times10^{-4}$. For the boxes, $C$ was constrained to values between $0.1\times10^6$ and $100\times10^6$ during tuning. For PICOP, 27% of the $E_0$ are larger than $100\times10^{-2}$ and 12% of the $C_d^{1/2}\Gamma_{TS}$ are larger than $25\times10^{-4}$. The circles represent the tuned "best-estimate" parameters and the crosses represent the parameters from previous literature.

The correction for the lower end of $C$ is introduced to avoid negative overturning coefficients.

One explanation for this behaviour is that, if the overturning coefficient is high, water masses are rapidly replaced and new heat to melt the ice is supplied more often. In that case, not as much turbulent heat exchange is needed at the ice-ocean interface to match the reference melting. Conversely, if $C$ is small, turbulent exchange has to be more efficient to extract enough heat from less rapidly changing water masses to lead to a similar melting.

Our best-estimate tuned parameters are located between the 33rd percentile and the median, while the parameters suggested by Reese et al. (2018a) and Reese et al. (2022) are close to our fit but cleary in the upper tail of the distribution. This might be a result of tuning towards two single ice shelves: Pine Island and Filchner-Ronne.

For the PICOP parameterisation, there is also a clear relationship between $C_d^{1/2}\Gamma_{TS}$ and $E_0$, which reflects a similar compensating behaviour as seen in the box and plume parameters (Fig. 9c). Like for the plume parameterisation, we suggest two fits. The inverse quadratic fit describing best the relationship for $E_0$ below $100\times10^{-2}$ (grey curve) is:

$$E_0 = \left[ \frac{37.7}{(C_d^{1/2}\Gamma_{TS} \times 10^4)^2} - 1.5 \right] \times 10^{-2} \tag{41}$$

Focussing on the pairs containing $E_0$ below $15 \times 10^{-2}$ (black curve), the relationship is:

$$E_0 = \left[ \frac{12.3}{(C_d^{1/2} \Gamma_{TS} \times 10^4)^2} + 0.6 \right] \times 10^{-2} \tag{42}$$

This relationship is different from Eq. (37) and Eq. (38), although these are supposed to be similar parameters. This highlights that these parameters potentially encompass different processes in the plume or in the PICOP parameterisation. Note that some of the higher values of $C_d^{1/2} \Gamma_{TS}$ and $E_0$ are several orders of magnitude higher than expected (see e.g. Table 12), which we cannot explain and therefore would not recommend using.

Our tuned best-estimate parameters for PICOP are located below the 5th percentile and not near the median. After a few experiments, we suggest that this is a consequence of the presence of a local minimum of the RMSE around Stanton numbers of $1.0 \times 10^{-4}$ and entrainment coefficients of $9 \times 10^{-2}$ and of a global minimum near the best-estimate parameters. Depending on which ice shelves are present in the synthetic sample, this local minimum becomes a global minimum. Our experiments (not shown) point to the Ross and Getz ice shelves mainly steering this behaviour. The optimal parameters are therefore very sensitive to a few specific ice shelves and their absence or (sometimes multiple) presence in the different synthetic samples can lead to large variations in the tuned PICOP parameters, making the circum-Antarctic tuning very uncertain.

For all complex parameterisations, the distribution of parameters is very large, showing that using constant parameters on the circum-Antarctic scale is challenging and uncertain. To sample this uncertainty in the parameters, we recommend to use the uncertainty intervals around the parameters presented in Table 12 in combination with the empirical relationships provided in Equations (37) to (42). Instead of varying both $\gamma_T^\star$ and C for the box parameterisation and both $C_d^{1/2} \Gamma_{TS}$ and $E_0$ for the plume and PICOP parameterisations, now only one of them needs to be varied and the uncertainty in both parameters can simultaneously be covered. Note, however, that these distributions might not be as robust as the distributions for the parameters of the simple parameterisations (as shown in Table 11) due to the much smaller sample size.

## 4.2 Recommendations and limitations

This study has pointed out different behaviours, and strengths and weaknesses for the different parameterisations. In the following, we discuss emerging recommendations and limitations for the use of the basal melt parameterisations in ice-sheet models.

### 4.2.1 Simple parameterisations

The simple parameterisations are most practical to implement in ice-sheet or ocean models. The good performance of the quadratic formulations is therefore a positive signal for further ice-sheet model development. In particular, we could show that the RMSE of the parameterised integrated melt is comparable when using the newly-tuned parameters and when using the parameters given as recommendations for ISMIP6 simulations for the "MeanAnt" case (see Jourdain et al., 2020). This means that basal melt rates in the standard ISMIP6 experiments are reasonably consistent with our 1/4° global ocean simulations. In addition, the use of the quadratic parameterisation with a constant Antarctic slope is especially promising because the RMSE remains low when evaluating both the performance on both a time block and an ice shelf "unseen" during tuning.

In contrast, the ISMIP6 parameters for the "PIGL" case and the parameters proposed by Favier et al. (2019) lead to high RMSE in our case, and therefore can only be used in combination with appropriate temperature corrections. Here we have decided not to use corrections as temperatures are perfectly known in our approach, but the PIGL corrections certainly partly have a physical origin (heat consumed by melt), which is not accounted for in our simple parameterisations.

We also found only modest improvements from the semilocal compared to the local parameterisations, which may be considered as a good reason to use the local version which is easier to implement. To our surprise, the inclusion of the slope gave poorer comparisons to NEMO at the scale of Antarctica, with a tendency to produce too strong melt rates near grounding lines. The slope-dependent parameterisation is nonetheless relatively good for capturing integrated melt rates and could be used to generate end members in ensemble simulations to include members with high melt rates near grounding lines.

Finally, the linear-local parameterisation yields the highest RMSE of all parameterisations and should therefore be replaced by a quadratic formulation in further ice-sheet model developments.

In regard to spatial patterns, the simple parameterisations do not exhibit any refreezing, while the box parameterisation results in large areas of refreezing. Currently, the ice shelves are largely melting so areas of refreezing might not be of interest for projections. However, further work is needed to characterise how important these regions are and, if they are crucial, how to better represent them in the different parameterisations.

### 4.2.2 Plume parameterisation

The results for the Lazeroms version of the plume parameterisation are very satisfying both for the integrated melt and the melt rate near the grounding line and suggest that it is a good approximation of the processes at work below the ice shelf. The modification, which takes into account more precisely the effect of upstream properties, does not lead to clear improvement in the RMSE in the cross-validation over time and clearly struggles to generalise to ice shelves "unseen" during training. At the same time, the quick check in Sec. 4.1.1 suggests that the modification adapts better to an increase in resolution. We therefore suggest, when possible, trying out both versions of the plume parameterisation presented here to increase our understanding of how they compare in different applications and which one to prefer in which context.

### 4.2.3 Box parameterisation

The box parameterisation is implemented as PICO in several ice-sheet models, most prominently in the Parallel Ice Sheet Model PISM (Winkelmann et al., 2011). Our study provides us with new insights into its limitations and potential improvements.

We find that the number of boxes slightly influences the tuning of $\gamma_T^\star$ and $C$ while there is nearly no difference between the tuned parameters using homogeneous or heterogeneous boxes. A higher number of boxes leads to a slightly lower RMSE. If the box parameterisation is used, we therefore recommend using a setup similar to the 10-box setup for all ice shelves. Nonetheless, this also means that if an implementation of the PICO boxes already exists in the ice-sheet model of interest, it is probably not worth changing that setup. Overall, the box parameterisation seems to be robust in its results, and shows no large variations when testing the number of boxes, the homogeneous or heterogeneous approach, and different ocean input regions.

While the box parameterisation mimics some of the physical processes at work (essentially advection and heat conservation), the resulting RMSE for the integrated melt is higher than the quadratic and plume parameterisations in most cases for the 50 km domain input. For the offshore input, it performs slightly better than the other parameterisations. The cross-validation shows that the box parameterisation struggles to generalise to ice shelves "unseen" during tuning for the integrated melt and 50 km domain input, which hints to some limitations. For the melt rate near the grounding line, the resulting melt rate depends less on the sample used for tuning as both cross-validations lead to similar RMSE.

We suggest that one limitation of the box parameterisation is using the ocean bottom temperature as input temperature. On the one hand, this means that this parameterisation requires less input than the other parameterisations that use vertical profiles of ocean properties. This is useful for applications where only sparse data is available such as paleo modelling studies. On the other hand, this means that this parameterisation does not react to changes in the water column above the bottom and this might explain the higher difference to NEMO melt rates in comparison to the quadratic and plume parameterisations. To investigate this limitation, we apply the input temperatures used for the box parameterisation (ocean temperatures at the average entrance depth of each ice-shelf cavity) as input for the simple parameterisations. We then re-tune the simple parameterisations over the full sample. The resulting RMSE using these re-tuned parameters for the simple parameterisations is about 10 Gt/yr higher than the RMSE shown in Sec. 3.1.2. Further in that direction, we also ran a tuning of the box parameterisation using the depth of the grounding line when it is shallower than the average entrance depth. In this case, the effect was a reduction in the RMSE by about 3 to 4 Gt/yr. One possible improvement for the future development is therefore refining the criteria for the ocean input.

Another limitation could be that the box parameterisation assumes a linear relationship between the thermal forcing and the melt, like in the linear-local parameterisation, which was shown to not adequately represent the melt (see e.g. Fig. 7). Our results suggest that incorporating a quadratic relationship in the further development of the box parameterisation might improve it significantly.

Note that the box parameterisation strongly underestimates the melt rates for Pine Island ice shelf (see Fig. 6 and 8) and Amundsen Sea ice shelves in general (Fig. E2) when using the newly tuned "best-estimate" parameters. This is a problem for ice-sheet models trying to reproduce the historical evolution. Our tuning approach was designed for an intercomparison on the circum-Antarctic scale. Alternative tuning approaches more focussed on the Amundsen Sea and the melt sensitivity can produce better results for that region, as shown in Reese et al. (2022). We thus recommend to carefully estimate the parameters for the box parameterisation with respect to the intended application.

### 4.2.4 PICOP parameterisation

Our results do not encourage the use of PICOP as a melt parameterisation on the circum-Antarctic scale. It represents well the melt rate near the grounding line, but performs less well for the integrated melt. The cross-validation also shows that it does not generalise well to ice shelves "unseen" during tuning for the integrated melt. In addition, the bootstrap experiments show that the uncertainty around the parameters is large and the tuning is particularly unstable. As it is based on PICO and the plume

parameterisation, any improvement in either of the two might improve PICOP's performance. Also, the way in which both are

900 combined might be improved as well.

### 4.2.5 Definition of input temperature and salinity

Our results clearly indicate that averaging temperatures and salinity on the continental shelf and close to the ice shelf front (50 km domain) give the best results in comparison to averaging offshore properties. This should be the way to derive temperatures from CMIP models which have relatively realistic properties on large portions of the continental shelf around Antarctica

(Purich and England, 2021). For coarser ocean models, like those used in climate models of intermediate complexity or the coarsest CMIP models, the parameter values tuned for offshore temperatures might be preferred as these models are too coarse to represent the continental shelf around most of Antarctica. A better approach for these very coarse models may be to complement the basal melt parameterisation with a sub-grid description of on-shelf processes.

Here, the parameter values have been tuned for yearly temperatures and salinity, and we therefore recommend to keep this

consistency. We have nonetheless shown that this fails to capture "mode 3" melt (Jacobs et al., 1992; Dinniman et al., 2016; Silvano et al., 2016) related to the seasonal variability for some ice shelves like the Fimbul ice shelf. Further work will be needed to evaluate other input frequencies, although this may require to retune the parameters and probably to estimate the lags between temperature and basal melt rate variations (see Holland, 2017).

Biases can also be introduced by the use of one single averaged input profile for a whole ice shelf, especially for large ice

shelves, under which complex circulation patterns can be found. However, as each ice shelf has a specific circulation pattern, it is not straightforward to define more precise regions of interest for the input properties that are applicable to all ice shelves. Further research in that direction could identify more delimited domains of inflow and strengthen the link between input properties and melt, and therefore reduce uncertainties in the use of parameterisations. At the same time, such regions might change in a changing climate. So, this identification should adapt to changing conditions.

Due to the limitations mentioned here and in Sec. 4.1.1 biases might remain between the properties in front of the ice shelf and basal melt rates if the newly tuned "best-estimate" parameters are applied to observational estimates. In that case, we suggest to nudge the melt rates towards observational estimates by using local temperature corrections as suggested in Jourdain et al. (2020) or by locally using other parameters taken from the uncertainty distributions presented in Sec. 4.1.3 and Sec. 4.1.4.

### 4.2.6 Other comments

All parameterisations investigated here heavily rely on a horizontally homogenous vertical circulation within the ice-shelf cavity. While the plume parameterisation and quadratic formulation take into account, to some extent, the horizontal component of the circulation of the water masses in their formulation of $U$, additional factors such as the asymmetry of the circulation related to the Coriolis force, tides, or irregularities in the sub-shelf bathymetry can also affect the thermal forcing at the ice-ocean interface. At the same time, as every ice-shelf cavity has a different geometry, such effects are challenging to parameterise in general

formulations. Further work is needed to overcome these challenges and include these effects into the melt parameterisations.

Also, our results and conclusions hold for circum-Antarctic applications, such as large-scale Antarctic ice-sheet simulations. We do not claim that the parameterisations performing best on the circum-Antarctic scale are also performing best for each individual ice shelf. Our cross-validation across ice shelves underlines that many of the parameterisations struggle to generalise to ice shelves not seen during training. Therefore, for regional studies, some parameterisations might perform better if tuned only for the region of interest. However, it is important to keep in mind that a parameterisation which struggles to generalise to different ice shelves will also be potentially prone to biases in changing climate conditions.

We also do not claim to have covered every possible melt sensitivity to input ocean properties. We use an ensemble of simulations to introduce variations in the input forcing (up to 2 K for some cavities) and therefore include the melt sensitivity to the input ocean temperatures in our tuning. Nevertheless, in some of the cavities, such as the ice shelves in the Weddell sector, the variations between the different simulations remain very small. The melt sensitivity to larger variations in all regions could be explored more in future work.

## 5 Summary & conclusions

In a perfect-model approach, we re-tune the most commonly used basal melt parameterisations and assess their performance in representing the melt at the base of Antarctic ice shelves, on a circum-Antarctic scale. Using cross-validation, we assess their performance to generalise to time blocks and ice shelves "unseen" during tuning. We also provide an uncertainty range for the tuned parameters.

We conclude that:

- Better performances are found when using input hydrographic profiles averaged over a domain of 50 km on the continental shelf in front of the ice shelf compared to averaged over an offshore domain, beyond the continental shelf.

- The tuned simple quadratic local and semilocal parameterisations using a constant Antarctic slope (i.e. no dependency on the ice slope), and the plume parameterisation yield the best compromise to represent well both integrated shelf melt and melt rates near the grounding line.

- If input is only available for the offshore domain, the box parameterisation, the PICOP parameterisation, and the quadratic parameterisation using one cavity slope per ice shelf yield the comparatively best results but with clearly lower accuracy than when using the better-performing parameterisations with 50 km input.

- Some parameterisations do not generalise well on ice shelves "unseen" during tuning. This shows that they might not have enough flexibility needed to adapt to changing conditions.

None of the parameterisations yields a negligible RMSE compared to the reference, with RMSEs on the same order as or even larger than the reference value. Parameterising basal melt therefore still remains a challenge. However, we are confident that ongoing development will further reduce uncertainties in the representation of basal melt rates in ice-sheet models. In particular, the growing number of high-resolution ocean simulations becoming available through the Ice Shelf Ocean MIP

(ISOMIP) and Marine Ice Sheet-Ocean MIP (MISOMIP) projects provides large amounts of data as a testbed to advance our understanding of basal melt and further refine basal melt parameterisations.

## Appendix A: More details about the NEMO configuration

The horizontal (vertical) advection of tracers is done using respectively a fourth order (second order) Flux Corrected Transport scheme based on Zalesak (1979). A free slip lateral boundary condition on momentum is applied with no slip condition applied locally at Bering strait, Gibraltar and along the West Greenland coast. A quadratic bottom and top (ocean/ice shelf interface) friction is used with an increased bottom friction in the Indonesian Throughflow, Denmark Strait and Bab-el-Mandel. The bottom (top) drag is set respectively at $1 \times 10^{-3}$ ($2.5 \times 10^{-3}$).

A polynomial approximation of the TEOS10 equation of state is used (Roquet et al., 2015). Internal wave mixing is parameterised following de Lavergne et al. (2016). Finally, in some simulations, a 3D temperature and salinity restoring of the Antarctic Bottom Water (AABW) is applied based on the method presented by Dufour et al. (2012). The data used for the AABW restoring are from Gouretski and Koltermann (2004). Other model setup choices as momentum advection, lateral diffusion of momentum and tracer, vertical mixing (TKE), convection, double diffusion, bottom boundary layer are as described 975 in Storkey et al. (2018).

The sea ice model used here is SI3 which is based on LIM 3.6 (Rousset et al., 2015). Five ice thickness categories are used to represent the sub-grid-scale ice thickness distribution. Halo-thermodynamics are represented with two ice layers and one snow layer. The ice dynamic is based on the modified elastic-viscous-plastic (EVP) rheology (Bouillon et al., 2013) or an adaptative EVP rheology (Kimmritz et al., 2016).

Freshwater fluxes and heat fluxes are represented as follows. A sea surface salinity restoring is applied toward 1980-2010 WOA2018 surface climatological salinity (Zweng et al., 2018) in order to avoid large drifts in the salinity and overturning circulation. The strength of the restoring is set to -166.666 mm day$^{-1}$ psu$^{-1}$ (piston velocity of about 60m/yr). In order to preserve coastal runoff, the restoring coefficient is fading toward the coast (length scale of 150 km as described in Dussin et al. (2012)). River runoff comes from the Dai and Trenberth (2002) climatology.

All the various settings are widely used parameters if not, reasonable choices. The isopycnal diffusion of 150 m$^2$/s (300 m$^2$/s) has been used respectively in Storkey et al. (2018) (Megann et al., 2014). AABW restoring has been used with success in Dufour et al. (2012). The use of an eddy induced velocity to parameterise eddy diffusivity in a 0.25° resolution simulation is debated. Storkey et al. (2018) in their global 0.25° resolution configuration do not use an eddy parameterisation but not in their next version (Storkey et al., pers. comm.). In a MITgcm simulation at similar resolution, Naughten et al. (2019) used such 990 a parameterisation. Change in the sea ice parameters has been done to increase Antarctic sea-ice production and indirectly HSSW production via brine rejection.

## Appendix B: More details about the ensemble of simulations

Here, we present some key indicators to assess the quality of the REALISTIC simulation in the vicinity of Antarctica. The ACC transport at Drake passage is about 125 Sv over the last twenty years of the simulation (Fig. B1c). It compares reasonably well with the estimates of 136.7±7.8 Sv derived from CTD and ADCP data (Cunningham et al., 2003). This low transport is mainly explained by the too light AABW on the shelf break. A restoring experiment (Dufour et al., 2012, , and WARMROSS) clearly shows that artificially maintaining the AABW decreases the spin-up time of the ACC and drives a stronger ACC at a level within the observation range. The strength of the two main gyres (Weddell Gyre and Ross Gyre) represented in the REALISTIC simulations is in agreement with previous NEMO simulations at similar resolution (Mathiot et al., 2011). The REALISTIC Weddell gyre strength (Fig. B1a) is however on the high side of observation-based estimates to 56±8 Sv (Klatt et al., 2005). REALISTIC Ross Gyre strength (Fig. B1b) is also on the high side compared to the 20±5 Sv SOSE estimates (Mazloff et al., 2010).

The REALISTIC simulation is able to reproduce a cold Ross and Weddell shelf and warm Bellingshausen and Amundsen Seas comparable to the WOA2018 observation (Fig. B2). The associated integrated basal melt for Ross Ice Shelf (RIS), Filchner Ronne Ice shelf (FRIS), Pine Island Glacier (PIG) and all Antarctic ice shelves is reasonably well represented comparing to observational (Rignot et al., 2013) estimates (Fig. B1d,e,f,g). With low melt rates and even weak freezing in the interior, the REALISTIC RIS basal melt pattern compares well against Rignot et al. (2013) estimates (Fig B3). We notice, however, too much melting on the East and West sides along the ice shelf front. The FRIS basal melt pattern is also well represented with respect to Rignot et al. (2013) estimates with freezing in the vicinity of the various rumples in the interior and along the East side of Berkner Island and stronger melting along the grounding line in the deepest part of the cavity and along the ice shelf front. For the PIG, it has been shown that the melt is driven by buoyancy plumes concentrated within the 20 km of the grounding line (Dutrieux et al., 2013) into 1 km scale channels. The buoyancy plume dynamics are not well represented in NEMO due to its z coordinates and the resolution needed to represent the channels is well beyond the resolution of most global ocean configuration. So, not surprisingly, we cannot expect to have realistic melt pattern close to the grounding lines. REALISTIC HSSW properties at the formation sites on Ross and Weddell continental shelf are slightly too fresh compared with WOA2018 (Fig. B1j,k) but salty enough to keep both RIS and FRIS driven by HSSW inflow (call mode 1 in Jacobs et al. (1992)). Similar HSSW has been modelised in Mathiot et al. (2017) at similar horizontal resolution with an ERA-Interim based atmospheric forcing. This is a known bias in NEMO simulations forced by reanalysis re-enforced by the fact West Antarctic Ice Shelf basal melt is on the high end compared to the observations (example with PIG in Fig. B1f). The REALISTIC Antarctic summer and winter sea-ice extents (Fig. B1h,i) are rather well represented compared to the satellite estimates Meier et al. (2017).

The Amundsen sea bottom temperature varies from a too cold state in COLDAMU to a too warm state (WARMROSS). WARMROSS also triggers a warm state over East Ross shelf and much fresher high salinity shelf water (HSSW). These changes in the shelf temperature lead to large variability in the integrated basal melt of the Ross ice shelf and Pine Island Glacier (Fig. B1e, f). In WARMROSS, the basal melt of the Eastern part of the Ross ice shelf is driven by warm modified circumpolar deep water (MCDW) intrusions and fresher HSSW compared to the REALISTIC run. In COLDAMU, the cooling

of the Amundsen and Bellingshausen Sea triggers a collapse in the basal melting of the ice shelves. The large variability between the simulations is also visible in the Antarctic integrated melt (Fig. B1g).

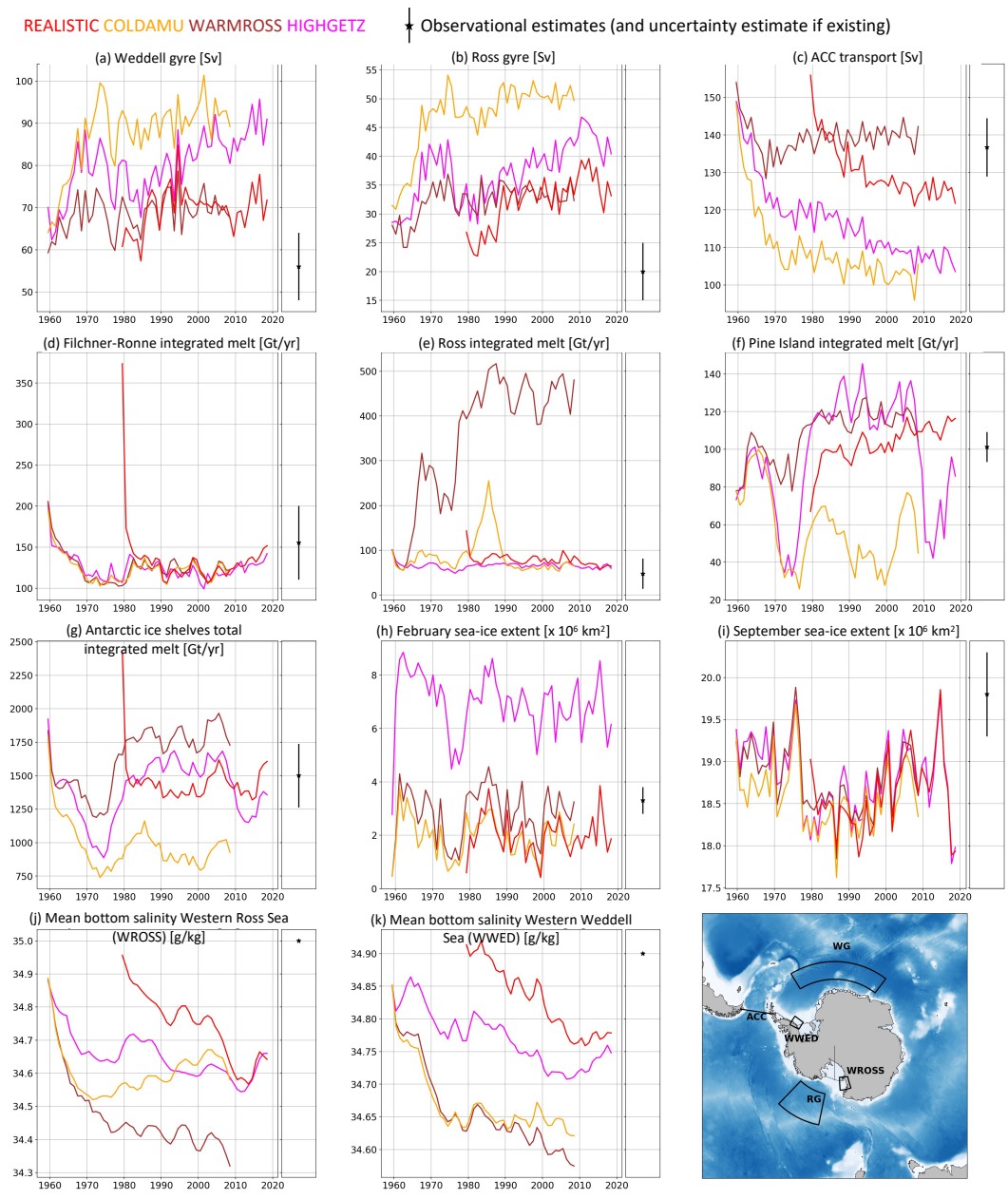

**Figure B1.** Time evolution of Weddell gyre (WG) strength (a), Ross gyre (RG) strength (b), ACC transport through Drake passage (c), Filchner-Ronne, Ross, Pine Island and Antarctica total integrated basal melt (respectively d, e, f and g), Antarctic sea ice extent in February (h) and September (i), and mean bottom salinity in regions of deep water formation WROSS (j) and WWED (k) shown on the map. In black: the observation estimates respectively from Klatt et al. (2005), Mazloff et al. (2010), Cunningham et al. (2003), Rignot et al. (2013) (d, e, f, g) and Meier et al. (2017) (h, i) and WOA2018 (1981-2010) by Zweng et al. (2018) (j,k). Panel (l) is a map showing the boxes used to compute the mean bottom salinity (j,k), gyre strength (a,b) and the Drake passage section used in (c).

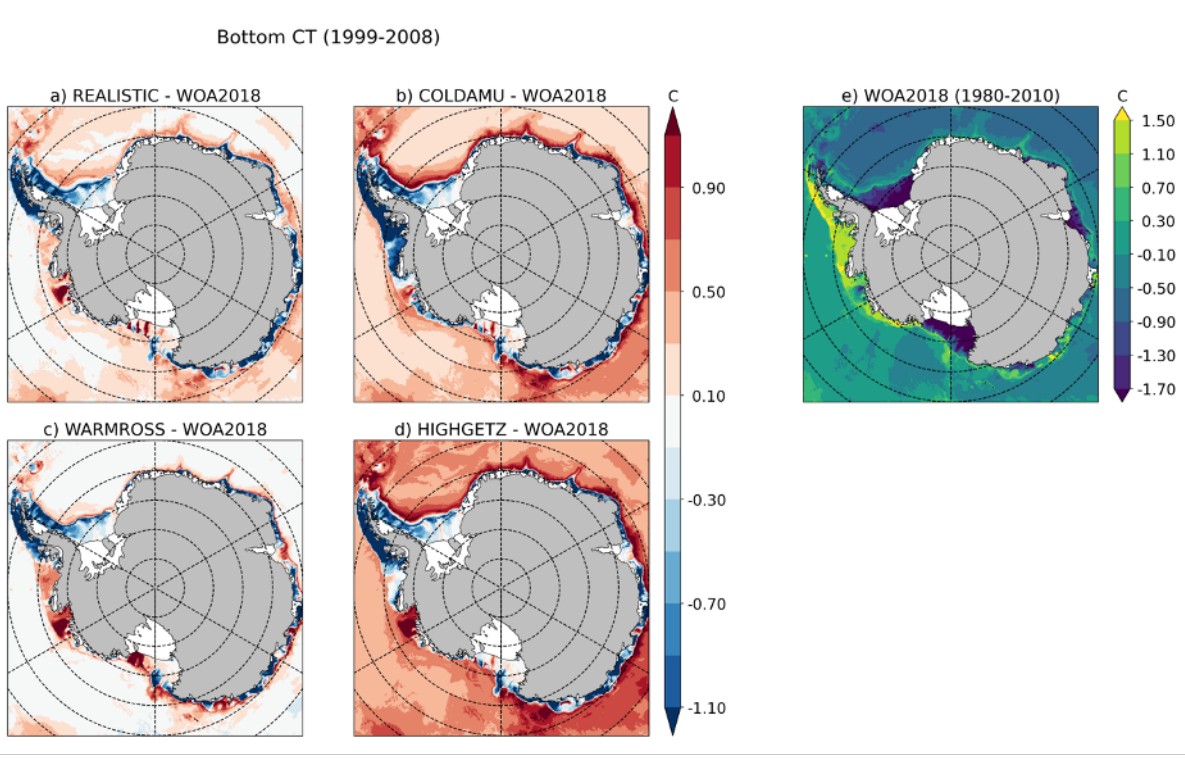

**Figure B2.** (a-d) Bottom conservative temperature (CT) averaged over the period 1999-2008 compared to WOA2018 bottom conservative temperature (1981-2010) [model - observation]. (e) Bottom conservative temperature in WOA2018 (1981-2010).

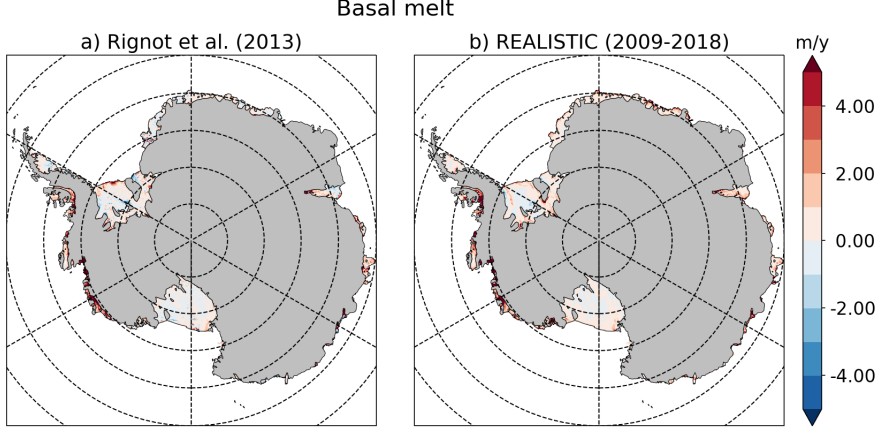

**Figure B3.** Comparison between basal melt patterns [in m/yr] from Rignot et al. (2013) and an average of our REALISTIC simulation between 2009 and 2018.

## Appendix C: Summary tables of the variables used in the description of the melt parameterisations

**Table C1.** Summary of definitions of physical variables used in the description of the melt parameterisations in Sec. 2.2. GL stands for grounding line, IF stands for ice front.

| Symbol | Variable | Unit | Introduced in ... |
|--------|----------|------|-------------------|
| $T_i$ | Ice temperature at the ocean-ice interface | °C | Eq. (6), can be ignored after scale analysis |
| $T_{\text{loc}}, S_{\text{loc}}$ | Temperature and salinity extrapolated from far-field to $z_{\text{draft}}$ | °C, PSU | Eq. (6) |
| $T_{\text{f, loc}}$ | Freezing temperature at $z_{\text{draft}}$, using $S_{\text{loc}}$ | °C | Eq. (6), Eq. (3) |
| $\langle T_{\text{loc}} - T_{\text{f, loc}} \rangle, \langle S_{\text{loc}} \rangle$ | $(T_{\text{loc}} - T_{\text{f, loc}})$ and $S_{\text{loc}}$ averaged over the ice shelf | °C, PSU | Eq. (17) |
| $T_{\text{cav}}, S_{\text{cav}}$ | Temperature and salinity extrapolated from far-field to $z_{\text{draft}}$ and averaged over the ice shelf | °C, PSU | Eq. (20) |
| $T_{\text{ups}}, S_{\text{ups}}$ | Average, for each point, of the portion of the far-field input profiles located between $z_{\text{draft}}$ and $z_{\text{gl}}$ | °C, PSU | Eq. (21) |
| $S_{\text{gl}}, S_{\text{deepest gl}}$ | Salinity extrapolated to $z_{\text{gl}}$ and to $z_{\text{deepest gl}}$, respectively | PSU | used for $T_{\text{f, gl}}$ and $T_{\text{f, deepest gl}}$ |
| $T_{\text{f, gl}}$ | Freezing temperature at $z_{\text{gl}}$, using $S_{\text{gl}}$ | °C | Eq. (20), Eq. (3) |
| $T_{\text{f, ups, gl}}$ | Freezing temperature at $z_{\text{gl}}$, using $S_{\text{ups}}$ | °C | Eq. (21), Eq. (3) |
| $T_{\text{f, deepest gl}}$ | Freezing temperature at $z_{\text{deepest gl}}$, using $S_{\text{deepest gl}}$ | °C | Eq. (26), Eq. (3) |
| $T_k, S_k$ | Temperature and salinity in box $k$ | °C, PSU | Eq. (29) |
| $T_{\text{f, k}}$ | Freezing temperature in box $k$ at either $\langle z_{\text{draft}} \rangle$ in $k$ (homogeneous boxes) or $z_{\text{draft}}$ (heterogeneous boxes), using $S_{\text{k}}$ | °C | Eq. (31), Eq. (3) |
| $U$ | Ocean current velocity in top boundary layer | m s$^{-1}$ | Eq. (4), Eq. (7) |
| $U_{\text{Ant}}$ | Characteristic $U$ over all ice shelves | m s$^{-1}$ | Eq. (9) |
| $\theta$ | Slope of the ice shelf relative to horizontal | degrees | Eq. (10) |
| $\theta_{\text{Ant}}$ | Constant average slope over all ice shelves | degrees | after Eq. (16) |
| $\theta_{\text{loc}}$ | Slope between the neighbouring grid cells in x- and y-directions at each point | degrees | after Eq. (16) |
| $\theta_{\text{cav}}$ | Angle opened by $z_{\text{deepest gl}}$, $\langle z_{\text{draft}} \rangle$ at the IF, and the maximum distance between GL and IF | degrees | after Eq. (16) |
| $\theta_{\text{loc, laz}}$ | Local slope for Lazeroms plume parameterisation | degrees | Eq. (20), Eq. (13b) in Lazeroms et al. (2018) |
| $\theta_{\text{ups}}$ | Angle opened by $z_{\text{gl}}$, $z_{\text{draft}}$ and shortest distance between GL and given point | degrees | Eq. (21) |
| $q$ | Overturning flux | m$^3$ s$^{-1}$ | Eq. (29), Eq. (30) |

**Table C2.** Summary of additional variables used in the description of the melt parameterisations in Sec. 2.2. GL stands for grounding line, IF stands for ice front.

| Symbol | Variable | Unit | Introduced in ... |
|---|---|---|---|
| Geometric properties | | | |
| $z_{\text{draft}}$ | Local ice-draft depth (negative below sea level) | m | Eq. (6) |
| $z_{\text{gl}}$ | Effective GL depth for Lazeroms plume parameterisation (negative below sea level) | m | Eq. (25), Eq. (13a) in Lazeroms et al. (2018) |
| $z_{\text{deepest gl}}$ | Deepest GL depth (negative below sea level) | m | Eq. (26) |
| $r$ | Relative distance to the GL at each point | | Eq. (27) |
| $A_k$ | Area of box $k$ | $\text{m}^2$ | Eq. (29) |
| Other variables | | | |
| $c_{\rho 1,\text{loc/cav/ups}}$ | Term in the Lazeroms version plume parameterisation | | Eq. (20), Eq. (21), Eq. (22) |
| $c_{\tau,\text{loc/cav/ups}}$ | Term in the Lazeroms version plume parameterisation | | Eq. (20), Eq. (21), Eq. (23) |
| $x$, $x_{\text{Lazeroms}}$, $x_{\text{modified}}$ | Characteristic length scale in plume parameterisation | | Eq. (24), Eq. (25), Eq. (26) |
| Tuning parameters | | | |
| $\gamma_{TS}$ | Turbulent exchange velocity, tuning parameter in linear parameterisation | $\text{m s}^{-1}$ | Eq. (7), Eq. (8) |
| $C_d^{1/2}\Gamma_{TS}$ | Stanton number, contained in tuning parameters $\gamma_{TS}$ and $K$ in simple parameterisations, tuning parameter in plume and PICOP parameterisation | | Eq. (6), Eq. (8) |
| $\epsilon$ | Parameter related to relative efficiency of mixing across thermocline between boundary current and warmer water below, contained in tuning parameter $K$ in quadratic parameterisation | | Eq. (11) |
| $K$ | Tuning parameter in quadratic parameterisation | | Eq. (16) |
| $E_0$ | Entrainment coefficient, tuning parameter in plume and PICOP parameterisation | | Eq. (20) |
| $C$ | Overturning coefficient, tuning parameter in box parameterisation | $\text{m}^6 \text{ s}^{-1} \text{ kg}^{-1}$ | Eq. (30) |
| $\gamma_T^{\star}$ | Effective turbulent temperature exchange velocity, tuning parameter in box parameterisation | $\text{m s}^{-1}$ | Eq. (31) |

**Appendix D: Number of boxes in the different box model configurations**

**Table D1.** number of boxes in the three different box model setups. When the number is smaller than the name of the setup, it means that the number of boxes needed to be reduced to have an ascendant slope or that the ice shelf is too small for this number of boxes.

| Ice shelf | 2 boxes setup | 5 boxes setup | 10 boxes setup | PICO setup |
|---|---|---|---|---|
| Filchner-Ronne | 2 | 5 | 10 | 5 |
| Venable | 1 | 2 | 3 | 2 |
| George VI | 1 | 1 | 2 | 3 |
| Abbot | 2 | 5 | 8 | 3 |
| Stange | 2 | 4 | 5 | 3 |
| Larsen C | 2 | 4 | 5 | 3 |
| Bach | 1 | 2 | 3 | 2 |
| Larsen D | 2 | 3 | 4 | 3 |
| Wilkins | 1 | 2 | 3 | 3 |
| Getz (REALISTIC, COLDAMU, WARMROSS) | 2 | 4 | 5 | 3 |
| Getz (HIGHGETZ) | 2 | 3 | 4 | 3 |
| Thwaites | 1 | 2 | 3 | 2 |
| Crosson | 2 | 5 | 8 | 3 |
| Dotson | 2 | 5 | 6 | 2 |
| Cosgrove | 1 | 1 | 2 | 2 |
| Pine Island | 2 | 5 | 6 | 2 |
| Ross | 2 | 5 | 10 | 5 |
| Cook | 1 | 2 | 3 | 2 |
| Nickerson | 2 | 4 | 5 | 2 |
| Sulzberger | 2 | 5 | 6 | 2 |
| Amery | 2 | 5 | 10 | 3 |
| Moscow University | 1 | 2 | 3 | 2 |
| Tracy Tremenchus | 1 | 1 | 2 | 2 |
| Totten | 2 | 5 | 9 | 2 |
| West | 2 | 4 | 5 | 3 |
| Shackleton | 2 | 5 | 10 | 3 |
| Ekstrom | 2 | 5 | 10 | 2 |
| Nivl | 2 | 5 | 7 | 3 |
| Prince Harald | 2 | 3 | 4 | 2 |
| Riiser-Larsen | 2 | 5 | 7 | 3 |
| Fimbul | 2 | 5 | 10 | 4 |
| Roi Baudoin | 2 | 5 | 8 | 3 |
| Lazarev | 2 | 5 | 9 | 3 |
| Stancomb Brunt | 2 | 5 | 10 | 4 |
| Jelbart | 2 | 5 | 7 | 3 |
| Borchgrevink | 2 | 5 | 9 | 3 |

## Appendix E:  Metrics on the ice-shelf level

Our study focusses on the circum-Antarctic performance of the parameterisations. The following figures provide a brief overview of the resulting performance on the individual ice shelves. Figure E1 shows the reference mean integrated melt and mean melt rate near the grounding line for the individual ice shelves in our NEMO ensemble. Figure E2 shows the average difference between the parameterised and reference integrated melt in the cross-validation, i.e. the average under- or overestimation of the integrated melt by each parameterisation for a given ice shelf when it was tuned over all other ice shelves. Figure E3 shows the average difference between the parameterised and reference mean melt rate near the grounding line in the cross-validation, i.e. the average under- or overestimation of the melt rate by each parameterisation for a given ice shelf when it was tuned over all other ice shelves.

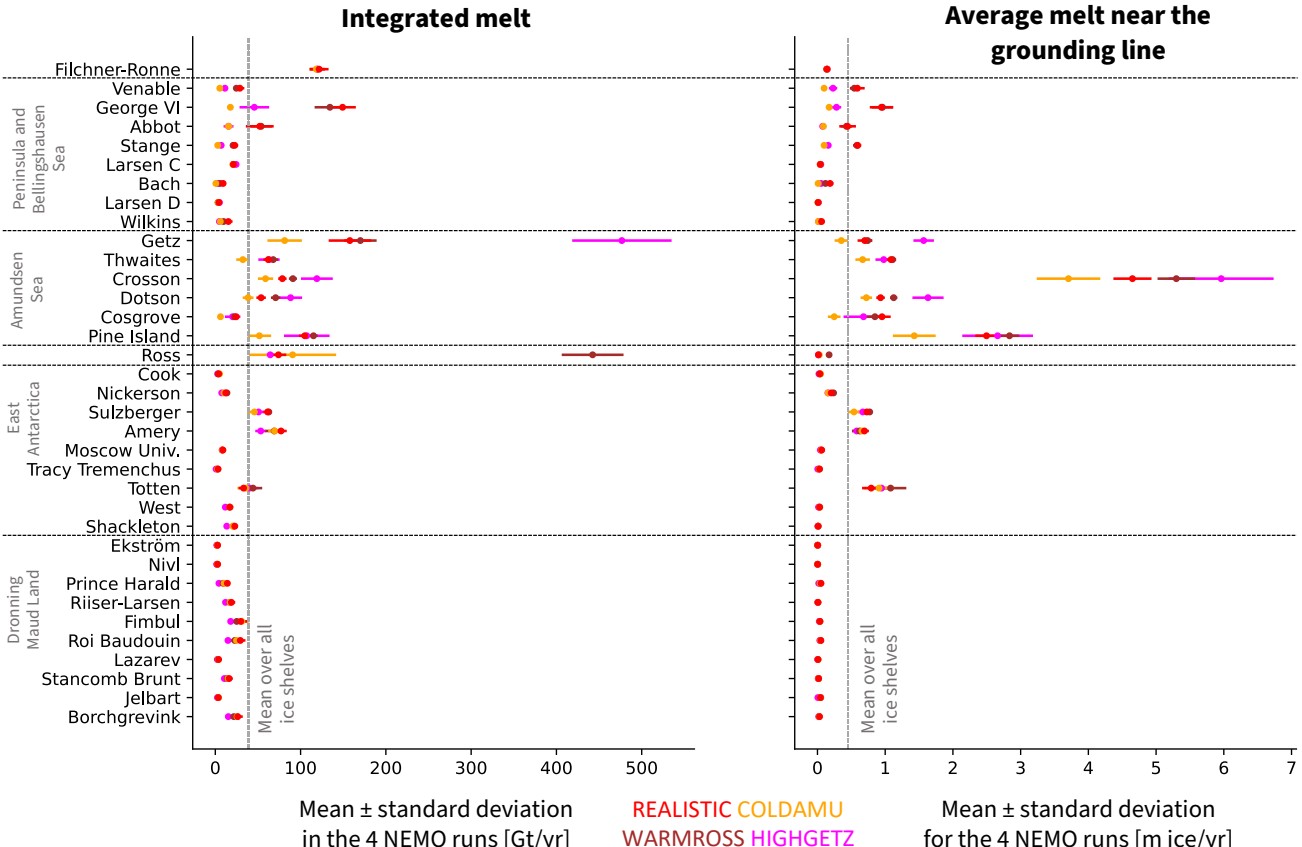

**Figure E1.** Reference integrated melt [in Gt/yr] (left) and mean melt rate near grounding line [in m ice/yr] (right) for the four simulations of NEMO.

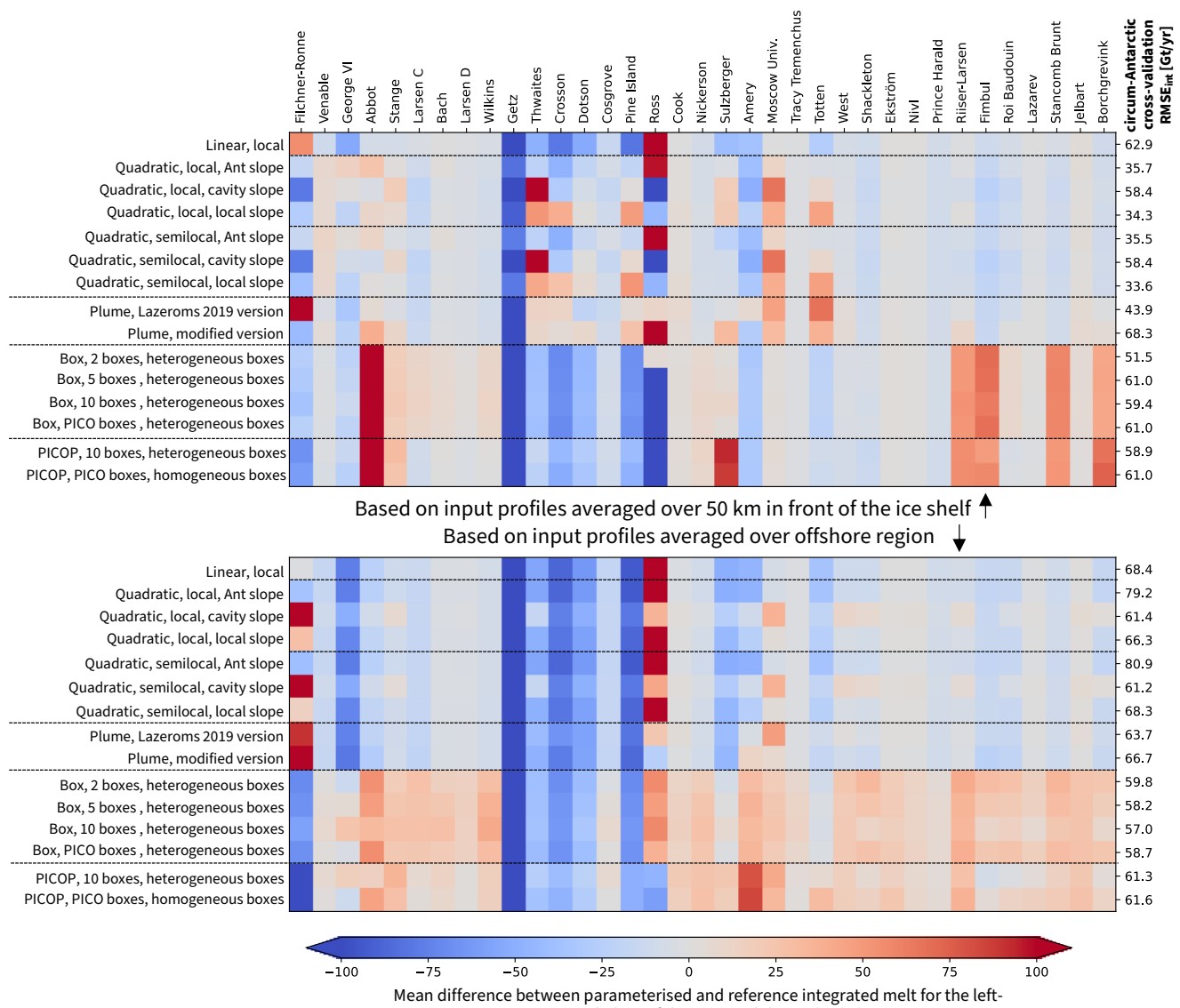

**Figure E2.** Mean difference between parameterised and reference integrated ice-shelf melt for the synthetically independent evaluation dataset for the cross-validation over ice shelves. The average is taken over yearly averages across all four simulations (127 years in total). The values on the right are the $RMSE_{\text{int}}$ for the cross-validation over ice shelves shown in Fig. 4. Results are shown for the 50 km (upper panel) and the offshore (lower panel) domains.

## Appendix F: Alternative statistical evaluation for the melt rate near the grounding line

We explored different statistical metrics to evaluate the melt rate near the grounding line. Like for the integrated melt, the question was if the RMSE should be conducted on the scale of ice shelves or on the grid-cell level. In Fig. F1, we show the

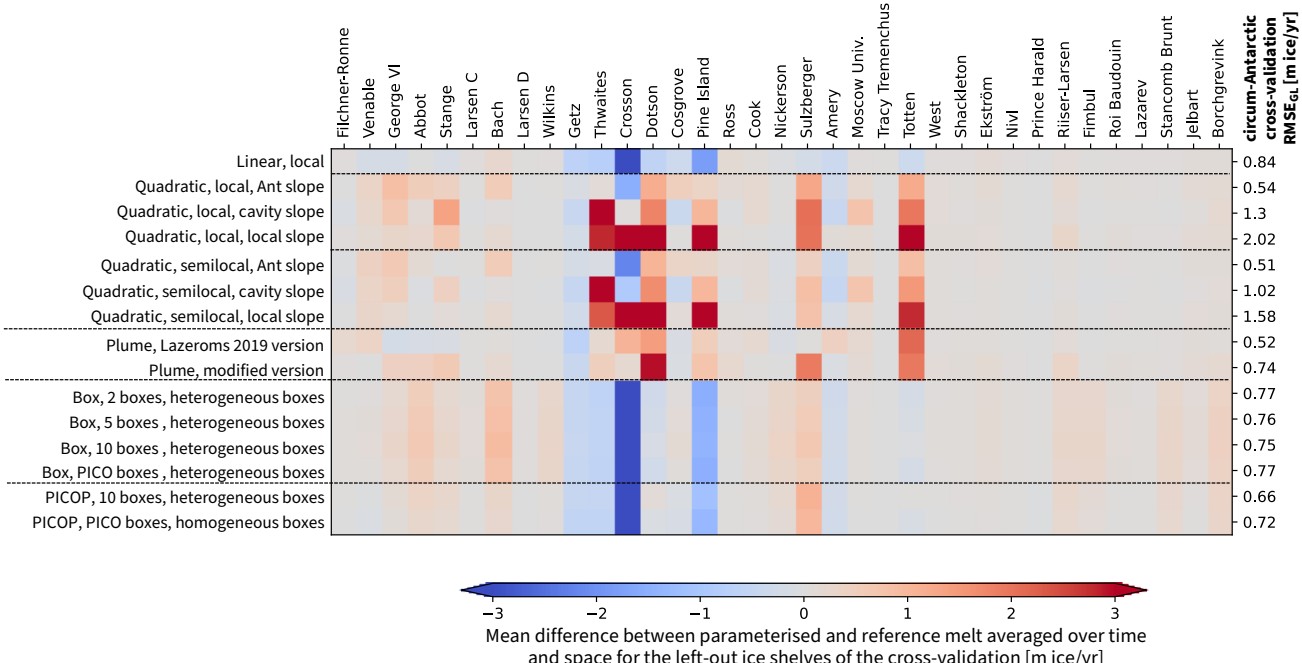

**Figure E3.** Mean difference between parameterised and reference melt rate averaged over time and space near the grounding line for the synthetically independent evaluation dataset for the cross-validation over ice shelves. The average is taken over the time-mean averaged spatially over approximately the 10% of the ice shelf nearest to the grounding line (area covered by the first box in the 5 boxes configuration of the box parameterisation) across the four simulations. The values on the right are the $RMSE_{GL}$ for the cross-validation over ice shelves shown in Fig. 7.

results using two different versions of computing the RMSe near the grounding line. On the left, we evaluate the RMSE of the time-averaged melt rate near the grounding line at the grid-cell level. First, the melt rate near the grounding line is averaged over time, for the parameterised and reference melt rate and for each ensemble run separately. Second, we take the difference

between parameterised and reference melt rate at each point and square it. Third, we average these squared difference over space and ensemble runs. Fourth, we take the square-root of these averaged squared differences. On the right, we evaluate the RMSE of the time- and space-averaged melt rate near the grounding line at the ice-shelf level. First, the melt rate near the grounding line is averaged over time and space, for the parameterised and reference melt rate and for each ensemble run separately. Second, we take the difference between these averaged parameterised and reference melt rates and square it. Third,

we average these squared differences over the ice shelves and ensemble runs. Fourth, we take the square-root of these averaged squared differences.

Generally, the RMSE evaluated on the grid cell level is higher than the RMSE_{GL} used in the main manuscript. It is between 1 and 3 m ice/yr. To put the values into context, the mean reference melt rate across all points near the grounding line is 0.17 m ice/yr. The RMSE is therefore high in comparison with the reference melt rate. Compared to the RMSE_{GL} used in

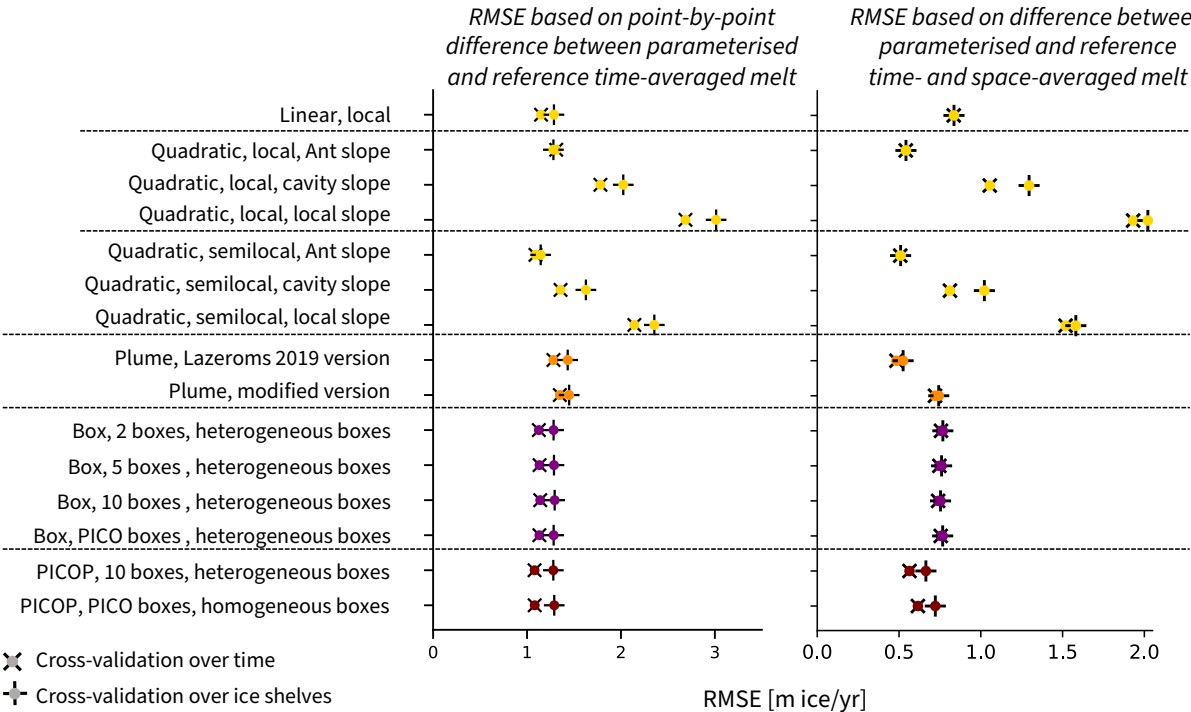

**Figure F1.** Comparison of two approaches to compute the RMSE of the melt rate near the grounding line ("near the grounding line" is defined as the first box in the 5-box setup of the box parameterisation). Left: First, the melt rate near the grounding line is averaged over time, for the parameterised and reference melt rate and for each ensemble run separately. Second, we take the difference between parameterised and reference melt rate at each point and square it. Third, we average these squared differences over space and ensemble runs. Fourth, we take the square-root of these averaged squared differences. Right: This is the method we use in the manuscript (same as shown in Fig. 7, right). First, the melt rate near the grounding line is averaged over time and space, for the parameterised and reference melt rate and for each ensemble run separately. Second, we take the difference between these averaged parameterised and reference melt rates and square it. Third, we average these squared differences over the ice shelves and ensemble runs. Fourth, we take the square-root of these averaged squared differences.

the main manuscript, the RMSE are closer between the parameterisations, with the linear parameterisation, the quadratic parameterisation using a constant Antarctic slope, the box parameterisation, and the PICOP parameterisation showing the lowest RMSE. The plume parameterisation has a slightly higher RMSE than the other parameterisations. In conclusion, evaluating the RMSE on the grid-cell level leads to slightly different conclusions than evaluating the RMSE on one averaged melt rate near the grounding line for each ice shelf. However, it does not alter the main conclusion, which is that using the quadratic parameterisation with Antarctic slope and the plume parameterisation is the best compromise when considering both integrated melt and melt rate near the grounding line.

*Code and data availability.* The simulation output, the data needed to produce the figures and tables, and the scripts can be found on Zenodo: https://doi.org/10.5281/zenodo.7308351. The package multimelt containing all parameterisations presented here is also available on PyPi: https://pypi.org/project/multimelt/.

*Author contributions.* CB and NCJ developed the original idea of this paper. CB carried out all analyses and wrote most of the manuscript. PM carried out the NEMO simulations and wrote the manuscript parts relating to the NEMO configuration and the description of the ensemble. AJ suggested the "modified" formulation of the plume parameterisation. CB, NCJ, RR, AJ, and PM contributed to discussions.

*Competing interests.* The authors declare that they have no conflict of interest.

*Acknowledgements.* We thank Xylar Asay-Davis and an anonymous reviewer for their very thorough review and constructive comments. 1070   We also thank the editor Reinhard Drews for remarks and comments. We thank David E. Shean for sharing his observational estimates of the basal melt rates of Pine Island Ice Shelf used in Fig. 8 with us. We also thank Jérémie Mouginot for sharing the melt patterns updated from Rignot et al. (2013) used in our Fig. B3. We deeply thank Tobias Finn for his invaluable support in the jungle of statistics. Most of the computations presented in this paper were performed using the GRICAD infrastructure (https://gricad.univ-grenoble-alpes.fr), which is supported by Grenoble research communities. The NEMO simulations were performed using HPC resources from GENCI–CINES (MISOCS 1075   project, allocations A0080106035 and A0100106035). This publication was supported by PROTECT. This project has received funding from the European Union's Horizon 2020 research and innovation programme under grant agreement No 869304, PROTECT contribution number 49. This research was also supported by the European Union's Horizon 2020 research and innovation programme under grant agreements no. 820575 (TiPACCs) and 101003536 (ESM2025).

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
