# Peer review of "An assessment of basal melt parameterisations for Antarctic ice shelves"

_The Cryosphere, 2022_

## Referee Comment (RC2)

**Review of Burgard et al. "An assessment of basal melt parameterisations for Antarctic ice shelves"**

Reviewer: Xylar Asay-Davis

I wish my name to be relayed to the authors, as I feel I am always a better reviewer when I am not anonymous and I encourage others to consider reviewing non-anonymously whenever they feel able.

**General Comments:**

For this first time, this paper performs thorough evaluation of a large number of ice-shelf melt parameterizations in a pan-Antarctic context. This is a topic of significant importance to the community, as melt parameterizations allow for Antarctic ice-sheet modeling on time scales that are not currently possible using ice-sheet models coupled to ocean or Earth system models. A number of aspects of this work are novel. First and most importantly, the evaluation is performed on a large number (36) of Antarctic ice-shelves that are fully representative of the continent and which use realistic topography and climate forcing. Second, the rich NEMO model ensemble provides over 500 years of cumulative simulation results, a far more robust dataset than current observations are able to provide. Third, new variants of melt parameterizations (including various ways of including ice-shelf slope dependence and a new approach to plume parameterization) are introduced for inclusion in the evaluation. Fourth, the paper carefully evaluates each parameterization for continental-scale "biases" (relative to the NEMO model), "biases" for individual ice shelves, melt patterns (with an emphasis on grounding lines) and the behavior of parameterizations at high resolution. Finally, the paper includes invaluable recommendations on which parameterizations show the most promise (in a few different categories) and the caveats and best practices for using each, suggested by the evaluation.

The paper is well written and arranged. The figures are quite effective, well crafted, and easy to follow. They give the reader a good, intuitive sense of the relative strengths and weaknesses of the different parameterizations in a variety of ways. The numerous tables will help future users of these parameterizations to know the correct coefficient values to use for each, including recommended values of the tunable parameters that best fit the NEMO simulation results.

I do have a few major concerns about the methodology, and addressing these concerns may entail a significant effort by the authors. For that, I am sorry. But I also feel a duty to be particularly careful in evaluating the methodology and suggesting improvements because of the way that the paper is framed: it aims to provide recommended best practices, including preferred coefficients for use in pan-Antarctic modeling using the evaluated parameterizations.

The first concern is one I share with the other reviewer. It would be preferable to tune the parameterizations with one subset of the available data and evaluate it with another, independent subset. Given the more than 500 years of simulation data, I agree that this

would seem to be fairly easy to do by using a random subset of years for tuning and the remainder for evaluation. It may be worth adopting this approach in revising th paper.

The second major concern is that the metrics used in the paper for evaluating the root-mean-squared "error" (RMSE) relative to the NEMO results, Eqs. (32) and (33), do not seem robust to me, nor are these definitions the standard ones used in Earth system modeling. This is especially unfortunate because the vast majority of the results show in the paper (from section 2.4 onward) are dependent on these metrics, including the process for identifying the best-fit parameters, the evaluation of parameterizations on the continental scale (both over each ice shelf and near grounding lines), and the methodology for evaluating parameter uncertainty. I am confident that the adoption of a more standard metric is unlikely to change the broad conclusions of the paper but it is hard to see how it would not change specific results such as the recommended values of coefficients used in the parameterizations. For this reason, I ask the authors to strongly consider redoing their analysis with a more robust and widely accepted version of the RMSE metric.

In my own work and I think in climate modeling broadly, we always use the area-weighted variance to compute the RMSE as follows:

$$RMSE = \sqrt{\frac{\sum\limits_{i=1}^{N}\left(m_i - m_{o,i}\right)^2 A_i}{\sum\limits_{i=1}^{N} A_i}}, \tag{1}$$

where, in the context of this paper, $m_i$ is the melt rate (in whatever units you choose) from the parameterization for sample $i$ (which might be one location and year); $m_{o,i}$ is the "observed" melt rate from NEMO (at the same location and year); $A_i$ is the area associated with the $ith$ sample (this could be a whole ice shelf or an individual NEMO grid cell, depending on how the evaluation is performed); and $N$ is the number of sample points. This metric is robust to different ways of sampling the evaluation points. It works just as well to evaluate the "error" over a single ice shelf as for any number of ice shelves aggregated together into a region. The RMSE values for individual ice shelves can be computed by sampling the pattern at locations across the ice shelf, and then these RMSE values can be combined to get regional- and continental-scale values of the RMSE. This approach is also robust in the sense that including or excluding any number of small ice shelves would have a small impact on the RMSE.

Unfortunately, the same is not true for either of the RMSE definitions, Eqs. (32) and (33), proposed in the paper. Instead of weighting by area, these metrics treat each ice shelf (independent of its area) as having equal weight. This approach has at least three problems. First, the metrics are significantly more sensitive to the number of ice shelves under consideration. Considering Eq. (32) first, repeated here:

$$RMSE_{\text{int}} = \sqrt{\frac{\sum\limits_{i}^{N_{\text{isf}}} \sum\limits_{t}^{N_{\text{years}}} (M_{\text{param}}[i,t] - M_{\text{ref}}[i,t])^2}{N_{\text{isf}} N_{\text{years}}}} \tag{2}$$

if we were to add several ice shelves with both negligible bias in melt flux and negligible area, the *RMSE~int~* would decrease (because we aren't adding significantly to the numerator but we are adding new ice shelves to the denominator). Now, considering Eq. (33):

$$RMSE_{\mathrm{GL}} = \sqrt{\frac{\sum\limits_{i}^{N_{\mathrm{isf}}} \sum\limits_{n}^{N_{\mathrm{simu}}} (m_{\mathrm{GL,param}}[i,n] - m_{\mathrm{GL,\,ref}}[i,n])^2}{N_{\mathrm{isf}} N_{\mathrm{simu}}}} \tag{3}$$

In this case, if we were to add several tiny ice shelves with negligible area but significant biases in melt rate, they could significantly increase the *RMSE~GL~*. In this way, these metrics are likely fairly sensitive to the number of ice shelves (36) chosen for the study.

Second, the metric in Eq. (2) here gives significantly more weight to ice shelves with large areas (since $M_i = m_i A_i$, the numerator in this equation is getting weighted by $A_i^2$), while the metric in Eq. (3) here gives significantly *less* weight to large ice shelves (because $A_i$ does not appear in the numerator at all).

Third, the definition of an ice shelf is not particularly robust. The Filchner and Ronne Ice Shelves are sometimes considered distinct (as in this paper) and sometimes are combined together. Similarly, the Eastern and Western Ross are sometimes considered separately (though Ross is a single ice shelf here). As grounding lines move, ice shelves that were once distinct may merge, or a single ice shelf may be divided into two. The RMSE metrics from the paper are sensitive to the authors' definition of the 36 ice shelves used in the paper, whereas the metric in Eq. (1) here would not change if a nice shelf were divided into 2.

In summary, I do not think robust statistics can be performed when treating ice shelves as distinct entities given equal weight regardless of size. I do not see how a meaningful RMSE in the integrated melt flux can be computed between ice shelves: it would only be meaningful if totalled over all ice shelves in a region (but averaged over samples in time and across ensembles).

My preference would be that the results of the paper be re-evaluated with the metric in Eq. (1) here instead of Eqs. (32) and (33) in the paper. I realize this is a monumental task. If the authors do not feel that this is necessary, I am willing to hear you out. But I would need to see a strong case for why the existing metrics are more robust and defensible than I currently believe.

I want to reiterate that I think this is amazing work! If one takes the metrics for evaluating "biases" as given, the rest of the results follow nicely: the evaluation of each parameterization is remarkably thorough and well executed. The service that this work does for the community is truly commendable, particularly because it gives future model developers not only the tools to make an informed choice among the existing parameterizations but also a good choice of coefficients to use.

Below, I have a number of specific comments and suggestions for typographical or grammatical changes. Please do not be intimidated by the number of comments. I have

been told by editors in the past that I seem to review papers as if I were a coauthor, which may be true.

**Specific Comments:**

l. 7-8: "The box, PICOP parameterization and quadratic parameterizations with slope dependency…" In the context of the abstract (in which these different parameterizations have not yet been introduced), I found it hard to understand what was being described here. The previous sentence does a nice job of introducing what the quadratic parameterization means and what the slope dependence is about. A similar introduction for the box parameterization and PICOP here would be helpful.

l. 19-20: "...interacting with land or shallow rocks…": I'm not sure I follow what you mean by interacting with shallow rocks. Maybe take this out?

l. 38: "...and do not resolve ice-shelf cavities…": Should this be "include" instead of "resolve"? It seems like you are referring to model configurations that don't have the cavities at all, rather than that they are there but at too coarse a resolution.

l. 66-69: When you refer to a "step" here, does each of these correspond to a section of the paper? If so, it might be clearer to replace "step" with "section".

l. 115 "The top boundary-layer thickness is set to 30 m." This likely needs a little more explanation: this is referring to the boundary-layer thickness over which temperature and salinity are averaged for use in the parameterization of sub-ice-shelf turbulent fluxes.

l. 147: "basal mass loss" might be better as "basal mass flux" or something like that because usually "mass loss" is used in our field to refer to the uncompensated loss of mass from the ice sheet that contributes to sea-level rise.

l. 188-191: Here, a bunch of physical constants and variables are introduced for the second time (they were introduced on l. 104-115). I think the only new ones here are $\Gamma_{TS}$, $T_i$ and $U$. Also, it seemed to me that $U$ might be the same as $u_{oc}$ introduced above. If so, maybe use the same symbol for both. If not, maybe clarify the distinction.

l. 213: "where the subscript *oc* denotes the far-field temperature and salinity that is advected into the cavity". This seems like a different definition than on l. 105: "$T_{oc}$ and $S_{oc}$ are the temperature and salinity averaged over a boundary layer below the ice shelf." It would be helpful if the *oc* subscript meant the same thing throughout the manuscript. Maybe another subscript like *ff* could be used for far-field quantities (or *bl* for boundary-layer quantities)?

l. 217: The liquidus slope has already been defined on l. 108.

l. 240-240: "However, through the resulting circulation, the temperature at one point of the ice shelf cavity is not necessarily independent from the temperature at the other points of the ice shelf cavity." This may be correct, but I understood the quadratic dependency a little differently. I thought the concept was that the overturning itself was driven by the aggregate

effect of the thermal forcing (so that the strength of the overturning circulation would be proportional to the mean thermal forcing in the cavity). Thus, the local velocity is represented in terms of the amount of heat available for melting and there isn't necessarily any implication that the temperatures at different parts of the ice shelf are correlated to one another.

Eq. (17): It seems like some explanation of the bracket < > notation is needed.

Eqs. (20) and (21): I think these should be moved after Eq. (24) (with Eq. (20) after Eq. (21)) because they don't currently have any context where they are.

Eqs. (23) and (25) are identical, right? If so, I don't think it's necessary to have Eq. (25), you can just state that $M_2$ is still given by Eq. (23).

l 311-312: "If one of the boxes has an area of zero…": It isn't clear to me how this could happen. Is this just for an ice shelf that is poorly resolved so that there aren't enough grid cells for each box to get some?

l. 313: "are non-zero": Should this be "have a non-zero area?"

l. 349-350: "We use yearly mean profiles as the residence time of water in ice shelf cavities might be longer than a month for some cavities." You explain this more in the discussion but it might be worth giving it more context here, too. The parameterizations do not include the finite advection time from the ice-shelf front to the ice-shelf base, meaning they are missing important delays that are present in the physical system. This means there is little hope of capturing seasonal melting in the proper phase.

l. 431: "We therefore only show results for heterogeneous boxes **hereafter**." This is not true in the next subsection when you talk about PICOP, so maybe change "hereafter" to "in the remainder of this subsection."

l. 458-460: "To put these RMSE in context, the integrated ice-shelf melt averaged over all ice shelves and simulation years in the reference is 38 Gt/yr, which means that even for the better-performing parameterisations, the RMSE remains very high compared to the reference melt as such."

I have concerns about this concept, related to my concerns about the metrics in Eqs. (22) and (23). I do not think it is meaningful that the total Antarctic melt flux (~1500 Gt/yr) averaged over 36 ice shelves (an arbitrary number related to the resolution of the NEMO simulations used for reference) comes out to ~38 Gt/yr. The different ice shelves have such different areas and melt rates that this average isn't informative. Given that I have problems with the metric from Eq. (22) that you're given context to, this might be a moot point but if you want to give context to what ~40 Gt/yr is like, maybe just point out an ice shelf by name that has about that much melting.

l. 504-505: "...likely because they are all forced by yearly ocean temperatures." This seems to assume that the parameterizations would be able to get the Mode 3 melt if they were forced with seasonal or monthly temperatures. As you state later, they are unlikely to have

the correct processes or the correct seasonal phasing to get these melt rates right even if temperature forcing were provided on a shorter time interval.

Figs. 4 and 7: I appreciated this visual representation of the RMSE – it was very intuitive.

Figs. 5 and 6: These are wonderfully done and very clear!

l. 575-576: "...and is composed of 36 random samples, with replacement, of the different ice shelves." Do I understand correctly that you double or triple (or whatever) count some ice shelves and omit other ice shelves in this process? That seems like an uncommon practice and that the approach (suggested by reviewer 1) of using only some of the years for training and holding back others for validation would be a more standard method for testing the robustness and parameter uncertainty.

Fig. 9a. On the light gray curve it says "fit on pairs with $E_0$ < 50" but in the text describing Eq. (34), I didn't see anything stating that it was for E0 < 50. Is (34) not for the gray curve? Is this description simply missing from the text?

l. 617-619: "Note that some of the higher values of $C_d^{1/2}\Gamma_{TS}$ and $E_0$ are several orders of magnitude higher than expected (see e.g. Table 12), which we cannot explain." This seems odd. Don't you have constraints on the parameters to prevent them from varying outside a physically likely range?

l. 648: "The slope-dependent parameterisation is nonetheless relatively **good for cavity melt rates…**" Do you mean "...good for capturing melt patterns" or something like that?

l. 744-745: "...with RMSEs on the same order as or even larger than the reference value." This is another reference to ice-shelf-by-ice-shelf statistics that I think needs to be rethought in terms of area-weighted statistics.

Appendix D: Could you say how the RMSE for the heatmaps are defined? Are these the same as Eq. (32) but with $N_{isf}$ = 1?

**Typographical and Grammatical Suggestions:**

l. 11: "Additionally to…" should be "In addition to…"

l. 28: "uncertainty source" should be "source of uncertainty"

l. 140: Probably remove "For example", since this sentence didn't clearly follow (at least for me) as an example of the changed parameters not having a significant impatct on the physical ocean.

l. 204: "from an ice shelf to another" should be "from **one** ice shelf to another".

l. 252: "(based on Sec. 2.2.2)" might be more natural as "(as described in the next section)".

l. 288-291: This is a lot in one sentence and gets a bit confusing. It might be clearer if you break the details of each subscript into its own sentence.

l. 308, 314, 315, 437, 444, 446, 575, 589, 666, 667, 747, Appendix C, Table C1: The word "amount" is used in several places where "number" is correct (because the object being referred to is countable, rather than indefinite): "number of boxes", "number of data points", "number of high-resolution ocean simulations", etc.

l. 381-382: "two constraints are additionally needed" would be better as "two additional constraints are needed"

l. 399, 434: "(2020)(Table 4, 1st and 3rd columns)" and similar: This is very picky on my part but if there's a way to avoid back-to-back parentheses like here, I would prefer it.

l. 404-405: "The new tuning only slightly achieves to reduce the RMSE further" would be better as "The new tuning achieves only a slight ruther reduction in RMSE"

l. 414-415 and 449: "input properties are **of** a similar order of magnitude" and "all **of** a similar order of magnitude"

l. 438: "...about twice as high **as** the…"

l. 454: "Slightly above…" should be "At slightly above…"

l. 460, 517, 565: "... to the reference melt **as such**." and "...the melt patterns **as such**". The phrase "as such" is used in a few places in the paper where I don't fully understand it. Maybe you mean "itself"? Maybe this can just be omitted?

l. 506: "additionally" should be "in addition"

l. 519: "This can be explained by **punctual** very strong melt". I do not know for sure what is meant here by "punctual". This usually means "on time".

l. 521: "should therefore be kept" would be a bit better as "is important to keep" and "On the opposite" should be "In contrast".

l. 527: "...in a more consistent **way** and with larger…

l. 543: "emulates well NEMO" should be "emulates NEMO well"

l. 549-550: "...and therefore  is a result of the circum-Antarctic tuning **rather** than of the resolution"

l. 550 "...on the contrary" should be "in contrast"

l. 568: "...require to account for…" should be "...require accounting for…"

l. 616: "This relation is different **from** Eq. (34)..."

l. 627: "On the opposite" should probably be "In contrast"

l. 637: "Especially" should be something like "In particular"

l. 648: "...used as end members" might be "...used to generate end members…"

l. 660-661: "We therefore **suggest, when possible, trying** out…"

l. 694: "...any improvement in **either** of the two…"

l. 716: "...take into account**, to some extent,** the horizontal…"

l. 719: missing a comma after "geometry"

---

## Author Comment (AC1)

**Author Response to Reviewer #1**

**An assessment of basal melt parameterisations for Antarctic ice shelves**

Burgard, C., Jourdain, N.C., Reese, R., Jenkins, A., Mathiot, P.
*The Cryosphere,* `#10.5194/tc-2022-32`
* * *
**RC: Reviewer Comment**,     AR: Author Response,     *changed manuscript text*

RC: **Reviewer Summary:**
**The manuscript by Burgard et al carries out a comprehensive analysis combining (I think) all of the leading parameterisations of ice-shelf melt that are currently used, with a set of global ocean model simulations, in which the parameterisations are rigorously analysed in terms of their ability to replicate ocean modelled melt rates given open-ocean properties, their stability in terms of optimal parameters, and their benefits and drawbacks in terms of use. I think this is a great study to have been carried out, as to date no other studies have collected all of the extant parameterisations together in one study, implemented them in a common framework, and tuned and evaluated them with identical data. The study also does not ignore the importance of spatial patterns of melt arising from the parameterisation, which are too often overlooked. Though it is a very long paper, it is formulaic and there is a progression in terms of the experimental setup and analysis, making it a less daunting read. The length is also owed to its comprehensive discussion of existing parameterisations, and any modifications made to them as part of this study, and it is really good to have all of this material together in one place. I think this is a worthwhile and interesting study, as it will be important to determine how the Antarctic ice sheet will evolve in response to oceanic change, and it is clear that ocean models which can resolve under-ice shelf circulation are the rate-limiting step in such investigations. Therefore the lessons learned from this study are valuable and I recommend for publication after some minor revision. I have comments below that I hope might help in this regard.**

AR: Thank you very much for the positive feedback and for your constructive comments on how to further improve our manuscript. We plan to address all your comments as described in the following.

RC: **Side note: (On a side note, the python library developed for this study will be of value as well, although its value may depend on how easily it can be implemented with C++ and Fortran ice-sheet models. However this is not directly relevant to the merits of the manuscript.)**

AR: We agree that our package is not designed for direct implementation into an ice-sheet model code-wise. Nonetheless, it could be used as an offline python interface between a coarse ocean model and an ice-sheet model (both of them possibly written in Fortran or C++).

RC: *I have only two general comments, and it is with regard to the global simulations used to force and tune the parameterisations and the assessment of parameterisation skill:*

RC: **With a resolution for the ensemble of 8 km at 70 S, this is quite coarse for a simulation that is meant to provide "truth" for ice-shelf melt in response to conditions on the shelf and the ACC. It is true the conditions on-shelf are also not necessarily realistic, but it is the continental-shelf-to-melt dependence that is important here. For instance 8 km is well above the deformation scale, so I question the model's ability to represent boundary currents that bring warm water into cavities and melt-laden water out, and transport around bathymetric obstacles and through bathymetric depressions, and these could potentially impact total melt, rather than just melt patterns. I think this potential caveat, as well as those mentioned in the discussion, should be more clearly stated up front in section 2.1.**

AR: We agree that a resolution of 8 km is still not perfect to reproduce the ocean circulation on the continental shelf. Nevertheless, it is a clear improvement compared to the data available typically used to force ISMIP6-type ice-sheet models, which comes from climate and ocean models that represent the topographic dynamical features even more crudely (typically at a resolution of $1°$). In our study, we focus on the link between the domain in front of the ice shelf and the melt inside the cavity. Therefore, very simply said, if the "wrong" water is in front of the cavity compared to reality, it will lead to a "wrong" melt compared to reality but the physical link between the two, which we are interested in, will be consistent.
In regard to the ocean circulation inside the cavity, we agree that we do not resolve all bathymetric ridges, basal channels and eddies that potentially affect the melt locally but also on the integrated level. Also, the resolution is not as high as ice-sheet models directly next to the grounding line. We will add these points to the list of caveats intoduced by NEMO listed in Sec. 4.1.1. We will also move Sec. 4.1.1. to the end of Sec. 2.1. so that the reader has the limitations in mind when interpreting the results.

RC: **I may have misunderstood but given the volume of data/NEMO output I found it a missed opportunity that the authors did not test any tuned models with data that was not used for tuning. There are 127 years of ocean conditions and corresponding melt; I would think it would be possible to tune with only a subset and then evaluate performance on the rest. Eq 32 and its explanation suggests this was not done.**

**Maybe the authors could comment on this in the manuscript or, if they feel it is worth doing, carry out additional experiments.**

AR: We agree that this approach would be statistically more robust. For the revision, we will conduct a cross-validation of the parameterisation. This means that we will conduct the tuning and evaluation several times on different periods to robustly estimate the generalisation performance of each parameterisation. To avoid autocorrelation influencing this cross-validation, we will divide the data into 10-year chunks, as the autocorrelation is typically 2 to 3 years in our input temperatures.

RC: *Specific comments*

RC: **Line 28: I don't think it is fair to say this, as the response of ice-sheet models to melt is an enormous source of uncertainty. This is really shown in the initMIP-Antarctica experiments (Seroussi et al, 2019; Fig 4c) where loss of grounded ice over 100 years in different models with melt anomaly treated in the same way across models varies by 400 mm. The papers you cite do not present any results that I can see where the melt treatment was controlled for and inter-model variance in response to melt can really be examined, so I don't think any of these results in these papers really isolate this uncertainty... but initMIP does, so we know it is there.**

AR: We agree that this might have been formulated too strongly. We will reformulate, saying that it is "one of the main sources of uncertainty".

RC: **Line 78: I'm not sure what you mean by "physically sound in time and space". I think you might be saying that by using a model you can perfectly match ocean conditions outside the shelf with melt rates, which you could not do with actual data.**

AR: Yes, this is what we meant. We will reformulate to clarify.

RC: **Line 153: when I saw this, I assumed you were comparing spatial patterns of melt so was confused by eq (32). Maybe be clear for what purpose you interpolate/reproject outputs.**

AR: Thank you for pointing out that this is a source of confusion. The regridding is done for all NEMO variables at the very beginning. All pre-processing and analysis uses the regridded data. This is because it is the preferred format for ice-sheet modellers and makes computing spatial metrics easier. The integrated melt is computed based on the regridded melt patterns. We will reformulate to avoid confusion.

**RC: Figure 1, legend: HIGHGETZ, not WARMGETZ?**

AR: Thank you for highlighting this mistake. We will correct accordingly.

**RC: Line 157: 5 Delta x = 40km? not sure I follow.**

AR: 40 km x 40 km is 1600 km$^2$ and not 2500 km$^2$, thank you for pointing out this mistake. The effective resolution of physical ocean models is typically 5 to 10 times the grid spacing (Bricaud et al., 2020), and this is why we took a slightly higher factor than 5 to multiply with $\Delta$x but forgot to correct this in the manuscript. We will adjust the effective resolution criteria accordingly to correct this mistake.

**RC: Line 203: "a lot" => "widely"**

AR: Will be changed.

**RC: Eq 22 and others: I don't think you say why some terms are bold.**

AR: Thank you for pointing out that this was not made clear in the text. The bold terms highlight the differences in the terms between the plume formulation as given in Lazeroms et al. (2019) and the new formulation of the plume suggested in this manuscript. We will clarify this further in the manuscript.

**RC: Eq 24: for those who are not already very familiar with Lazeroms' method, it might seem strange how you can relate a height difference to length of a plume path without actually integrating the plume equations. Can you give some intuition regarding this definition?**

AR: The plume formulation assumes that the ice shelf base rises from the grounding line up to any point of the ice-shelf base with a constant slope, which brings the formulation to a single dimension along which the plume properties are integrated. This is why height difference and length of the plume path are directly related.

**RC: Line 399: Favier et al 2019 is carried out in the ISOMIP+ domain, correct? Should it be surprising that the parameters are not appropriate? Similarly, does the PIGL situation not assume that all ice shelves are flooded with CDW? Should it be any surprise these give high RMSE?**

AR: The aim of all these basal melt parameterisations is to avoid the use of a computation-heavy cavity-resolving ocean model by representing a simplified version of the ocean physics in the cavity beneath the ice shelf. This means, in principle, that these parameterisations should have the same parameters independently of the cavity geometry (ISOMIP+

or realistic) and of the input temperature and salinity (e.g. warm or cold conditions). If one assumes that the parameters should be retuned for each specific situation, we would argue that the aim of the parameterisation is not reached. We nonetheless agree that this is not genuinely surprising.

Still, we did not expect that our results would diverge so largely from the results from Favier et al. (2019). In regard to the "PIGL" parameter, as we point out in the manuscript, the high RMSE is less of a surprise because the parameter was designed to be used with temperature corrections, which we did not apply. Besides, we would like to point out that the PIGL formulation does not assume that ice shelves are flooded with CDW, it assumes that the melt-temperature quadratic relationship is constrained by the highest melt rates found near Pine Island's grounding line.

**RC:** **Line 425: 5x smaller isn't an order of magnitude**

AR: Thank you for pointing this out. We will reformulate.

**RC:** **Line 425: 3rd column => 2nd column**

AR: We will correct this mistake.

**RC:** **Line 458: just wanted to point out I like this comparison.**

AR: Thank you! :)

**RC:** **Line 471: I would add Reese 2018 to this list.**

AR: Will be added.

**RC:** **Line 503: looks like an error within the brackets about Jacobs 1992. Also this is a really good point to bring up – and there is more recent work done regarding mode-3 melt (Silvano et al, 2016) which would be good to bring up here and in the discussion.**

AR: Thank you for bringing this up, we will look into it and reformulate accordingly.

**RC:** **Line 510: can you elaborate more on your reason for using average over integrated, please. What is the risk of not doing so.**

AR: There is no "risk" per se, it is just a choice to vary our evaluation metrics and evaluate different aspects of the parameterisations. In the first part, we evaluate the integrated melt and this way focus on an ice-shelf-wide metric, which implicitly contains information

about the size of the ice shelf, and its variability with time. By evaluating the average melt (over time and space) near the grounding line, we evaluate if, on average, the right melt rate is occurring near the grounding line, independently of the size of the ice shelf. We will reformulate to explain our choices more clearly.

**RC:** **Lines 544-553: can you explain your experiment more clearly please. I do not understand what you have done here. Is this is new tuning, based on new melt and ocean conditions, or other?**

AR: We apologise that this was not clear enough. We used the tuned parameters but applied them to observational estimates (opposed to simulations in the previous sections) of one ice shelf. So there was no new tuning here. The input temperatures are from Dutrieux et al. (2014), the topography from BedMachine v2 Morlighem (2020); Morlighem et al. (2020). We then compare the resulting melt rates with observational melt rate estimates from Shean et al. (2019) to evaluate the effect of the geometry resolution on the resulting melt rates. We will reformulate to clarify.

**RC:** **Line 560: why would a sigma coordinate model fare better? Sigma coordinate models have singularities and wild errors where the column goes to zero and the surface gradient is high, i.e. near the grounding line.**

AR: We do not argue that a sigma-coordinate model would fare better, we just think that it might lead to different results. We are aware that they do not work well when there are high slopes but they tend to better resolve the ice-ocean interface in regions with non-zero columns and low gradients.

**RC:** **Line 562-566: I think you are being too hard on yourself. Given the aims of the study, im not sure why you would need to consider evolving cavity geometry.**

AR: We thank you for this encouragement. However, we would like to keep this limitation in the manuscript. As you pointed out earlier, the parameters unfortunately seem to depend on the cavity geometry. With further climate change and increased sub-shelf melt and retreat, the ice shelf geometry and the cavity itself are prone to change. Taking this evolution into account during tuning or at least during evaluation would have been interesting for future projections.

**RC:** **Line 567-571. These are really good points. You might add a discussion on why Mode 3 melt is important.**

AR: Thank you for pointing that out. We will reformulate, following your earlier comment.

RC: **4.1.2 and 4.1.3. These are really interesting experiments but I do not understand the initial procedure at all. If I understand correctly, you are attempting to see how your parameter results would vary if you fit with a subsample of your data, or tweak your data somehow (so, perhaps this addresses my #2 general comment?) but I don't understand lines 573-576. What is meant by bootstrapping? What is the nature of each sample – because I read that each sample represents melt of each shelf in each of the 127 years.. so not a subsample. "What is meant by 36 random sub-samples, with replacement"? It is impossible to interpret the rest of the sections without knowing this.**

AR: The procedure is based on the standard statistical method of bootstrapping, which aims to estimate parameter uncertainty. As explained for example in Wilks (2006), bootstrapping relies on resampling different samples with the *same* size. This is why it works "with replacement". In our case, this means that the sample always has the size of 36 ice shelves over 127 years. However, the ice shelves are drawn and then replaced into the selection pool before drawing a new one. This means that, in each sample of 36 ice shelves, different ice shelves are present. For example, in one sample, we could draw Pine Island ice shelf 4 times and not draw Thwaites, Totten, Ross, and Fimbul. In another sample, we could draw twice Totten and twice Ross, but only one time Fimbul and never Pine Island. We will reformulate in the manuscript to further clarify the method for readers who are not familiar with bootstrapping.

RC: **Line 620-622: I do not follow. The way I interpret Fig 9 is that it is essentially impossible to infer the correct parameter "pair" because they so strongly covary, that depending on the specifics of the tuning data you can get e.g a low C and high gamma$_T^\star$, or vice versa, with either fitting the data reasonably well. But in e.g. a future projection with an ice-sheet model, the difference between using one or the other parameter pair could be quite large. So im not sure simply fixing one of these parameters addresses this difficulty.**

AR: The distributions shown in Fig 9 are the results of the bootstrapping method, which aims to give an uncertainty estimate of the parameter(s). The resulting distributions are aimed at scientists who want to sample the uncertainty in basal melt rates introduced by the parameters. Instead of varying both $\gamma_T^\star$ and C, they can now only vary one of them and, using the inferred relationships, simultaneously cover the uncertainty in both parameters. This is one of the main conclusions of this subsection. We will reformulate the goal and conclusion of this subsection to make sure this is clear.

RC: **Lines 700-702. I would think this of CMIP models too. Ill not attempt a list here but there is quite a lot of literature on how global ocean models have difficulty with shelf-offshore exchange.**

AR: Thank you for pointing this out. We will reformulate to mention that some CMIP models struggle with this exchange too.

RC: **Appendix:**

RC: **Line 792: you talk about disagreement in melt with Rignot 2013 here but do not show any images.**

AR: We will include this additional figure showing a comparison between the basal melt rate patterns of our REALISTIC run and the patterns from Rignot et al. (2013).

[Figure]

Figure 1: Mean basal melt rates from (a) Rignot et al. (2013) (2003-2008) and (b) our REALISTIC run (2009-2018) in m ice per year.

RC: **Figure B1: add a legend**

AR: The legend will be clarified.

**References**

Bricaud, C., Le Sommer, J., Madec, G., Calone, C., Deshayes, J., Ethe, C., Chanut, J., and Levy, M.: Multi-grid algorithm for passive tracer transport in the NEMO ocean circulation model: a case study with the NEMO OGCM (version 3.6), Geoscientific Model Development, 13, 5465–5483, #10.5194/tc-2022-3210.5194/gmd-13-5465-2020, 2020.

Dutrieux, P., De Rydt, J., Jenkins, A., Holland, P., Ha, H., Lee, S., Steig, E., Ding,

Q., Abrahamsen, E., and Schröder, M.: Strong Sensitivity of Pine Island Ice-Shelf Melting to Climatic Variability, Science, 343, 174–178, #10.5194/tc-2022-3210.1126/science.1244341, 2014.

Favier, L., Jourdain, N., Jenkins, A., Merino, N., Durand, G., Gagliardini, O., Gillet-Chaulet, F., and Mathiot, P.: Assessment of sub-shelf melting parameterisations using the ocean–ice-sheet coupled model NEMO(v3.6)–Elmer/Ice(v8.3), Geoscientific Model Development, 12, 2255–2283, #10.5194/tc-2022-3210.5194/gmd-12-2255-2019, 2019.

Lazeroms, W., Jenkins, A., Rienstra, S., and van de Wal, R.: An Analytical Derivation of Ice-Shelf Basal Melt Based on the Dynamics of Meltwater Plumes, Journal of Physical Oceanography, 49, 917–939, #10.5194/tc-2022-3210.1175/JPO-D-18-0131.1, 2019.

Morlighem, M.: MEaSUREs BedMachine Antarctica, Version 2., #10.5194/tc-2022-3210.5067/E1QL9HFQ7A8M, boulder, Colorado USA. NASA National Snow and Ice Data Center Distributed Active Archive Center., 2020.

Morlighem, M., Rignot, E., Binder, T., Blankenship, D., Drews, R., Eagles, G., Eisen, O., Ferraccioli, F., Forsberg, R., Fretwell, P., Goel, V., Greenbaum, J., Gudmundsson, H., Guo, J., Helm, V., Hofstede, C., Howat, I., Humbert, A., Jokat, W., Karlsson, N., Lee, W., Matsuoka, K., Millan, R., Mouginot, J., Paden, J., Pattyn, F., Roberts, J., Rosier, S., Ruppel, A., Seroussi, H., Smith, E., Steinhage, D., Sun, B., van den Broeke, M., van Ommen, T., van Wessem, M., and Young, D.: Deep glacial troughs and stabilizing ridges unveiled beneath the margins of the Antarctic ice sheet, Nature Geoscience, 13, 132–137, #10.5194/tc-2022-3210.1038/s41561-019-0510-8, 2020.

Rignot, E., Jacobs, S., Mouginot, J., and Scheuchl, B.: Ice-shelf melting around Antarctica, Science, 341, 266–270, #10.5194/tc-2022-3210.1126/science.1235798, 2013.

Shean, D., Joughin, I., Dutrieux, P., Smith, B., and Berthier, E.: Ice shelf basal melt rates from a high-resolution digital elevation model (DEM) record for Pine Island Glacier, Antarctica, The Cryosphere, 13, 2633–2656, #10.5194/tc-2022-3210.5194/tc-13-2633-2019, 2019.

Wilks, D.: Statistical methods in the atmospheric sciences, 2nd ed., Elsevier, Amsterdam Paris, 2006.

---

## Author Comment (AC2)

**Author Response to Reviewer #2**

**An assessment of basal melt parameterisations for Antarctic ice shelves**

Burgard, C., Jourdain, N.C., Reese, R., Jenkins, A., Mathiot, P.
*The Cryosphere,* `#10.5194/tc-2022-32`

**RC: Reviewer Comment**,     AR: Author Response,     *changed manuscript text*

RC:  **Reviewer Summary:**
**For this first time, this paper performs thorough evaluation of a large number of ice-shelf melt parameterizations in a pan-Antarctic context. This is a topic of significant importance to the community, as melt parameterizations allow for Antarctic ice-sheet modeling on time scales that are not currently possible using ice-sheet models coupled to ocean or Earth system models. A number of aspects of this work are novel. First and most importantly, the evaluation is performed on a large number (36) of Antarctic ice-shelves that are fully representative of the continent and which use realistic topography and climate forcing. Second, the rich NEMO model ensemble provides over 500 years of cumulative simulation results, a far more robust dataset than current observations are able to provide. Third, new variants of melt parameterizations (including various ways of including ice-shelf slope dependence and a new approach to plume parameterization) are introduced for inclusion in the evaluation. Fourth, the paper carefully evaluates each parameterization for continental-scale "biases" (relative to the NEMO model), "biases" for individual ice shelves, melt patterns (with an emphasis on grounding lines) and the behavior of parameterizations at high resolution. Finally, the paper includes invaluable recommendations on which parameterizations show the most promise (in a few different categories) and the caveats and best practices for using each, suggested by the evaluation.**

**The paper is well written and arranged. The figures are quite effective, well crafted, and easy to follow. They give the reader a good, intuitive sense of the relative strengths and weaknesses of the different parameterizations in a variety of ways. The numerous tables will help future users of these parameterizations to know the correct coefficient values to use for each, including recommended values of the tunable parameters that best fit the NEMO simulation results.**

AR:  Thank you very much for your constructive and very detailed comments on how to further improve our paper. We plan to address you comments as described in the following.

RC:  *Major comments*

**RC:** **I do have a few major concerns about the methodology, and addressing these concerns may entail a significant effort by the authors. For that, I am sorry. But I also feel a duty to be particularly careful in evaluating the methodology and suggesting improvements because of the way that the paper is framed: it aims to provide recommended best practices, including preferred coefficients for use in pan-Antarctic modeling using the evaluated parameterizations.**

**AR:** We appreciate your feedback on the methodology. We agree that this paper recommends best practices and needs to be robust. We thank you for raising concerns and plan to address them as follows.

**RC:** **The first concern is one I share with the other reviewer. It would be preferable to tune the parameterizations with one subset of the available data and evaluate it with another, independent subset. Given the more than 500 years of simulation data, I agree that this would seem to be fairly easy to do by using a random subset of years for tuning and the remainder for evaluation. It may be worth adopting this approach in revising the paper.**

**AR:** We agree that splitting the data into a "tuning" and "evaluation" dataset is worth adopting during revision. However, we want to clarify that we do not have 500 simulation years (as suggested by the reviewer), but "only" 127 years. These 127 years are a result of the accumulation of simulation years from four simulations spanning 30 to 40 years (see Table 1 in the manuscript).

For the revision, as suggested by both reviewers, we will conduct a cross-validation of the parameterisation. This means that we will conduct the tuning and evaluation several times on different periods to robustly estimate the generalisation performance of each parameterisation. To avoid autocorrelation influencing this cross-validation, we will divide the data into 10-year chunks, as the autocorrelation is typically 2 to 3 years in our input temperatures.

**RC:** **The second major concern is that the metrics used in the paper for evaluating the root-mean-squared "error" (RMSE) relative to the NEMO results, Eqs. (32) and (33), do not seem robust to me, nor are these definitions the standard ones used in Earth system modeling. This is especially unfortunate because the vast majority of the results shown in the paper (from section 2.4 onward) are dependent on these metrics, including the process for identifying the best-fit parameters, the evaluation of parameterizations on the continental scale (both over each ice shelf and near grounding lines), and the methodology for evaluating parameter uncertainty. I am confident that the adoption of a more standard metric is unlikely to change the broad conclusions of the paper but it is hard to see how it would not**

**change specific results such as the recommended values of coefficients used in the parameterizations. For this reason, I ask the authors to strongly consider redoing their analysis with a more robust and widely accepted version of the RMSE metric.**

AR: We deliberately chose non standard metrics instead of the standard melt rate RMSE, but your bad appreciation of our metrics shows that we need to improve the justification for these metrics and provide evidence for their robustness. In short, we want to tune and evaluate the parameterisation in a way that gives the most interesting results for an application in either an ice sheet or an ocean model (or possibly at the interface between both). In the following, we address your points and comments and hope that the more detailed explanation of our choices will convince you.

RC: **In my own work and I think in climate modeling broadly, we always use the area-weighted variance to compute the RMSE as follows:**

$$RMSE = \sqrt{\frac{\sum\limits_{i=1}^{N}(m_i - m_{o,i})^2 A_i}{\sum\limits_{i=1}^{N} A_i}} \tag{1}$$

**where, in the context of this paper, $m_i$ is the melt rate (in whatever units you choose) from the parameterization for sample $i$ (which might be one location and year); $m_{o,i}$ is the "observed" melt rate from NEMO (at the same location and year); $A_i$ is the area associated with the $i$th sample (this could be a whole ice shelf or an individual NEMO grid cell, depending on how the evaluation is performed); and N is the number of sample points. This metric is robust to different ways of sampling the evaluation points. It works just as well to evaluate the "error" over a single ice shelf as for any number of ice shelves aggregated together into a region. The RMSE values for individual ice shelves can be computed by sampling the pattern at locations across the ice shelf, and then these RMSE values can be combined to get regional- and continental-scale values of the RMSE. This approach is also robust in the sense that including or excluding any number of small ice shelves would have a small impact on the RMSE.**

**Unfortunately, the same is not true for either of the RMSE definitions, Eqs. (32) and (33), proposed in the paper. Instead of weighting by area, these metrics treat each ice shelf (independent of its area) as having equal weight. This approach has at least three problems. First, the metrics are significantly more sensitive to the number of ice shelves under consideration. Considering Eq. (32) first, repeated here:**

$$RMSE_{\text{int}} = \sqrt{\frac{\sum\limits_{i}^{N_{\text{isf}}} \sum\limits_{t}^{N_{\text{years}}} (M_{\text{param}}[i,t] - M_{\text{ref}}[i,t])^2}{N_{\text{isf}} N_{\text{years}}}} \tag{2}$$

**if we were to add several ice shelves with both negligible bias in melt flux and negligible area, the RMSE$_{\text{int}}$ would decrease (because we aren't adding significantly to the numerator but we are adding new ice shelves to the denominator). Now, considering Eq. (33):**

$$RMSE_{\text{GL}} = \sqrt{\frac{\sum\limits_{i}^{N_{\text{isf}}} \sum\limits_{n}^{N_{\text{simu}}} (m_{\text{GL,param}}[i,n] - m_{\text{GL, ref}}[i,n])^2}{N_{\text{isf}} N_{\text{simu}}}} \tag{3}$$

**In this case, if we were to add several tiny ice shelves with negligible area but significant biases in melt rate, they could significantly increase the RMSE$_{GL}$. In this way, these metrics are likely fairly sensitive to the number of ice shelves (36) chosen for the study.**

AR: We plan to address the two points raised as follows:

*Tuning and evaluating at the grid cell level*
We are aware that the most common way to compute the RMSE in climate modeling is to use the area-weighted variance taking into account each grid cell (as you showed in Eq.(1)). However, very early in our study, we decided that tuning and evaluating at the grid-cell level would not yield the best compromise for the tuning result and for the conclusion drawn from the evaluation. There are several reasons that motivated this choice:

1. The standard RMSE would be highly biased towards Filchner-Ronne and Ross ice shelves, which cover a much larger area (many more grid cells) than the others. As a comparison here are the areas of Ross and Filchner-Ronne compared to the third largest ice shelf Amery or to an important ice shelf like Pine Island: Ross = $4.7 \times 10^5$ km$^2$, Filchner-Ronne = $4.2 \times 10^5$ km$^2$, Amery = $5.9 \times 10^4$ km$^2$, Pine Island = $5.7 \times 10^3$ km$^2$. We do not think that tuning on the grid cell level is the most interesting way to go, because Filchner-Ronne and Ross are not necessarily the ice shelves that (1) affect most near-future ice dynamics (Seroussi et al., 2020) or (2) contribute the most to the present-day meltwater release into the ocean (Rignot et al., 2013).

2. The standard RMSE would give the same importance to all grid cells although we know that there are regions that are more or less important for buttressing and therefore for the influence of melt on the ice-sheet evolution (Reese et al., 2018). Many grid cells of Ross and Filchner-Ronne as well as other smaller ice shelves are "passive" and can therefore suffer from acceptable biases that will not significantly affect the ice dynamics. In terms of impacts on the ocean circulation, we believe that getting a correct freshwater budget around Antarctica (i.e. cavity integrated melt) is a priority before getting the correct depth distribution of the freshwater release at a given location.

3. We consider that the melt parameterisations we tune and evaluate are too simple to reproduce all the details of the spatial melt patterns. If they can reproduce the main pattern (e.g. more melt near the grounding line) but the pattern is shifted a little in space, this will result in a high RMSE, penalising a parameterisation that could reproduce some of the complexities of the melt patterns but not at the exact correct location.

4. We only consider relatively large ice shelves (those that are well resolved by NEMO) so there is no issue with the number of very small ice shelves that matter for neither ice nor ocean dynamics. This is an important aspect of our metrics that we will emphasise more in the revised manuscript.

In conclusion, we would like to stay with our decision of tuning toward an integrated metric, but we will improve the part of our manuscript explaining our choice.

*Weighting the RMSE for tuning*
Our RMSE based on integrated values gives the same weight for two ice shelves with the same integrated melt irrespectively of their size, buttressing effect on ice dynamics, or effect on ocean convection. This is an empirical choice, and we propose to add a discussion about this choice and the associated robustness of our model selection. Weighting the RMSE with the ice-shelf area results in the same problem as raised for the tuning on the grid-cell level: the tuning will be particularly biased towards Filchner-Ronne and Ross ice shelves, which cover a much larger area than the others, although they are not necessarily the most "interesting" ones for the reasons mentioned above. Instead, the integrated melt implicitly contains the importance of the area for the melt and therefore influences the tuning when it is relevant for the integrated melt, for example if a high area is the main driver behind a high integrated melt.

**RC:** **Second, the metric in Eq. (2) here gives significantly more weight to ice shelves with large areas (since $M_i = m_i A_i$, the numerator in this equation is getting weighted by $A_i^2$ ), while the metric in Eq. (3) here gives significantly less weight to large ice shelves (because $A_i$ does not appear in the numerator at all).**

AR: To assess the parameterisations, we chose to evaluate different aspects of the resulting melt. With the integrated melt, we evaluated the ability of the parameterisations to reproduce the time variability and the integrated value of the melt for a given cavity. With the mean melt rate near the grounding line, we aimed to assess if the parameterisation reproduced the mean pattern well, i.e. if the mean melt in the grid cells near the grounding line was on the order of the reference. By averaging the pattern over time and space, we aimed to avoid that this part of the evaluation was biased due to mismatch in time variability (already evaluated before) and slightly shifted patterns in the vicinity of the grounding line. We will explain the procedure and choices more thoroughly in the revised manuscript to clarify.

**RC:** **Third, the definition of an ice shelf is not particularly robust. The Filchner and Ronne Ice Shelves are sometimes considered distinct (as in this paper) and sometimes are combined together. Similarly, the Eastern and Western Ross are sometimes considered separately (though Ross is a single ice shelf here). As grounding lines move, ice shelves that were once distinct may merge, or a single ice shelf may be divided into two. The RMSE metrics from the paper are sensitive to the authors' definition of the 36 ice shelves used in the paper, whereas the metric in Eq. (1) here would not change if an ice shelf were divided into 2.**

**AR:** We agree that the choice of division of ice shelves is somewhat subjective. From an oceanographic point of view, we agree that combining Filchner and Ronne as one ice shelf makes more sense. This is what we will use in the new procedure.

**RC:** **In summary, I do not think robust statistics can be performed when treating ice shelves as distinct entities given equal weight regardless of size. I do not see how a meaningful RMSE in the integrated melt flux can be computed between ice shelves: it would only be meaningful if totalled over all ice shelves in a region (but averaged over samples in time and across ensembles).**

**AR:** We hope that the responses to the previous comments have addressed this issue of robust statistics. We would like to emphasize that our metrics were not defined for a universal use, but are rather carefully designed for the specific purpose and datasets of our study. We nonetheless think that our results are robust in the sense that our recommendations are not overly sensitive to our empirical choices.

**RC:** **My preference would be that the results of the paper be re-evaluated with the metric in Eq. (1) here instead of Eqs. (32) and (33) in the paper. I realize this is a monumental task. If the authors do not feel that this is necessary, I am willing to hear you out. But I would need to see a strong case for why the existing metrics are more robust and defensible than I currently believe.**
**I want to reiterate that I think this is amazing work! If one takes the metrics for evaluating "biases" as given, the rest of the results follow nicely: the evaluation of each parameterization is remarkably thorough and well executed. The service that this work does for the community is truly commendable, particularly because it gives future model developers not only the tools to make an informed choice among the existing parameterizations but also a good choice of coefficients to use.**

**AR:** We thank the reviewer for the detailed description of their concerns. After consideration, we sincerely believe that our initial choice is more appropriate for the aforementioned reasons. As a consequence, we will re-run the tuning and evaluation, taking into account that we now combine Filchner-Ronne as one ice shelf and that we will conduct several

iterations of the tuning and evaluation on different chunks of time, but we will keep the initial metrics.

**RC:** **Below, I have a number of specific comments and suggestions for typographical or grammatical changes. Please do not be intimidated by the number of comments. I have been told by editors in the past that I seem to review papers as if I were a coauthor, which may be true.**

AR: We thank you for the thoroughness of the comments. We deeply appreciate the time you have put into this review.

**RC:** *Specific comments*

**RC:** **l. 7-8: "The box, PICOP parameterization and quadratic parameterizations with slope dependency..." In the context of the abstract (in which these different parameterizations have not yet been introduced), I found it hard to understand what was being described here. The previous sentence does a nice job of introducing what the quadratic parameterization means and what the slope dependence is about. A similar introduction for the box parameterization and PICOP here would be helpful.**

AR: We understand the concern and will clarify this part of the abstract.

**RC:** **l. 19-20: "...interacting with land or shallow rocks...": I'm not sure I follow what you mean by interacting with shallow rocks. Maybe take this out?**

AR: We will take this out.

**RC:** **l. 38: "...and do not resolve ice-shelf cavities...": Should this be "include" instead of "resolve"? It seems like you are referring to model configurations that don't have the cavities at all, rather than that they are there but at too coarse a resolution.**

AR: Thank you for pointing this out. We agree and will reformulate as suggested.

**RC:** **l. 66-69: When you refer to a "step" here, does each of these correspond to a section of the paper? If so, it might be clearer to replace "step" with "section".**

AR: The first step is one section but the second and third step refer to subsections of the results. We will change to "section" and "subsection".

**RC:** **l. 115 "The top boundary-layer thickness is set to 30 m." This likely needs a little more explanation: this is referring to the boundary-layer thickness over which temperature and salinity are averaged for use in the parameterization of sub-ice-shelf turbulent fluxes.**

**AR:** Thank you for pointing this out. We will reformulate to clarify that this is the boundary-layer thickness over which temperature, salinity, and ocean velocities $u$ and $v$ are averaged (and we corrected to 20 m, as 30 m was an error).

**RC:** **l. 147: "basal mass loss" might be better as "basal mass flux" or something like that because usually "mass loss" is used in our field to refer to the uncompensated loss of mass from the ice sheet that contributes to sea-level rise.**

**AR:** Thank you for pointing this out. We will use "integrated melt rate" to stay consistent with the rest of the manuscript.

**RC:** **l. 188-191: Here, a bunch of physical constants and variables are introduced for the second time (they were introduced on l. 104-115). I think the only new ones here are $\Gamma_{TS}$, $T_i$ and $U$. Also, it seemed to me that $U$ might be the same as $u_{oc}$ introduced above. If so, maybe use the same symbol for both. If not, maybe clarify the distinction.**

**AR:** Thank you for pointing out that this was confusing. We re-introduced them because the first introduction of the variables happened well earlier. We will clarify the definitions and avoid defining the same variable twice.

**RC:** **l. 213: "where the subscript oc denotes the far-field temperature and salinity that is advected into the cavity". This seems like a different definition than on l. 105: "$T_{oc}$ and $S_{oc}$ are the temperature and salinity averaged over a boundary layer below the ice shelf." It would be helpful if the oc subscript meant the same thing throughout the manuscript. Maybe another subscript like ff could be used for far-field quantities (or bl for boundary-layer quantities)?**

**AR:** Here also, thank you for this suggestion. We will re-read through the manuscript to make the definitions more homogeneous throughout the manuscript.

**RC:** **l. 217: The liquidus slope has already been defined on l. 108.**

**AR:** We defined it more losely in L108 but will make it more precise to avoid having to re-define it in L217.

**RC:** **l. 240-240: "However, through the resulting circulation, the temperature at one**

**point of the ice shelf cavity is not necessarily independent from the temperature at the other points of the ice shelf cavity." This may be correct, but I understood the quadratic dependency a little differently. I thought the concept was that the overturning itself was driven by the aggregate effect of the thermal forcing (so that the strength of the overturning circulation would be proportional to the mean thermal forcing in the cavity). Thus, the local velocity is represented in terms of the amount of heat available for melting and there isn't necessarily any implication that the temperatures at different parts of the ice shelf are correlated to one another.**

AR:   We agree that this was not clearly formulated. We will reformulate as suggested.

**RC:   Eq. (17): It seems like some explanation of the bracket < > notation is needed.**

AR:   We thought this was a notation commonly used. We will add an explanation.

**RC:   Eqs. (20) and (21): I think these should be moved after Eq. (24) (with Eq. (20) after Eq. (21)) because they don't currently have any context where they are.**

AR:   We will move them as suggested.

**RC:   Eqs. (23) and (25) are identical, right? If so, I don't think it's necessary to have Eq. (25), you can just state that $M_2$ is still given by Eq. (23).**

AR:   They are identical but the characteristic length scale $x$ used in both cases is different. We will re-arrange to avoid repeating Eq.(23) twice.

**RC:   l 311-312: "If one of the boxes has an area of zero...": It isn't clear to me how this could happen. Is this just for an ice shelf that is poorly resolved so that there aren't enough grid cells for each box to get some?**

AR:   Yes, this is how it can happen that a box has an area of zero. We will add a short explanation.

**RC:   l. 313: "are non-zero": Should this be "have a non-zero area?"**

AR:   Yes, we will reformulate as suggested.

**RC:   l. 349-350: "We use yearly mean profiles as the residence time of water in ice shelf cavities might be longer than a month for some cavities." You explain this more in the discussion but it might be worth giving it more context here, too. The**

**parameterizations do not include the finite advection time from the ice-shelf front to the ice-shelf base, meaning they are missing important delays that are present in the physical system. This means there is little hope of capturing seasonal melting in the proper phase.**

AR: We will add a sentence to briefly mention this issue here as well.

RC: **l. 431: "We therefore only show results for heterogeneous boxes hereafter." This is not true in the next subsection when you talk about PICOP, so maybe change "hereafter" to "in the remainder of this subsection."**

AR: Thank you for pointing this out. We will reformulate as suggested.

RC: **l. 458-460: "To put these RMSE in context, the integrated ice-shelf melt averaged over all ice shelves and simulation years in the reference is 38 Gt/yr, which means that even for the better-performing parameterisations, the RMSE remains very high compared to the reference melt as such."**
**I have concerns about this concept, related to my concerns about the metrics in Eqs. (22) and (23). I do not think it is meaningful that the total Antarctic melt flux ( 1500 Gt/yr) averaged over 36 ice shelves (an arbitrary number related to the resolution of the NEMO simulations used for reference) comes out to  38 Gt/yr. The different ice shelves have such different areas and melt rates that this average isn't informative. Given that I have problems with the metric from Eq. (22) that you're given context to, this might be a moot point but if you want to give context to what 40 Gt/yr is like, maybe just point out an ice shelf by name that has about that much melting.**

AR: Thank you for expressing your concern, we will think about a new way to put the RMSE in context.

RC: **l. 504-505: "...likely because they are all forced by yearly ocean temperatures." This seems to assume that the parameterizations would be able to get the Mode 3 melt if they were forced with seasonal or monthly temperatures. As you state later, they are unlikely to have the correct processes or the correct seasonal phasing to get these melt rates right even if temperature forcing were provided on a shorter time interval.**

AR: We do not completely agree. If we used monthly temperature and salinity profiles as input, it would be possible to represent mode 3 melt. Due to the lag between inflow time and melt, the response in the melt would probably occur too early compared to the reference but the mechanisms might be grasped, and as mode 3 melt mostly occurs near the ice shelf front, the advection times are shorter than for grounding line melt. We will

clarify this in one sentence.

**RC: Figs. 4 and 7: I appreciated this visual representation of the RMSE – it was very intuitive.**

AR: Thank you!

**RC: Figs. 5 and 6: These are wonderfully done and very clear!**

AR: Thank you!

**RC: l. 575-576: "...and is composed of 36 random samples, with replacement, of the different ice shelves." Do I understand correctly that you double or triple (or whatever) count some ice shelves and omit other ice shelves in this process? That seems like an uncommon practice and that the approach (suggested by reviewer 1) of using only some of the years for training and holding back others for validation would be a more standard method for testing the robustness and parameter uncertainty.**

AR: As explained for example in Wilks (2006), bootstrapping relies on resampling different samples with the *same* size. This is why it works "with replacement". Bootstrapping gives an estimate of the parameter uncertainty. The cross-validation, which is the procedure used in the iterations of tuning (holding back part of the sample to evaluate on this part), provides an estimate of the generalisation performance of the parameterisation. These are two different conclusions from a statistical point of view.
We agree that the method presented here only shows the uncertainty introduced by the choice of included ice shelves. We will include the time-chunk dimension in the resampling to cover the full uncertainty of the parameters.
We will explain the procedure and methods more thoroughly.

**RC: Fig. 9a. On the light gray curve it says "fit on pairs with E0 < 50" but in the text describing Eq. (34), I didn't see anything stating that it was for E0 < 50. Is (34) not for the gray curve? Is this description simply missing from the text?**

AR: Yes, this is missing, thank you for pointing it out. We will add it.

**RC: l. 617-619: "Note that some of the higher values of $C_d^{1/2}\Gamma_{TS}$ and $E_0$ are several orders of magnitude higher than expected (see e.g. Table 12), which we cannot explain." This seems odd. Don't you have constraints on the parameters to prevent them from varying outside a physically likely range?**

AR: We agree that this is odd and is not satisfying. Introducing lower constraints led to parameters "stuck" at the constraint, showing that the lowest RMSE was reached with

unrealistically high parameter values. If after the new tuning and evaluation procedure this problem still exists, we see no other option than acknowledging it.

**RC: l. 648: "The slope-dependent parameterisation is nonetheless relatively good for cavity melt rates..." Do you mean "...good for capturing melt patterns" or something like that?**

AR: Yes, this is what we meant. We will reformulate to clarify.

**RC: l. 744-745: "...with RMSEs on the same order as or even larger than the reference value." This is another reference to ice-shelf-by-ice-shelf statistics that I think needs to be rethought in terms of area-weighted statistics.**

AR: We will think about an alternative formulation.

**RC: Appendix D: Could you say how the RMSE for the heatmaps are defined? Are these the same as Eq. (32) but with Nisf = 1?**

AR: Yes, thank you for pointing out that this is not clear. We will add a little explanation. It is the same as Eq (32) for Fig. D1 and Eq (33) for Fig. D2 but with $N_{isf}=1$.

**RC: *Typographical and Grammatical Suggestions***

**RC: l. 11: "Additionally to..." should be "In addition to..."**

AR: Thanks, will be changed.

**RC: l. 28: "uncertainty source" should be "source of uncertainty"**

AR: Thanks, will be changed.

**RC: l. 140: Probably remove "For example", since this sentence didn't clearly follow (at least for me) as an example of the changed parameters not having a significant impact on the physical ocean.**

AR: Maybe this was not clear enough then. We mean that the changed parameters don't have a significant impact on the physical link between the ocean in front of the ice shelf and the basal melt rates. They have, however, an impact on the physical ocean *outside* of the cavity. So this "for example" actually should follow the sentence before. We will reformulate to clarify.

**RC:** **l. 204: "from an ice shelf to another" should be "from one ice shelf to another".**

AR: Thanks, will be changed.

**RC:** **l. 252: "(based on Sec. 2.2.2)" might be more natural as "(as described in the next section)".**

AR: Thanks, will be changed.

**RC:** **l. 288-291: This is a lot in one sentence and gets a bit confusing. It might be clearer if you break the details of each subscript into its own sentence.**

AR: Thank you for pointing it out. We will break this into several sentences as suggested.

**RC:** **l. 308, 314, 315, 437, 444, 446, 575, 589, 666, 667, 747, Appendix C, Table C1: The word "amount" is used in several places where "number" is correct (because the object being referred to is countable, rather than indefinite): "number of boxes", "number of data points", "number of high-resolution ocean simulations", etc.**

AR: Thank you for this very helpful comment. We will correct this mistake.

**RC:** **l. 381-382: "two constraints are additionally needed" would be better as "two additional constraints are needed"**

AR: Thanks, will be changed.

**RC:** **l. 399, 434: "(2020)(Table 4, 1st and 3rd columns)" and similar: This is very picky on my part but if there's a way to avoid back-to-back parentheses like here, I would prefer it.**

AR: Thanks, we will try to change.

**RC:** **l. 404-405: "The new tuning only slightly achieves to reduce the RMSE further" would be better as "The new tuning achieves only a slight ruther reduction in RMSE"**

AR: Thanks, will be changed.

**RC:** **l. 414-415 and 449: "input properties are of a similar order of magnitude" and "all of a similar order of magnitude"**

AR: Thanks, will be changed.

**RC:**  **l. 438: "...about twice as high as the..."**

 AR:  Thanks, will be changed.

**RC:**  **l. 454: "Slightly above..." should be "At slightly above..."**

 AR:  Thanks, will be changed.

**RC:**  **l. 460, 517, 565: "... to the reference melt as such." and "...the melt patterns as such". The phrase "as such" is used in a few places in the paper where I don't fully understand it. Maybe you mean "itself"? Maybe this can just be omitted?**

 AR:  We will identify the occurrences and look for a better formulation.

**RC:**  **l. 506: "additionally" should be "in addition"**

 AR:  Thanks, will be changed.

**RC:**  **l. 519: "This can be explained by punctual very strong melt". I do not know for sure what is meant here by "punctual". This usually means "on time".**

 AR:  Thank you for pointing this out, this was a wrong translation from our French mind then. We meant "in a very small region" (like a point). Will be changed.

**RC:**  **l. 521: "should therefore be kept" would be a bit better as "is important to keep" and "On the opposite" should be "In contrast".**

 AR:  Thanks, will be changed.

**RC:**  **l. 527: "...in a more consistent way and with larger...**

 AR:  Thanks, will be changed.

**RC:**  **l. 543: "emulates well NEMO" should be "emulates NEMO well"**

 AR:  Thanks, will be changed.

**RC:**  **l. 549-550: "...and therefore rather is a result of the circum-Antarctic tuning rather than of the resolution"**

**RC:** **l. 550 "...on the contrary" should be "in contrast"**

AR: Thanks, will be changed.

**RC:** **l. 568: "...require to account for..." should be "...require accounting for..."**

AR: Thanks, will be changed.

**RC:** **l. 616: "This relation is different from Eq. (34)..."**

AR: Thanks, will be changed.

**RC:** **l. 627: "On the opposite" should probably be "In contrast"**

AR: Thanks, will be changed.

**RC:** **l. 637: "Especially" should be something like "In particular"**

AR: Thanks, will be changed.

**RC:** **l. 648: "...used as end members" might be "...used to generate end members..."**

AR: Thanks, will be changed.

**RC:** **l. 660-661: "We therefore suggest, when possible, trying out..."**

AR: Thanks, will be changed.

**RC:** **l. 694: "...any improvement in either of the two..."**

AR: Thanks, will be changed.

**RC:** **l. 716: "...take into account, to some extent, the horizontal..."**

AR: Thanks, will be changed.

**RC:** **l. 719: missing a comma after "geometry"**

AR: Thanks, will be changed.

**References**

Reese, R., Gudmundsson, G., Levermann, A., and Winkelmann, R.: The far reach of ice-shelf thinning in Antarctica, Nature Climate Change, 8, 53–57, #10.5194/tc-2022-3210.1038/s41558-017-0020-x, 2018.

Rignot, E., Jacobs, S., Mouginot, J., and Scheuchl, B.: Ice-shelf melting around Antarctica, Science, 341, 266–270, #10.5194/tc-2022-3210.1126/science.1235798, 2013.

Seroussi, H., Nowicki, S., Payne, A., Goelzer, H., Lipscomb, W., Abe-Ouchi, A., Agosta, C., Albrecht, T., Asay-Davis, X., Barthel, A., Calov, R., Cullather, R., Dumas, C., Galton-Fenzi, B., Gladstone, R., Golledge, N., Gregory, J., Greve, R., Hattermann, T., Hoffman, M., Humbert, A., Huybrechts, P., Jourdain, N., Kleiner, T., Larour, E., Leguy, G., Lowry, D., Little, C., Morlighem, M., Pattyn, F., Pelle, T., Price, S., Quiquet, A., Reese, R., Schlegel, N.-J., Shepherd, A., Simon, E., Smith, R., Straneo, F., Sun, S., Trusel, L., van Breedam, J., van de Wal, R., Winkelmann, R., Zhao, C., Zhang, T., and Zwinger, T.: ISMIP6 Antarctica: a multi-model ensemble of the Antarctic ice sheet evolution over the 21st century, The Cryosphere, 14, 3033–3070, #10.5194/tc-2022-3210.5194/tc-14-3033-2020, 2020.

Wilks, D.: Statistical methods in the atmospheric sciences, 2nd ed., Elsevier, Amsterdam Paris, 2006.

---

## Author Response (AR1)

**Author Response to Reviewer #1**

**An assessment of basal melt parameterisations for Antarctic ice shelves**

Burgard, C., Jourdain, N.C., Reese, R., Jenkins, A., Mathiot, P.
*The Cryosphere,* `#10.5194/tc-2022-32`
* * *
**RC: Reviewer Comment**,  AR: Author Response,  *changed manuscript text*

RC: **Reviewer Summary:**
**The manuscript by Burgard et al carries out a comprehensive analysis combining (I think) all of the leading parameterisations of ice-shelf melt that are currently used, with a set of global ocean model simulations, in which the parameterisations are rigorously analysed in terms of their ability to replicate ocean modelled melt rates given open-ocean properties, their stability in terms of optimal parameters, and their benefits and drawbacks in terms of use. I think this is a great study to have been carried out, as to date no other studies have collected all of the extant parameterisations together in one study, implemented them in a common framework, and tuned and evaluated them with identical data. The study also does not ignore the importance of spatial patterns of melt arising from the parameterisation, which are too often overlooked. Though it is a very long paper, it is formulaic and there is a progression in terms of the experimental setup and analysis, making it a less daunting read. The length is also owed to its comprehensive discussion of existing parameterisations, and any modifications made to them as part of this study, and it is really good to have all of this material together in one place. I think this is a worthwhile and interesting study, as it will be important to determine how the Antarctic ice sheet will evolve in response to oceanic change, and it is clear that ocean models which can resolve under-ice shelf circulation are the rate-limiting step in such investigations. Therefore the lessons learned from this study are valuable and I recommend for publication after some minor revision. I have comments below that I hope might help in this regard.**

AR: Thank you very much for the positive feedback and for your constructive comments on how to further improve our manuscript. Here is an overview on the main modifications to the manuscript:

- We made the evaluation of the parameterisations more statistically robust by using cross-validations over time and space and worked on clarifying the goal and procedure around our choice of RMSE, the cross-validations and bootstrapping by rewriting Sec. 2.4. We also added a discussion about the choice of statistical metrics in Sec. 4.1.2.

- We reconducted all the analysis following your comments and the comments of reviewer #2, i.e. combining Filchner and Ronne into one ice shelf and conduct cross-validations (tuning on one fraction of the sample and evaluation on the other fraction). We also corrected small errors in the code for the tuning of the simple parameterisations and the PICOP parameterisation and in the formulation of the plume parameterisation. Combining Filchner-Ronne and these slight changes in the code lead to different best-estimate parameters compared to the previous version of the manuscript but do not strongly affect the conclusions.

- We reworked on Appendix D, to provide some context for our results on the ice-shelf level, adding a figure showing the reference integrated melt and melt near the grounding line for the individual ice shelves and changing the heatmaps to show the average difference between parameterised and reference melt to show if there is a clear tendency towards under- or overestimation for individual ice shelves.

- We added an Appendix E, which presents a comparison of an RMSE of the melt near the grounding line evaluated on the grid-cell level and an RMSE of the melt near the grounding line evaluated on time- and space-averages for each ice shelf.

Regarding your specific comments, we address them as described in the following.

**RC: Side note: (On a side note, the python library developed for this study will be of value as well, although its value may depend on how easily it can be implemented with C++ and Fortran ice-sheet models. However this is not directly relevant to the merits of the manuscript.)**

AR: We agree that our package is not designed for direct implementation into an ice-sheet model code-wise. Nonetheless, it could be used as an offline python interface between a coarse ocean model and an ice-sheet model (both of them possibly written in Fortran or C++).

**RC: *I have only two general comments, and it is with regard to the global simulations used to force and tune the parameterisations and the assessment of parameterisation skill:***

**RC: With a resolution for the ensemble of 8 km at 70 S, this is quite coarse for a simulation that is meant to provide "truth" for ice-shelf melt in response to conditions on the shelf and the ACC. It is true the conditions on-shelf are also not necessarily realistic, but it is the continental-shelf-to-melt dependence that is important here. For instance 8 km is well above the deformation scale, so I question the model's ability to represent boundary currents that bring warm water into cavities and melt-laden water out, and transport around bathymetric obstacles and through bathymetric**

**depressions, and these could potentially impact total melt, rather than just melt patterns. I think this potential caveat, as well as those mentioned in the discussion, should be more clearly stated up front in section 2.1.**

AR: We agree that a resolution of 8 km is still not perfect to reproduce the ocean circulation on the continental shelf. Nevertheless, it is an improvement compared to the data available typically used to force ISMIP6-type ice-sheet models, which comes from climate and ocean models that represent the topographic dynamical features even more crudely (typically at a resolution of 1°). Our "REALISTIC" simulation is indeed able to capture strong temperature gradients across the shelf break in East Antarctica as well as in the Ross and Weddell Sea, while capturing the penetration of CDW in the Amundsen sector (compare offshore to onshore in Fig.2). Furthermore, we focus on the link between the domain in front of the ice shelf and the melt inside the cavity. Therefore, very simply said, if the "wrong" water is in front of the cavity compared to reality, it is expected to lead to a "wrong" melt compared to reality but in a physically consistent way.

In regard to the ocean circulation inside the cavity, we agree that we do not resolve all bathymetric ridges, basal channels and eddies that potentially affect the melt. Also, the resolution is not as high as ice-sheet models directly next to the grounding line. We added these points to the list of caveats introduced by NEMO (listed in Sec. 4.1.1). We have left the list of uncertainties introduced by NEMO in the discussion section because the evaluation of the influence of the resolution on the parameterised melt requires the best-estimate parameters that are presented in the results section. However we point to this discussion in the beginning of Sec. 2 so that the reader knows where to find a discussion of the limitations. The manuscript now reads:

*Note that the perfect model approach relies on the assumption that NEMO results in a realistic approximation of the circulation in the ice-shelf cavity and melt behaviour at the ice-ocean interface. We discuss the limitations of this assumption in Sec. 4.1.1.*

and we have added the following in Sec. 4.1.1:

*Still, some uncertainty remains because not resolving small-scale geometric features, such as thin bathymetric ridges and basal channels, and eddies may also affect the ocean circulation in the whole cavity, which could affect both local and integrated melt.*

RC: **I may have misunderstood but given the volume of data/NEMO output I found it a missed opportunity that the authors did not test any tuned models with data that was not used for tuning. There are 127 years of ocean conditions and corresponding melt; I would think it would be possible to tune with only a subset and then evaluate performance on the rest. Eq 32 and its explanation suggests this was not done. Maybe the authors could comment on this in the manuscript or, if they feel it is worth doing, carry out additional experiments.**

AR: We agree that this approach would be statistically more robust. For the revision, we

conducted a cross-validation of the parameterisation. This means that we conducted the tuning and evaluation several times on different periods and ice shelves to robustly estimate the generalisation performance of each parameterisation. To avoid autocorrelation influencing this cross-validation, we divided the data into 10-year chunks, as the autocorrelation is typically 2 to 3 years in our input temperatures. We explain the procedure in detail in Sec. 2.4.1 of the revised manuscript:

*To evaluate the performance of the different parameterisations and the robustness of their tuning, we conduct two variations of leave-one-block-out cross-validation on the minimisation of $RMSE_{int}$, one on the ice shelf dimension and one on the time dimension. This approach consists of dividing the dataset into $N$ blocks, tuning the parameterisation by minimising the evaluation metric on $N - 1$ blocks and applying the tuned parameter(s) on the left-out block (Wilks, 2006; Roberts et al., 2017). The procedure is re-iterated $N$ times, leaving out each of the $N$ blocks successively, so that, in the end, each $N$th block has been left out once. All left-out blocks, using the separately tuned parameters, can then be concatenated to form a "synthetically independent" evaluation dataset. Applying the evaluation metric on this evaluation dataset, we can assess how well the parameterisation generalises to blocks not "seen" during tuning. We apply the cross-validations to each input domain (i.e. the 50 km and the offshore domain) separately, using $RMSE_{int}$ as an evaluation metric. On the ice-shelf dimension, we use $N$=35 for the cross-validation over ice shelves. On the time dimension, we divide the years into blocks of approximately 10 years (ten 10-year blocks and three 9-year blocks) to reduce the effect of autocorrelation, which is typically 2 to 3 years in our input temperatures. This results in $N$=13 for the cross-validation over time.*

**RC:** *Specific comments*

**RC:** **Line 28: I don't think it is fair to say this, as the response of ice-sheet models to melt is an enormous source of uncertainty. This is really shown in the initMIP-Antarctica experiments (Seroussi et al, 2019; Fig 4c) where loss of grounded ice over 100 years in different models with melt anomaly treated in the same way across models varies by 400 mm. The papers you cite do not present any results that I can see where the melt treatment was controlled for and inter-model variance in response to melt can really be examined, so I don't think any of these results in these papers really isolate this uncertainty... but initMIP does, so we know it is there.**

AR: We agree that this might have been formulated too strongly. We reformulated as follows:
*Still, it is currently one of the main sources of uncertainty in such projections.*

**RC:** **Line 78: I'm not sure what you mean by "physically sound in time and space". I think you might be saying that by using a model you can perfectly match ocean conditions outside the shelf with melt rates, which you could not do with actual data.**

AR: Yes, this is what we meant. We reformulated as follows:
*... and (2) using a model provides a self-consistent framework where the ocean properties in front of the ice shelf, which we feed into the different parameterisations, perfectly match the melt rates at the base of the ice shelf, a perfect link which is currently not achievable with observational estimates, except at a few specific locations.*

**RC:** **Line 153: when I saw this, I assumed you were comparing spatial patterns of melt so was confused by eq (32). Maybe be clear for what purpose you interpolate/reproject outputs.**

AR: Thank you for pointing out that this is a source of confusion. The regridding is done for all NEMO variables at the very beginning. All pre-processing and analysis uses the regridded data. This is because it is the preferred format for ice-sheet modellers and makes computing spatial metrics easier. The integrated melt is computed based on the regridded melt patterns. We reformulated as follows:
*For this study, we start by interpolating the NEMO output bilinearly to a stereographic grid of 5 km spacing as all parameterisations are coded for stereographic grids, which are commonly used for ice sheet models. All pre-processing and analysis is conducted using this regridded data.*

**RC:** **Figure 1, legend: HIGHGETZ, not WARMGETZ?**

AR: Thank you for highlighting this mistake. We corrected accordingly.

**RC:** **Line 157: 5 Delta x = 40km? not sure I follow.**

AR: 40 km x 40 km is 1600 km$^2$ and not 2500 km$^2$, thank you for pointing out this mistake. The effective resolution of physical ocean models is typically 5 to 10 times the grid spacing (Bricaud et al., 2020), and this is why we took a slightly higher factor than 5 to multiply with $\Delta$x but forgot to correct this in the manuscript. We adjusted the effective resolution criteria accordingly and reformulated as follows:
*In a third step, we only keep the largest ice shelves. The effective resolution of physical ocean models, i.e. the resolution below which the circulation might not be resolved well, is typically 5 to 10 times the grid spacing (Bricaud et al., 2020). We empirically choose a cutoff at an area of 2500 km$^2$ (i.e. 6.25 $\Delta x$) to be in this range while keeping a sufficiently large number of ice shelves.*

**RC:** **Line 203: "a lot" => "widely"**

AR: Changed.

**RC:** **Eq 22 and others: I don't think you say why some terms are bold.**

AR: Thank you for pointing out that this was not made clear in the text. The bold terms highlight the differences in the terms between the plume formulation as given in Lazeroms et al. (2019) and the new formulation of the plume suggested in this manuscript. We reformulated as follows:
*In the following, we highlight the subscripts in bold when they differ between the formulations.*

**RC:** **Eq 24: for those who are not already very familiar with Lazeroms' method, it might seem strange how you can relate a height difference to length of a plume path without actually integrating the plume equations. Can you give some intuition regarding this definition?**

AR: The plume formulation assumes that the ice shelf base rises from the grounding line up to any point of the ice-shelf base with a constant slope, which brings the formulation to a single dimension along which the plume properties are integrated. This is why height difference and length of the plume path are directly related.

**RC:** **Line 399: Favier et al 2019 is carried out in the ISOMIP+ domain, correct? Should it be surprising that the parameters are not appropriate? Similarly, does the PIGL situation not assume that all ice shelves are flooded with CDW? Should it be any surprise these give high RMSE?**

AR: The aim of all these basal melt parameterisations is to avoid the use of a computation-heavy cavity-resolving ocean model by representing a simplified version of the ocean physics in the cavity beneath the ice shelf. This means, in principle, that these parameterisations should have the same parameters independently of the cavity geometry (ISOMIP+ or realistic) and of the input temperature and salinity (e.g. warm or cold conditions). If one assumes that the parameters should be retuned for each specific situation, we would argue that the aim of the parameterisation is not reached. We nonetheless agree that this is not genuinely surprising.
Still, we did not expect that our results would diverge so largely from the results from Favier et al. (2019). In regard to the "PIGL" parameter, as we point out in the manuscript, the high RMSE is less of a surprise because the parameter was designed to be used with temperature corrections, which we did not apply. Besides, we would like to point out that the PIGL formulation does not assume that ice shelves are flooded with CDW, it

assumes that the melt-temperature quadratic relationship is constrained by the highest melt rates found near Pine Island's grounding line.

**RC:** **Line 425: 5x smaller isn't an order of magnitude**

AR: Thank you for pointing this out. The relationship has changed now due to the new tuning so we compare differently.

**RC:** **Line 425: 3rd column => 2nd column**

AR: We changed the columns of the tables due to the cross-validation results.

**RC:** **Line 458: just wanted to point out I like this comparison.**

AR: Thank you! :)

**RC:** **Line 471: I would add Reese 2018 to this list.**

AR: We added it to the list.

**RC:** **Line 503: looks like an error within the brackets about Jacobs 1992. Also this is a really good point to bring up – and there is more recent work done regarding mode-3 melt (Silvano et al, 2016) which would be good to bring up here and in the discussion.**

AR: Thank you for bringing this up, we looked into it and reformulated as follows:
*This is a result of a seasonal melt cycle, where surface water heated by the atmosphere in summer is transported to the ice-shelf front by tides, eddies, and Ekman transport, leading to seasonally high melt near the front (third mode of melting described by Jacobs et al., 1992; Silvano et al., 2016).*

**RC:** **Line 510: can you elaborate more on your reason for using average over integrated, please. What is the risk of not doing so.**

AR: There is no "risk" per se, it is just a choice to vary our evaluation metrics and evaluate different aspects of the parameterisations. In the first part, we evaluate the integrated melt and this way focus on an ice-shelf-wide metric, which implicitly contains information about the size of the ice shelf, and its variability with time. By evaluating the average melt (over time and space) near the grounding line, we evaluate if, on average, the right melt rate is occurring near the grounding line, independently of the size of the ice shelf. We reformulated as follows to explain our choices more clearly:

*We use average melt rates $m$ (in m ice/yr) rather than integrated melt $M$ (in Gt/yr) used in the previous section to have a complementary metric. With the integrated melt, we focused on an ice-shelf-wide metric, which implicitly contains information about the size of the ice shelf, and its variability with time. By evaluating the average melt over time and space near the grounding line, we evaluate if, on average, the right melt rate is occurring near the grounding line, independently of the size of the ice shelf.*

**RC:** **Lines 544-553: can you explain your experiment more clearly please. I do not understand what you have done here. Is this is new tuning, based on new melt and ocean conditions, or other?**

AR: We apologise that this was not clear enough. We used the tuned parameters but applied them to observational estimates (opposed to simulations in the previous sections) of one ice shelf. So there was no new tuning here. The input temperatures are from Dutrieux et al. (2014), the topography from BedMachine v2 Morlighem (2020); Morlighem et al. (2020). We then compare the resulting melt rates with observational melt rate estimates from Shean et al. (2019) to evaluate the effect of the geometry resolution on the resulting melt rates. We reformulated as follows:
*We use these temperature and salinity profiles and high-resolution topography as input to our parameterisations, and apply our "best-estimate" parameters. We compare the resulting melt patterns (Fig. 8b) to the high-resolution observational melt estimates (Fig. 8a)*

**RC:** **Line 560: why would a sigma coordinate model fare better? Sigma coordinate models have singularities and wild errors where the column goes to zero and the surface gradient is high, i.e. near the grounding line.**

AR: We do not argue that a sigma-coordinate model would fare better, we just think that it might lead to different results. We are aware that they do not work well when there are high slopes but they may better resolve the ice-ocean interface in regions with non-zero columns and low gradients.

**RC:** **Line 562-566: I think you are being too hard on yourself. Given the aims of the study, I'm not sure why you would need to consider evolving cavity geometry.**

AR: We thank you for this encouragement. However, we would like to keep this limitation in the manuscript. As shown in the cross-validation over ice shelves, the parameters unfortunately seem to depend on the cavity geometry. With further climate change and increased sub-shelf melt and retreat, the ice shelf geometry and the cavity itself are prone to change. Taking this evolution into account during tuning or at least during evaluation would have been interesting for future projections.

**RC:** **Line 567-571. These are really good points. You might add a discussion on why Mode 3 melt is important.**

AR: Thank you for pointing that out. We reformulated as follows:
*As we have seen with the Fimbul ice shelf, the use of yearly average input temperature and salinity does not allow the representation of seasonal melt, which takes place at the front of a few ice shelves due to ocean surface warming in summer (Silvano et al., 2016).*

**RC:** **4.1.2 and 4.1.3. These are really interesting experiments but I do not understand the initial procedure at all. If I understand correctly, you are attempting to see how your parameter results would vary if you fit with a subsample of your data, or tweak your data somehow (so, perhaps this addresses my #2 general comment?) but I don't understand lines 573-576. What is meant by bootstrapping? What is the nature of each sample – because I read that each sample represents melt of each shelf in each of the 127 years.. so not a subsample. "What is meant by 36 random sub-samples, with replacement"? It is impossible to interpret the rest of the sections without knowing this.**

AR: The procedure is based on the standard statistical method of bootstrapping, which aims to estimate the uncertainty arising from the limited sample size. As explained for example in Wilks (2006), bootstrapping relies on resampling different samples with the *same* size. This is why it works "with replacement". In our case, this means that the sample always has the size of 35 ice shelves over thirteen 10-year blocks. However, the ice shelves or time blocks are drawn and then replaced into the selection pool before drawing a new one. This means that, in each sample of 35 ice shelves, some can be present several times. We added an explanation of the bootstrapping procedure in Sec. 2.4.1:
*To further estimate the uncertainty around the best-estimate parameters (see Sec. 4), we turn to block-bootstrapping, as cross-validation per definition provides an overview of the generalisation capabilities of the parameterisations but is not the most robust way to estimate the uncertainty in the parameters themselves (Wilks, 2006). Block-bootstrapping consists of iterating the tuning on different resampled datasets of the same size as the original one, here 35 ice shelves x 13 time blocks (Wilks, 2006). To achieve a variety of such samples, we randomly draw an ice shelf and a time block from our data, replace them in the selection pool and repeat the drawing 35 times for the ice shelves and 13 times for the time blocks. This creates a "synthetic" sample of our data with the same sample size as the original sample, which is essential to evaluate uncertainty via bootstrapping. A very large number of synthetic samples, ideally of the order of $10^4$ or higher, can be created this way. The tuning is applied to each of them, resulting in a large distribution of the parameters.*

**RC:** **Line 620-622: I do not follow. The way I interpret Fig 9 is that it is essentially impossible to infer the correct parameter "pair" because they so strongly covary, that depending on the specifics of the tuning data you can get e.g a low C and high gamma$_T^\star$, or vice versa, with either fitting the data reasonably well. But in e.g. a future projection with an ice-sheet model, the difference between using one or the other parameter pair could be quite large. So im not sure simply fixing one of these parameters addresses this difficulty.**

**AR:** The distributions shown in Fig 9 are the results of the bootstrapping method, which aims to give an uncertainty estimate of the parameter(s). The resulting distributions are aimed at scientists who want to sample the uncertainty in basal melt rates introduced by the parameters. Instead of varying both $\gamma_T^\star$ and C, they can now only vary one of them and, using the inferred relationships, simultaneously cover the uncertainty in both parameters. This is one of the main conclusions of this subsection. We reformulated as follows:
*For all complex parameterisations, the distribution of parameters is very large, showing that using constant parameters on the circum-Antarctic scale is challenging and uncertain. To sample this uncertainty in the parameters, we recommend to use the uncertainty intervals around the parameters presented in Table 12 in combination with the empirical relationships provided in Equations (37) to (42). Instead of varying both $\gamma_T^\star$ and C for the box parameterisation and both $C_d^{1/2}\Gamma_{TS}$ and $E_0$ for the plume and PICOP parameterisations, now only one of them needs to be varied and the uncertainty in both parameters can simultaneously be covered. Note, however, that these distributions might not be as robust as the distributions for the parameters of the simple parameterisations (as shown in Table 11) due to the much smaller sample size.*

**RC:** **Lines 700-702. I would think this of CMIP models too. Ill not attempt a list here but there is quite a lot of literature on how global ocean models have difficulty with shelf-offshore exchange.**

**AR:** Thank you for pointing this out. We reformulated as follows:
*This should be the way to derive temperatures from CMIP models which have relatively realistic properties on large portions of the continental shelf around Antarctica (Purich and England, 2021). For coarser ocean models, like those used in climate models of intermediate complexity or the coarsest CMIP models, the parameter values tuned for offshore temperatures might be preferred as these models are too coarse to represent the continental shelf around most of Antarctica.*

**RC:** **Appendix:**

**RC:** **Line 792: you talk about disagreement in melt with Rignot 2013 here but do not show any images.**

**AR:** We included this additional figure showing a comparison between the basal melt rate patterns of our REALISTIC run and the patterns from Rignot et al. (2013).

[Figure]

Figure 1: Mean basal melt rates from (a) Rignot et al. (2013) (2003-2008) and (b) our REALISTIC run (2009-2018) in m ice per year.

**RC:** **Figure B1: add a legend**

**AR:** We clarified the legend.

**References**

Bricaud, C., Le Sommer, J., Madec, G., Calone, C., Deshayes, J., Ethe, C., Chanut, J., and Levy, M.: Multi-grid algorithm for passive tracer transport in the NEMO ocean circulation model: a case study with the NEMO OGCM (version 3.6), Geoscientific Model Development, 13, 5465–5483, #10.5194/tc-2022-3210.5194/gmd-13-5465-2020, 2020.

Dutrieux, P., De Rydt, J., Jenkins, A., Holland, P., Ha, H., Lee, S., Steig, E., Ding, Q., Abrahamsen, E., and Schröder, M.: Strong Sensitivity of Pine Island Ice-Shelf Melting to Climatic Variability, Science, 343, 174–178, #10.5194/tc-2022-3210.1126/science.1244341, 2014.

Favier, L., Jourdain, N., Jenkins, A., Merino, N., Durand, G., Gagliardini, O., Gillet-Chaulet, F., and Mathiot, P.: Assessment of sub-shelf melting parameterisations

using the ocean–ice-sheet coupled model NEMO(v3.6)–Elmer/Ice(v8.3), Geoscientific Model Development, 12, 2255–2283, #10.5194/tc-2022-3210.5194/gmd-12-2255-2019, 2019.

Jacobs, S., Helmer, H., Doake, C., Jenkins, A., and Frolich, R.: Melting of ice shelves and the mass balance of Antarctica, Journal of Glaciology, 38, #10.5194/tc-2022-3210.1017/S0022143000002252, 1992.

Lazeroms, W., Jenkins, A., Rienstra, S., and van de Wal, R.: An Analytical Derivation of Ice-Shelf Basal Melt Based on the Dynamics of Meltwater Plumes, Journal of Physical Oceanography, 49, 917–939, #10.5194/tc-2022-3210.1175/JPO-D-18-0131.1, 2019.

Morlighem, M.: MEaSUREs BedMachine Antarctica, Version 2., #10.5194/tc-2022-3210.5067/E1QL9HFQ7A8M, boulder, Colorado USA. NASA National Snow and Ice Data Center Distributed Active Archive Center., 2020.

Morlighem, M., Rignot, E., Binder, T., Blankenship, D., Drews, R., Eagles, G., Eisen, O., Ferraccioli, F., Forsberg, R., Fretwell, P., Goel, V., Greenbaum, J., Gudmundsson, H., Guo, J., Helm, V., Hofstede, C., Howat, I., Humbert, A., Jokat, W., Karlsson, N., Lee, W., Matsuoka, K., Millan, R., Mouginot, J., Paden, J., Pattyn, F., Roberts, J., Rosier, S., Ruppel, A., Seroussi, H., Smith, E., Steinhage, D., Sun, B., van den Broeke, M., van Ommen, T., van Wessem, M., and Young, D.: Deep glacial troughs and stabilizing ridges unveiled beneath the margins of the Antarctic ice sheet, Nature Geoscience, 13, 132–137, #10.5194/tc-2022-3210.1038/s41561-019-0510-8, 2020.

Purich, A. and England, M.: Historical and Future Projected Warming of Antarctic Shelf Bottom Water in CMIP6 Models, Geophysical Research Letters, 48, #10.5194/tc-2022-3210.1029/2021GL092752, 2021.

Rignot, E., Jacobs, S., Mouginot, J., and Scheuchl, B.: Ice-shelf melting around Antarctica, Science, 341, 266–270, #10.5194/tc-2022-3210.1126/science.1235798, 2013.

Roberts, D., Bahn, V., Ciuti, S., Boyce, M., Elith, J., Guillera-Arroita, G., Hauenstein, S., Lahoz-Monfort, J., Schröder, B., Thuiller, W., Warton, D., Wintle, B., Hartig, F., and Dormann, C.: Cross-validation strategies for data with temporal, spatial, hierarchical, or phylogenetic structure, Ecography, 40, 913–929, #10.5194/tc-2022-3210.1111/ecog.02881, 2017.

Shean, D., Joughin, I., Dutrieux, P., Smith, B., and Berthier, E.: Ice shelf basal melt rates from a high-resolution digital elevation model (DEM) record for Pine Island Glacier, Antarctica, The Cryosphere, 13, 2633–2656, #10.5194/tc-2022-3210.5194/tc-13-2633-2019, 2019.

Silvano, A., Rintoul, S., and Herraiz-Borreguero, L.: Ocean-Ice Shelf Interaction in East Antarctica, Oceanography, 29, 130–143, #10.5194/tc-2022-3210.5670/oceanog.2016.105, 2016.

Wilks, D.: Statistical methods in the atmospheric sciences, 2nd ed., Elsevier, Amsterdam Paris, 2006.

**Author Response to Reviewer #2**

**An assessment of basal melt parameterisations for Antarctic ice shelves**

Burgard, C., Jourdain, N.C., Reese, R., Jenkins, A., Mathiot, P.
*The Cryosphere,* `#10.5194/tc-2022-32`
* * *
**RC: Reviewer Comment**,    AR: Author Response,    *changed manuscript text*

RC:    **Reviewer Summary:**
**For this first time, this paper performs thorough evaluation of a large number of ice-shelf melt parameterizations in a pan-Antarctic context. This is a topic of significant importance to the community, as melt parameterizations allow for Antarctic ice-sheet modeling on time scales that are not currently possible using ice-sheet models coupled to ocean or Earth system models. A number of aspects of this work are novel. First and most importantly, the evaluation is performed on a large number (36) of Antarctic ice-shelves that are fully representative of the continent and which use realistic topography and climate forcing. Second, the rich NEMO model ensemble provides over 500 years of cumulative simulation results, a far more robust dataset than current observations are able to provide. Third, new variants of melt parameterizations (including various ways of including ice-shelf slope dependence and a new approach to plume parameterization) are introduced for inclusion in the evaluation. Fourth, the paper carefully evaluates each parameterization for continental-scale "biases" (relative to the NEMO model), "biases" for individual ice shelves, melt patterns (with an emphasis on grounding lines) and the behavior of parameterizations at high resolution. Finally, the paper includes invaluable recommendations on which parameterizations show the most promise (in a few different categories) and the caveats and best practices for using each, suggested by the evaluation.**

**The paper is well written and arranged. The figures are quite effective, well crafted, and easy to follow. They give the reader a good, intuitive sense of the relative strengths and weaknesses of the different parameterizations in a variety of ways. The numerous tables will help future users of these parameterizations to know the correct coefficient values to use for each, including recommended values of the tunable parameters that best fit the NEMO simulation results.**

AR:    Thank you very much for your constructive and very detailed comments on how to further improve our paper. Here is an overview on the main modifications to the manuscript:

- We made the evaluation of the parameterisations more statistically robust by using cross-validations over time and space and worked on clarifying the goal and

procedure around our choice of RMSE, the cross-validations and bootstrapping by rewriting Sec. 2.4. We also added a discussion about the choice of statistical metrics in Sec. 4.1.2.

- We reconducted all the analysis following your comments and the comments of reviewer #1, i.e. combining Filchner and Ronne into one ice shelf and conduct cross-validations (tuning on one fraction of the sample and evaluation on the other fraction). We also corrected small errors in the code for the tuning of the simple parameterisations and the PICOP parameterisation and in the formulation of the plume parameterisation. Combining Filchner-Ronne and these slight changes in the code lead to different best-estimate parameters compared to the previous version of the manuscript but do not strongly affect the conclusions.

- We reworked on Appendix D, to provide some context for our results on the ice-shelf level, adding a figure showing the reference integrated melt and melt near the grounding line for the individual ice shelves and changing the heatmaps to show the average difference between parameterised and reference melt to show if there is a clear tendency towards under- or overestimation for individual ice shelves.

- We added an Appendix E, which presents a comparison of an RMSE of the melt near the grounding line evaluated on the grid-cell level and an RMSE of the melt near the grounding line evaluated on time- and space-averages for each ice shelf.

Regarding your specific comments, we address them as described in the following.

**RC:** *Major comments*

**RC:** **I do have a few major concerns about the methodology, and addressing these concerns may entail a significant effort by the authors. For that, I am sorry. But I also feel a duty to be particularly careful in evaluating the methodology and suggesting improvements because of the way that the paper is framed: it aims to provide recommended best practices, including preferred coefficients for use in pan-Antarctic modeling using the evaluated parameterizations.**

AR: We appreciate your feedback on the methodology. We agree that this paper recommends best practices and needs to be robust. We thank you for raising concerns which we addressed as follows.

**RC:** **The first concern is one I share with the other reviewer. It would be preferable to tune the parameterizations with one subset of the available data and evaluate it with another, independent subset. Given the more than 500 years of simulation**

**data, I agree that this would seem to be fairly easy to do by using a random subset of years for tuning and the remainder for evaluation. It may be worth adopting this approach in revising the paper.**

AR: We agree that splitting the data into a "tuning" and "evaluation" dataset is worth adopting during revision. However, we want to clarify that we do not have 500 simulation years (as suggested by the reviewer), but "only" 127 years. These 127 years are a result of the accumulation of simulation years from four simulations spanning 30 to 40 years (see Table 1 in the manuscript).

For the revision, as suggested by both reviewers, we conducted a cross-validation of the parameterisation. This means that we conducted the tuning and evaluation several times on different periods to robustly estimate the generalisation performance of each parameterisation. To avoid autocorrelation influencing this cross-validation, we divided the data into 10-year chunks, as the autocorrelation is typically 2 to 3 years in our input temperatures. We then provide best-estimate parameters tuned using the whole sample and investigate the uncertainty around these parameters in Sec. 4. We explain the procedure in detail in Sec. 2.4.1:

*To evaluate the performance of the different parameterisations and the robustness of their tuning, we conduct two variations of leave-one-block-out cross-validation on the minimisation of $RMSE_{int}$, one on the ice shelf dimension and one on the time dimension. This approach consists of dividing the dataset into $N$ blocks, tuning the parameterisation by minimising the evaluation metric on $N-1$ blocks and applying the tuned parameter(s) on the left-out block (Wilks, 2006; Roberts et al., 2017). The procedure is re-iterated $N$ times, leaving out each of the $N$ blocks successively, so that, in the end, each $N$th block has been left out once. All left-out blocks, using the separately tuned parameters, can then be concatenated to form a "synthetically independent" evaluation dataset. Applying the evaluation metric on this evaluation dataset, we can assess how well the parameterisation generalises to blocks not "seen" during tuning. We apply the cross-validations to each input domain (i.e. the 50 km and the offshore domain) separately, using $RMSE_{int}$ as an evaluation metric. On the ice-shelf dimension, we use $N=35$ for the cross-validation over ice shelves. On the time dimension, we divide the years into blocks of approximately 10 years (ten 10-year blocks and three 9-year blocks) to reduce the effect of autocorrelation, which is typically 2 to 3 years in our input temperatures. This results in $N=13$ for the cross-validation over time.*

*Finally, to provide a recommendation for the "best-estimate" parameters to use, we conduct one additional tuning, in which we use all ice shelves and time blocks. To further estimate the uncertainty around the best-estimate parameters (see Sec. 4), we turn to block-bootstrapping, as cross-validation per definition provides an overview of the generalisation capabilities of the parameterisations but is not the most robust way to estimate the uncertainty in the parameters themselves (Wilks, 2006). Block-bootstrapping consists of iterating the tuning on*

*different resampled datasets of the same size as the original one, here 35 ice shelves x 13 time blocks (Wilks, 2006). To achieve a variety of such samples, we randomly draw an ice shelf and a time block from our data, replace them in the selection pool and repeat the drawing 35 times for the ice shelves and 13 times for the time blocks. This creates a "synthetic" sample of our data with the same sample size as the original sample, which is essential to evaluate uncertainty via bootstrapping. A very large number of synthetic samples, ideally of the order of $10^4$ or higher, can be created this way. The tuning is applied to each of them, resulting in a large distribution of the parameters.*

**RC:** **The second major concern is that the metrics used in the paper for evaluating the root-mean-squared "error" (RMSE) relative to the NEMO results, Eqs. (32) and (33), do not seem robust to me, nor are these definitions the standard ones used in Earth system modeling. This is especially unfortunate because the vast majority of the results shown in the paper (from section 2.4 onward) are dependent on these metrics, including the process for identifying the best-fit parameters, the evaluation of parameterizations on the continental scale (both over each ice shelf and near grounding lines), and the methodology for evaluating parameter uncertainty. I am confident that the adoption of a more standard metric is unlikely to change the broad conclusions of the paper but it is hard to see how it would not change specific results such as the recommended values of coefficients used in the parameterizations. For this reason, I ask the authors to strongly consider redoing their analysis with a more robust and widely accepted version of the RMSE metric.**

AR: We deliberately chose non standard metrics instead of the standard melt rate RMSE, but your bad appreciation of our metrics shows that we need to improve the justification for these metrics and provide evidence for their robustness. In short, we want to tune and evaluate the parameterisation in a way that gives the most interesting results for an application in either an ice sheet or an ocean model (or possibly at the interface between both). In the following, we address your points and comments and hope that the more detailed explanation of our choices will convince you.

**RC:** **In my own work and I think in climate modeling broadly, we always use the area-weighted variance to compute the RMSE as follows:**

$$RMSE = \sqrt{\frac{\sum_{i=1}^{N}(m_i - m_{o,i})^2 A_i}{\sum_{i=1}^{N} A_i}} \qquad (1)$$

**where, in the context of this paper, m$_i$ is the melt rate (in whatever units you choose) from the parameterization for sample $i$ (which might be one location and year);**

$m_{o,i}$ **is the "observed" melt rate from NEMO (at the same location and year);** $A_i$ **is the area associated with the** $i$**th sample (this could be a whole ice shelf or an individual NEMO grid cell, depending on how the evaluation is performed); and N is the number of sample points. This metric is robust to different ways of sampling the evaluation points. It works just as well to evaluate the "error" over a single ice shelf as for any number of ice shelves aggregated together into a region. The RMSE values for individual ice shelves can be computed by sampling the pattern at locations across the ice shelf, and then these RMSE values can be combined to get regional- and continental-scale values of the RMSE. This approach is also robust in the sense that including or excluding any number of small ice shelves would have a small impact on the RMSE.**

**Unfortunately, the same is not true for either of the RMSE definitions, Eqs. (32) and (33), proposed in the paper. Instead of weighting by area, these metrics treat each ice shelf (independent of its area) as having equal weight. This approach has at least three problems. First, the metrics are significantly more sensitive to the number of ice shelves under consideration. Considering Eq. (32) first, repeated here:**

$$RMSE_{\text{int}} = \sqrt{\frac{\sum\limits_i^{N_{\text{isf}}} \sum\limits_t^{N_{\text{years}}} (M_{\text{param}}[i,t] - M_{\text{ref}}[i,t])^2}{N_{\text{isf}} N_{\text{years}}}} \tag{2}$$

**if we were to add several ice shelves with both negligible bias in melt flux and negligible area, the RMSE_int would decrease (because we aren't adding significantly to the numerator but we are adding new ice shelves to the denominator). Now, considering Eq. (33):**

$$RMSE_{\text{GL}} = \sqrt{\frac{\sum\limits_i^{N_{\text{isf}}} \sum\limits_n^{N_{\text{simu}}} (m_{\text{GL,param}}[i,n] - m_{\text{GL, ref}}[i,n])^2}{N_{\text{isf}} N_{\text{simu}}}} \tag{3}$$

**In this case, if we were to add several tiny ice shelves with negligible area but significant biases in melt rate, they could significantly increase the RMSE_{GL}. In this way, these metrics are likely fairly sensitive to the number of ice shelves (36) chosen for the study.**

AR: We are aware that the most common way to compute the RMSE in climate modeling is to use the area-weighted variance taking into account each grid cell (as you showed in Eq.(32)). However, very early in our study, we decided that tuning and evaluating at the grid-cell level would not yield the best compromise for the tuning result and for the conclusion drawn from the evaluation. There are several reasons that motivated this choice:

  1. The standard RMSE would be highly biased towards Filchner-Ronne and Ross ice

shelves, which cover a much larger area (many more grid cells) than the others. As a comparison here are the areas of Ross and Filchner-Ronne compared to the third largest ice shelf Amery or to an important ice shelf like Pine Island: Ross = $4.7 \times 10^5$ km$^2$, Filchner-Ronne = $4.2 \times 10^5$ km$^2$, Amery = $5.9 \times 10^4$ km$^2$, Pine Island = $5.7 \times 10^3$ km$^2$. We do not think that tuning on the grid cell level is the most interesting way to go, because Filchner-Ronne and Ross are not necessarily the ice shelves that (1) affect most near-future ice dynamics (Seroussi et al., 2020) or (2) contribute the most to the present-day meltwater release into the ocean (Rignot et al., 2013).

2. The standard RMSE would give the same importance to all grid cells although we know that there are regions which are more or less important for buttressing and therefore for the influence of melt on the ice-sheet evolution (Reese et al., 2018). Many grid cells of Ross and Filchner-Ronne as well as other smaller ice shelves are "passive" and can therefore suffer from acceptable biases that will not significantly affect the ice dynamics. In terms of impacts on the ocean circulation, we believe that getting a correct freshwater budget around Antarctica (i.e. cavity integrated melt) is a priority before getting the correct depth distribution of the freshwater release at a given location.

3. We consider that the melt parameterisations we tune and evaluate are too simple to reproduce all the details of the spatial melt patterns. If they can reproduce the main pattern (e.g. more melt near the grounding line) but the pattern is shifted a little in space, this will result in a high RMSE, penalising a parameterisation that could reproduce some of the complexities of the melt patterns but not at the exact correct location.

4. We only consider relatively large ice shelves (those that are well resolved by NEMO) so there is no issue with the number of very small ice shelves that matter for neither ice nor ocean dynamics. This is an important aspect of our metrics that we will emphasise more in the revised manuscript. Furthermore, our metric is used to inter-compare several parameterisations given this number of ice shelves, and we do not consider it an issue that the metric would have a different meaning if many more small ice shelves were included. Finally, the effect of slight changes in the ice-shelf distribution is accounted for in both our cross-validation and our bootstrapping methods.

In conclusion, we would like to stay with our decision of tuning toward an integrated metric, but worked on improving the part of our manuscript explaining our choice.

We added a discussion about the choice of metric in Sec. 2.4.1:
*Our motivation for re-tuning the parameterisations is to minimise the difference between the melt simulated by NEMO and the parameterised melt (Fig. 3).*

*A common way to conduct this minimisation is to tune towards and evaluate this difference at the grid-cell level by computing an area-weighted root-mean-squared error:*

$$RMSE_{local} = \sqrt{\frac{\sum\limits_{i=1}^{N_{grid\ cells}} (m_{param}[i] - m_{ref}[i])^2 a_i}{\sum\limits_{i=1}^{N_{grid\ cells}} a_i}} \qquad (4)$$

*where $m_{param}[i]$ and $m_{ref}[i]$ are the parameterised and reference melt rates, respectively, in m ice per year in the grid cell $i$, $N_{grid\ cells}$ is the total number of grid cells covered by ice shelves, and $a_i$ is the ice-covered area of the grid cell. However, for this assessment, we argue that minimising Eq. (32) would not yield the best compromise for the tuning and for the conclusion drawn from the evaluation for three reasons. First, this RMSE would be highly biased towards the Filchner-Ronne and Ross ice shelves, which cover a much larger area (i.e. many more grid cells) than the others, while they are not necessarily the ice shelves that (1) affect most near-future ice dynamics (Seroussi et al., 2020) or (2) contribute the most to the present-day meltwater release into the ocean (Rignot et al., 2013). Second, this RMSE gives the same importance to all grid cells although we know that there are regions which are more or less important for buttressing and therefore for the influence of melt on the ice-sheet evolution (Reese et al., 2018). Many grid cells of Ross and Filchner-Ronne as well as other smaller ice shelves are "passive" and can therefore suffer from acceptable biases that will not significantly affect the ice dynamics. Third, we consider that the melt parameterisations we tune and evaluate are too simple to reproduce all the details of the spatial melt patterns. If they can reproduce the main pattern (e.g. more melt near the grounding line) but the pattern is shifted a little in space, this will result in a high RMSE, penalising a parameterisation that could reproduce some of the complexities of the melt patterns but not at the exact correct location. Instead, we use the RMSE between the simulated and parameterised yearly integrated melt ($M$) of the individual ice shelves (in Gt/yr) as follows:*

$$RMSE_{int} = \sqrt{\frac{\sum\limits_{k}^{N_{isf}} \sum\limits_{t}^{N_{years}} (M_{param}[k,t] - M_{ref}[k,t])^2}{N_{isf}N_{years}}} \qquad (5)$$

*where $N_{isf}$ is the number of ice shelves and $N_{years}$ the number of simulated years, and the integrated melt $M$ of ice shelf $k$ (in Gt/yr) is:*

$$M[k] = \rho_i \cdot 10^{-12} \cdot \sum\limits_{j}^{N_{grid\ cells\ in\ k}} m_j \cdot a_j \qquad (6)$$

*$RMSE_{int}$ gives more importance to ice shelves with higher integrated melt and gives the same importance for two ice shelves with the same integrated melt irrespectively of their size, buttressing effect on ice dynamics, or effect on ocean convection. Note that we only consider relatively large ice shelves (those that are well resolved by NEMO) so there is no issue with the number of very small ice shelves that matter for neither ice nor ocean dynamics. In summary, $RMSE_{int}$ is a careful choice following our motivation to make conclusions useful for both ocean and ice-sheet modelling on a circum-Antarctic scale. In terms of impact on the ice sheet, this metric will give more importance to ice shelves with a higher integrated melt, which are currently the most important for the ice-sheet evolution. In terms of impacts on the ocean circulation, we believe that getting a correct freshwater budget around Antarctica (i.e. cavity integrated melt) is a priority before getting the correct depth distribution of the freshwater release at a given location.*

**RC:** **Second, the metric in Eq. (2) here gives significantly more weight to ice shelves with large areas (since $M_i = m_i A_i$, the numerator in this equation is getting weighted by $A_i^2$), while the metric in Eq. (3) here gives significantly less weight to large ice shelves (because $A_i$ does not appear in the numerator at all).**

AR: To assess the parameterisations, we chose to evaluate different aspects of the resulting melt. With the integrated melt, we evaluated the ability of the parameterisations to reproduce the time variability and the integrated value of the melt for a given cavity. With the mean melt rate near the grounding line, we aimed to assess if the parameterisation reproduced the mean pattern well, i.e. if the mean melt rates in the grid cells near the grounding line was on the order of the reference. By averaging the pattern over time and space, we aimed to avoid that this part of the evaluation was biased due to mismatch in time variability (already evaluated before) and slightly shifted patterns in the vicinity of the grounding line. In addition, a metric on the grid-cell level would again be biased towards Filchner-Ronne and Ross ice shelves, which have very long grounding lines. We now explain the procedure and choices more thoroughly in Sec. 3.2.2:

*To quantify the performance of the different parameterisations, in addition to the visual evaluation, we conduct a statistical evaluation of the melt near the grounding line (GL). To do so, we expand the evaluation of the cross-validations conducted in Sec. 3. Instead of inferring the integrated melt, we compute the mean over time and space of the melt in the region defined through the first box of the box parameterisation in the 5-box setup, which represents ≈10% of the ice-shelf area. We use average melt rates $m$ (in m ice/yr) rather than integrated melt $M$ (in Gt/yr) used in the previous section to have a complementary metric. With the integrated melt, we focused on an ice-shelf-wide metric, which implicitly contains information about the size of the ice shelf, and its variability with time. By evaluating the average melt over time and space near the grounding line, we*

*evaluate if, on average, the right melt rate is occurring near the grounding line, independently of the size of the ice shelf. Again, we compute a RMSE over the synthetically independent evaluation dataset*

$$RMSE_{GL} = \sqrt{\frac{\sum_{k}^{N_{isf}} \sum_{n}^{N_{simu}} (m_{GL,param}[k,n] - m_{GL,\,ref}[k,n])^2}{N_{isf} \cdot N_{simu}}} \tag{7}$$

*where $N_{simu}$ is the number of simulations in the ensemble and where $m_{GL}$ for ice shelf $k$ and simulation $n$ is:*

$$m_{GL}[k,n] = \frac{1}{N_{years\,in\,n}} \sum_{t}^{N_{years\,in\,n}} \frac{\sum_{i}^{N_{grid\,cells\,near\,GL\,in\,k}} (m_i \cdot a_i)}{\sum_{i}^{N_{grid\,cells\,near\,GL\,in\,k}} a_i} \tag{8}$$

*Note that we do not take the average over all 127 years at once but average over the individual time periods covered by the four different simulations in our ensemble. This is to avoid taking one single average over four different oceanic states that are not necessarily consistent with each other..*

We also computed the RMSE on the grid-cell level and included the results in the new Appendix section E, which we now point to in the manuscript as follows:

*Similarly to the RMSE of the integrated melt, our choice to evaluate our RMSE on the ice-shelf level, i.e. one average per ice-shelf and not on the grid-cell level, is motivated mainly by two aspects. First, an RMSE evaluating on the grid-cell level might be biased towards Filchner-Ronne and Ross ice shelves, which have the longest grounding lines. Second, such an RMSE would also penalise small shifts in the spatial patterns, resulting in possibly higher RMSE for a parameterisation that could reproduce some of the complexities of the melt patterns near the grounding line but not at the exact correct location. For completeness, we show the results for an RMSE of the melt near the grounding line evaluated on the grid-cell level and discuss it in Appendix E.*

RC: **Third, the definition of an ice shelf is not particularly robust. The Filchner and Ronne Ice Shelves are sometimes considered distinct (as in this paper) and sometimes are combined together. Similarly, the Eastern and Western Ross are sometimes considered separately (though Ross is a single ice shelf here). As grounding lines move, ice shelves that were once distinct may merge, or a single ice shelf may be divided into two. The RMSE metrics from the paper are sensitive to the authors' definition of the 36 ice shelves used in the paper, whereas the metric in Eq. (1) here would not change if an ice shelf were divided into 2.**

AR: We agree that the choice of division of ice shelves is somewhat subjective. From an oceanographic point of view, we agree that combining Filchner and Ronne as one ice shelf

makes more sense. This is what used in the new procedure. It leads to small variations in the best-estimate parameters compared to the original tuning but the parameters remain of a similar order. For example, for the box paramaterisation the previous $\gamma_{TS}^{\star}$ varied between 0.41 and 0.47$\times 10^{-5}$ and now vary between 0.39 and 0.44$\times 10^{-5}$. The previous $C$ varied between 12 and 15$\times 10^{6}$ and now vary between 16 and 21$\times 10^{6}$. Note that in the parameters of the other parameterisations, there are sometimes larger differences to the previous manuscript. As mentioned in the beginning of this response to reviewers, this is because we discovered and corrected small errors in the code for the tuning of the simple parameterisations and in the code of the plume and PICOP parameterisation. Again, we are not concerned by the sensitivity of our metric to the number of ice shelves as long as this number remains constant throughout our intercomparison.

**RC:** **In summary, I do not think robust statistics can be performed when treating ice shelves as distinct entities given equal weight regardless of size. I do not see how a meaningful RMSE in the integrated melt flux can be computed between ice shelves: it would only be meaningful if totalled over all ice shelves in a region (but averaged over samples in time and across ensembles).**

AR: We hope that the responses to the previous comments have addressed this issue of robust statistics. We would like to emphasize that our metrics were not defined for a universal use, but are rather carefully designed for the specific purpose and datasets of our study. We acknowledge that our results could be sensitive to our choices, but we think that our choices are better suited for the goals of this study than the less empirical approach suggested by the Reviewer. We added a section in the discussion (Sec. 4.1.2) to discuss the different approaches to tuning depending on the goal of the tuning:

*The tuning and evaluation of the parameterisations relies heavily on the statistical metrics used. We decided to tune to the integrated shelf melt of the 35 largest ice shelves and evaluated against that metric and the time- and space-averaged melt near the grounding line. As already mentioned in Sec. 2.4.1, this was a careful choice following our motivation to make conclusions useful for both ocean and ice-sheet modelling on a circum-Antarctic scale. For ice-sheet modelling, it is important that ice shelves with a higher integrated melt are more important during tuning, because they are currently the most important for the ice-sheet evolution. For ocean modelling, it is important to get a correct freshwater budget around Antarctica, in the form of integrated melt. However, we acknowledge that the tuning and evaluation can be done differently, depending on the goal of the tuning.*

*If the goal is to match the melt of all ice shelves better, one possibility is to use region-dependent parameters. As our study showed that one constant parameter on the circum-Antarctic scale leads to high RMSE, the parameters could be tuned separately for each region or basin. While this is a practical way to reduce regional biases introduced by the parameters, we argue that the parameters*

*would lose some of their physical meaning in that case and would compensate even more biases that are not directly related to melt physics than when tuned on the circum-Antarctic scale.*

*If the goal is to evaluate the performance of the parameterisation for each grid cell, regardless of where it is situated, one possibility is to use the RMSE on the grid-cell level described in Eq. (32). This would be the most objective and universal evaluation method but we argue that the parameterisations are too simple yet to correctly reproduce the melt patterns at the grid-cell level.*

*Finally, if the goal is to evaluate on the grid-cell level but giving more importance to some cells and less importance to others, one possibility is to use a RMSE weighted with the buttressing flux response numbers presented in Reese et al. (2018). This would give more importance to points that are more important for the buttressing of the ice sheet. As most of the points with high buttressing flux response numbers are situated near the grounding line, we argue that this is close to evaluating the melt near the grounding line.*

RC: **My preference would be that the results of the paper be re-evaluated with the metric in Eq. (1) here instead of Eqs. (32) and (33) in the paper. I realize this is a monumental task. If the authors do not feel that this is necessary, I am willing to hear you out. But I would need to see a strong case for why the existing metrics are more robust and defensible than I currently believe.**
**I want to reiterate that I think this is amazing work! If one takes the metrics for evaluating "biases" as given, the rest of the results follow nicely: the evaluation of each parameterization is remarkably thorough and well executed. The service that this work does for the community is truly commendable, particularly because it gives future model developers not only the tools to make an informed choice among the existing parameterizations but also a good choice of coefficients to use.**

AR: We thank the reviewer for the detailed description of their concerns. After consideration, we sincerely believe that our initial choice is more appropriate for the aforementioned reasons. As a consequence, we re-ran the tuning and evaluation, taking into account that we now combine Filchner-Ronne as one ice shelf and that we conducted several iterations of the tuning and evaluation on different chunks of time and ice shelves, but we kept the initial metrics.

RC: **Below, I have a number of specific comments and suggestions for typographical or grammatical changes. Please do not be intimidated by the number of comments. I have been told by editors in the past that I seem to review papers as if I were a coauthor, which may be true.**

AR: We thank you for the thoroughness of the comments. We deeply appreciate the time you have put into this review.

**RC:** *Specific comments*

**RC:** **l. 7-8: "The box, PICOP parameterization and quadratic parameterizations with slope dependency..." In the context of the abstract (in which these different parameterizations have not yet been introduced), I found it hard to understand what was being described here. The previous sentence does a nice job of introducing what the quadratic parameterization means and what the slope dependence is about. A similar introduction for the box parameterization and PICOP here would be helpful.**

AR: We understand the concern and reformulated as follows:
*The box parameterisation, which separates the sub-shelf circulation into boxes, the PICOP parameterisation, which combines the box and plume parameterisation, and quadratic parameterisations with dependency on the ice slope yield basal melt rates further from the model reference.*

**RC:** **l. 19-20: "...interacting with land or shallow rocks...": I'm not sure I follow what you mean by interacting with shallow rocks. Maybe take this out?**

AR: We reformulated as follows:
*Being several hundreds of meters thick and locally constrained by land or pinning points, they act as natural barriers to restrain the grounded ice-sheet flow into the ocean.*

**RC:** **l. 38: "...and do not resolve ice-shelf cavities...": Should this be "include" instead of "resolve"? It seems like you are referring to model configurations that don't have the cavities at all, rather than that they are there but at too coarse a resolution.**

AR: Thank you for pointing this out. We agree and reformulated as suggested:
*... do not include ice-shelf cavities...*

**RC:** **l. 66-69: When you refer to a "step" here, does each of these correspond to a section of the paper? If so, it might be clearer to replace "step" with "section".**

AR: The first step is one section but the second and third step refer to subsections of the results. We reformulated as follows:
*In Sec. 2, we describe the ensemble of ocean simulations that we use as our "virtual" reality, the data, and the different parameterisations we assess. We also revisit the formulation of several simple parameterisations to emphasize the physical hypotheses behind them. In Sec. 3, we tune the parameterisations*

*using the ocean simulation as a reference to assess how close they can be to the reference and make them comparable between each other and compare the resulting melt rates with the melt rates simulated by the ocean model. In Sec. 4, we investigate uncertainties around the parameters and discuss recommendations and limitations for applications in pan-Antarctic ice-sheet simulations.*

**RC:** **l. 115 "The top boundary-layer thickness is set to 30 m." This likely needs a little more explanation: this is referring to the boundary-layer thickness over which temperature and salinity are averaged for use in the parameterization of sub-ice-shelf turbulent fluxes.**

**AR:** Thank you for pointing this out. We reformulated to clarify that this is the boundary-layer thickness over which temperature, salinity, and ocean velocities $u$ and $v$ are averaged (and we corrected to 20 m, as 30 m was an error). The manuscript now reads as follows: *The top boundary layer is the layer over which temperature, salinity, and the ocean velocities $u_{oc}$ and $v_{oc}$ (where $U = \sqrt{u_{oc}^2 + v_{oc}^2}$) are averaged. Its thickness is set to 20 m.*

**RC:** **l. 147: "basal mass loss" might be better as "basal mass flux" or something like that because usually "mass loss" is used in our field to refer to the uncompensated loss of mass from the ice sheet that contributes to sea-level rise.**

**AR:** Thank you for pointing this out. We replaced by "integrated melt rate" to stay consistent with the rest of the manuscript.

**RC:** **l. 188-191: Here, a bunch of physical constants and variables are introduced for the second time (they were introduced on l. 104-115). I think the only new ones here are $\Gamma_{TS}$, $T_i$ and $U$. Also, it seemed to me that $U$ might be the same as $u_{oc}$ introduced above. If so, maybe use the same symbol for both. If not, maybe clarify the distinction.**

**AR:** Thank you for pointing out that this was confusing. We re-introduced them because the first introduction of the variables happened well earlier. We went again throught the manuscript and now every variable is only introduced once.

**RC:** **l. 213: "where the subscript oc denotes the far-field temperature and salinity that is advected into the cavity". This seems like a different definition than on l. 105: "$T_{oc}$ and $S_{oc}$ are the temperature and salinity averaged over a boundary layer below the ice shelf." It would be helpful if the oc subscript meant the same thing throughout the manuscript. Maybe another subscript like ff could be used for far-field quantities (or bl for boundary-layer quantities)?**

AR: Here also, thank you for this suggestion. We now use the subscript "loc" for far-field properties extrapolated to the local ice draft depth, as this is what we use in the description of the plume parameterisation anyway.

RC: **l. 217: The liquidus slope has already been defined on l. 108.**

AR: We removed the second introduction of the liquidus slope.

RC: **l. 240-240: "However, through the resulting circulation, the temperature at one point of the ice shelf cavity is not necessarily independent from the temperature at the other points of the ice shelf cavity." This may be correct, but I understood the quadratic dependency a little differently. I thought the concept was that the overturning itself was driven by the aggregate effect of the thermal forcing (so that the strength of the overturning circulation would be proportional to the mean thermal forcing in the cavity). Thus, the local velocity is represented in terms of the amount of heat available for melting and there isn't necessarily any implication that the temperatures at different parts of the ice shelf are correlated to one another.**

AR: We agree that this was not clearly formulated. Our aim was more to explain the non-local aspect than the quadratic aspect. We reformulated as follows:
*So far, we assumed a local geostrophic balance due to a gradient between the ambient ocean properties and the local properties of the top boundary layer influenced by melting. However, the melt-induced circulation takes place at the scale of the cavity with non-local effects of melt rates (Jourdain et al., 2017).*

RC: **Eq. (17): It seems like some explanation of the bracket < > notation is needed.**

AR: We thought this was a notation commonly used. We added a brief explanation:
*...where the $\langle \cdot \rangle$ notation denotes a spatial average.*

RC: **Eqs. (20) and (21): I think these should be moved after Eq. (24) (with Eq. (20) after Eq. (21)) because they don't currently have any context where they are.**

AR: We moved them as suggested.

RC: **Eqs. (23) and (25) are identical, right? If so, I don't think it's necessary to have Eq. (25), you can just state that $M_2$ is still given by Eq. (23).**

AR: They are identical but the characteristic length scale $x$ used in both cases is different. We re-arranged the text and equations in Sec. 2.2.2 to avoid repeating Eq.(23) twice and further clarifying the different subscripts.

**RC:** l 311-312: "If one of the boxes has an area of zero...": It isn't clear to me how this could happen. Is this just for an ice shelf that is poorly resolved so that there aren't enough grid cells for each box to get some?

**AR:** Yes, this is how it can happen that a box has an area of zero. We reformulated as follows: *If one of the boxes has an area of zero because the resolution of the ice shelf is too low,...*

**RC:** l. 313: "are non-zero": Should this be "have a non-zero area?"

**AR:** Yes, we reformulated as suggested.

**RC:** l. 349-350: "We use yearly mean profiles as the residence time of water in ice shelf cavities might be longer than a month for some cavities." You explain this more in the discussion but it might be worth giving it more context here, too. The parameterizations do not include the finite advection time from the ice-shelf front to the ice-shelf base, meaning they are missing important delays that are present in the physical system. This means there is little hope of capturing seasonal melting in the proper phase.

**AR:** We added a sentence to briefly mention this issue here as well: *Note that this also means that we assume that the advection between the shelf front and the grounding line takes less than a year and we ignore possible longer advection time (see Sec. 4.1.1 for further discussion).*

**RC:** l. 431: "We therefore only show results for heterogeneous boxes hereafter." This is not true in the next subsection when you talk about PICOP, so maybe change "hereafter" to "in the remainder of this subsection."

**AR:** Thank you for pointing this out. We reformulated as follows: *We therefore only show results for heterogeneous boxes when we discuss the box parameterisation hereafter.*

**RC:** l. 458-460: "To put these RMSE in context, the integrated ice-shelf melt averaged over all ice shelves and simulation years in the reference is 38 Gt/yr, which means that even for the better-performing parameterisations, the RMSE remains very high compared to the reference melt as such."
I have concerns about this concept, related to my concerns about the metrics in Eqs. (22) and (23). I do not think it is meaningful that the total Antarctic melt flux ( 1500 Gt/yr) averaged over 36 ice shelves (an arbitrary number related to the

**resolution of the NEMO simulations used for reference) comes out to 38 Gt/yr. The different ice shelves have such different areas and melt rates that this average isn't informative. Given that I have problems with the metric from Eq. (22) that you're given context to, this might be a moot point but if you want to give context to what 40 Gt/yr is like, maybe just point out an ice shelf by name that has about that much melting.**

AR: Thank you for expressing your concern. We have chosen to keep this number (39 Gt/yr) in the main manuscript to give context for what the values of the RMSE represent. However, we now also leave the reader with additional information in Appendix D to get an idea of how this number compares to the melt of the individual ice shelves. We have added a figure showing the mean and standard deviation of the reference integrated melt and melt near the grounding line for the individual ice shelves, and the heatmaps now show the average difference between parameterised and reference melt (instead of the RMSE) for the individual ice shelves to show if there is under- or overestimation of the melt on average.

RC: **l. 504-505: "...likely because they are all forced by yearly ocean temperatures." This seems to assume that the parameterizations would be able to get the Mode 3 melt if they were forced with seasonal or monthly temperatures. As you state later, they are unlikely to have the correct processes or the correct seasonal phasing to get these melt rates right even if temperature forcing were provided on a shorter time interval.**

AR: We do not completely agree. If we used monthly temperature and salinity profiles as input, it would be possible to represent mode 3 melt as long as the residence time of water is significantly shorter than the seasonal period. This is likely the case for a majority of ice shelves and could be valid even for larger cavities as mode 3 melt usually occurs close to the ice-shelf front. We have added the following:
*If we used monthly temperature and salinity profiles as input, it should be possible to represent this seasonal melt ("mode 3" melt), except for the box and PICOP parameterisations. This would be possible both for relatively small cavities, for which the residence time of the water in the cavity is significantly shorter than the seasonal period, and for larger cavities, as seasonal melt usually occurs relatively close to the ice-shelf front. However, while going to shorter time scales would improve the representation of mode 3 melt close to the ice shelf front, it would probably require accounting for the possible time lag between the input forcing entering the cavity and the occurrence of the melt near the grounding line (Holland, 2017), which has been ignored in this study.*

RC: **Figs. 4 and 7: I appreciated this visual representation of the RMSE – it was very intuitive.**

AR:   Thank you!

RC:   **Figs. 5 and 6: These are wonderfully done and very clear!**

AR:   Thank you!

RC:   **l. 575-576: "...and is composed of 36 random samples, with replacement, of the
      different ice shelves." Do I understand correctly that you double or triple (or what-
      ever) count some ice shelves and omit other ice shelves in this process? That seems
      like an uncommon practice and that the approach (suggested by reviewer 1) of us-
      ing only some of the years for training and holding back others for validation would
      be a more standard method for testing the robustness and parameter uncertainty.**

AR:   As explained for example in Wilks (2006), bootstrapping relies on resampling different
      samples with the *same* size. This is why it works "with replacement". Bootstrapping
      gives an estimate of the parameter uncertainty. The cross-validation, which is the pro-
      cedure used in the iterations of tuning (holding back part of the sample to evaluate on
      this part), provides an estimate of the generalisation performance of the parameterisation.
      These are two different conclusions from a statistical point of view.
      We agree that the method presented here only shows the uncertainty introduced by the
      choice of included ice shelves. We will include the time-chunk dimension in the resam-
      pling to cover the full uncertainty of the parameters.
      We have reformulated the explanation of the procedure and included it in Sec. 2.4.1:
      *To further estimate the uncertainty around the best-estimate parameters (see
      Sec. 4), we turn to block-bootstrapping, as cross-validation per definition pro-
      vides an overview of the generalisation capabilities of the parameterisations
      but is not the most robust way to estimate the uncertainty in the parameters
      themselves (Wilks, 2006). Block-bootstrapping consists of iterating the tuning on
      different resampled datasets of the same size as the original one, here 35 ice
      shelves x 13 time blocks (Wilks, 2006). To achieve a variety of such samples, we
      randomly draw an ice shelf and a time block from our data, replace them in the
      selection pool and repeat the drawing 35 times for the ice shelves and 13 times
      for the time blocks. This creates a "synthetic" sample of our data with the same
      sample size as the original sample, which is essential to evaluate uncertainty
      via bootstrapping. A very large number of synthetic samples, ideally of the order
      of $10^4$ or higher, can be created this way. The tuning is applied to each of them,
      resulting in a large distribution of the parameters.*

RC:   **Fig. 9a. On the light gray curve it says "fit on pairs with E0 < 50" but in the text
      describing Eq. (34), I didn't see anything stating that it was for E0 < 50. Is (34) not
      for the gray curve? Is this description simply missing from the text?**

AR: Yes, this is missing, thank you for pointing it out. We reformulated as follows:
*The resulting relationship, ignoring $E_0$ above $50 \times 10^{-2}$ (Fig. 9a, grey line)...*

RC: **l. 617-619: "Note that some of the higher values of $C_d^{1/2}\Gamma_{TS}$ and $E_0$ are several orders of magnitude higher than expected (see e.g. Table 12), which we cannot explain." This seems odd. Don't you have constraints on the parameters to prevent them from varying outside a physically likely range?**

AR: We agree that this is odd and is not satisfying. Introducing lower constraints led to parameters "stuck" at the constraint, showing that the lowest RMSE was reached with unrealistically high parameter values. We see no other option than acknowledging it and added the following:
*We do not recommend using these parameters as they are not physically plausible.*

RC: **l. 648: "The slope-dependent parameterisation is nonetheless relatively good for cavity melt rates..." Do you mean "...good for capturing melt patterns" or something like that?**

AR: Yes, this is what we meant. We reformulated as follows:
*...relatively good for capturing integrated melt rates...*

RC: **l. 744-745: "...with RMSEs on the same order as or even larger than the reference value." This is another reference to ice-shelf-by-ice-shelf statistics that I think needs to be rethought in terms of area-weighted statistics.**

AR: Again, we point to Appendix D:
*For context for these values, the mean reference melt near the grounding line is 0.45 m ice/yr and the values for the individual ice shelves is shown in the right panel of Fig. D1.*

RC: **Appendix D: Could you say how the RMSE for the heatmaps are defined? Are these the same as Eq. (32) but with Nisf = 1?**

AR: Yes, thank you for pointing out that this is not clear. We now added a small introduction to the figures in Appendix D:
*Our study focusses on the circum-Antarctic performance of the parameterisations. The following figures provide a brief overview of the resulting performance on the individual ice shelves. Figure D1 shows the reference mean integrated melt and mean melt near the grounding line for the individual ice shelves in our NEMO ensemble. Figure D2 shows the average difference between the parameterised and reference integrated melt in the cross-validation, i.e. the average under- or*

*overestimation of the melt by each parameterisation for a given ice shelf when it was tuned over all other ice shelves. Figure D3 shows the average difference between the parameterised and reference melt near the grounding line in the cross-validation, i.e. the average under- or overestimation of the melt by each parameterisation for a given ice shelf when it was tuned over all other ice shelves.*

**RC:** *Typographical and Grammatical Suggestions*

**RC: l. 11: "Additionally to..." should be "In addition to..."**

 AR: Thanks, was changed.

**RC: l. 28: "uncertainty source" should be "source of uncertainty"**

 AR: Thanks, was changed.

**RC: l. 140: Probably remove "For example", since this sentence didn't clearly follow (at least for me) as an example of the changed parameters not having a significant impact on the physical ocean.**

 AR: Maybe this was not clear enough then. We mean that the changed parameters don't have a significant impact on the physical link between the ocean in front of the ice shelf and the basal melt rates. They have, however, an impact on the physical ocean *outside* of the cavity. So this "for example" actually should follow the sentence before. We reformulated the preceding sentence as follows:
*None of the changed parameters has a significant impact on the physics of ocean—ice-shelf interactions, and they mostly change the physical ocean properties outside the cavities.*

**RC: l. 204: "from an ice shelf to another" should be "from one ice shelf to another".**

 AR: Thanks, was changed.

**RC: l. 252: "(based on Sec. 2.2.2)" might be more natural as "(as described in the next section)".**

 AR: Thanks, was changed.

**RC: l. 288-291: This is a lot in one sentence and gets a bit confusing. It might be clearer if you break the details of each subscript into its own sentence.**

AR: Thank you for pointing it out. We have rewritten the whole description of the subscripts:
*The two versions mainly differ in the definition of the grounding line depth and of the input temperature $T$, salinity $S$ and slope $\theta$ used to compute $M_1$ and $M_2$. The grounding line can be the effective grounding line depth (subscript gl) or the deepest grounding line point (subscript deepest gl). The hydrographic input properties and the slope can be taken on the cavity scale (subscript cav), on the local scale (subscript loc), or on the upstream scale (subscript ups). The cavity scale means that the far-field temperature and salinity are extrapolated to the ice draft depth for each point and then averaged over the ice shelf area, and that one single slope is estimated for the whole cavity as described in Sec. 2.2.1. The local scale means that the far-field properties are extrapolated to the local ice draft depth and that we use the local slope as defined in Lazeroms et al. (2018). Note that this definition of local slope differs from the definition in the simple parameterisations (Sec. 2.2.1), so we add "laz" (for "Lazeroms") to the subscript. The effective grounding line depth and the local slope are computed as described in Lazeroms et al. (2018), evaluating possible plume origins in 16 directions for each ice shelf point and averaging the local slope and grounding line depth, respectively, over the plausible plume origin directions. The upstream scale means that we average, for each point of the ice shelf, the portion of the far-field input profiles located between the local ice draft depth and local effective grounding line depth. For the upstream slope, we take, at each point, the angle opened by the effective grounding line depth, the local ice draft depth, and the shortest distance between the grounding line and the given point. In the following, we highlight the subscripts in bold when they differ between the formulations.*

RC: **l. 308, 314, 315, 437, 444, 446, 575, 589, 666, 667, 747, Appendix C, Table C1: The word "amount" is used in several places where "number" is correct (because the object being referred to is countable, rather than indefinite): "number of boxes", "number of data points", "number of high-resolution ocean simulations", etc.**

AR: Thank you for this very helpful comment. We corrected this mistake.

RC: **l. 381-382: "two constraints are additionally needed" would be better as "two additional constraints are needed"**

AR: Thanks, was changed.

RC: **l. 399, 434: "(2020)(Table 4, 1st and 3rd columns)" and similar: This is very picky on my part but if there's a way to avoid back-to-back parentheses like here, I would prefer it.**

AR: Thanks, we thought about it but did not find a more elegant solution.

**RC:** **l. 404-405: "The new tuning only slightly achieves to reduce the RMSE further" would be better as "The new tuning achieves only a slight ruther reduction in RMSE"**

AR: Thanks, was changed.

**RC:** **l. 414-415 and 449: "input properties are of a similar order of magnitude" and "all of a similar order of magnitude"**

AR: Thanks, was changed.

**RC:** **l. 438: "...about twice as high as the..."**

AR: Thanks, was changed.

**RC:** **l. 454: "Slightly above..." should be "At slightly above..."**

AR: Thanks, was changed.

**RC:** **l. 460, 517, 565: "... to the reference melt as such." and "...the melt patterns as such". The phrase "as such" is used in a few places in the paper where I don't fully understand it. Maybe you mean "itself"? Maybe this can just be omitted?**

AR: We removed these formulations.

**RC:** **l. 506: "additionally" should be "in addition"**

AR: Thanks, was changed.

**RC:** **l. 519: "This can be explained by punctual very strong melt". I do not know for sure what is meant here by "punctual". This usually means "on time".**

AR: Thank you for pointing this out, this was a wrong translation from our French mind then. We meant "in a very small region" (like a point). Was changed.

**RC:** **l. 521: "should therefore be kept" would be a bit better as "is important to keep" and "On the opposite" should be "In contrast".**

AR: Thanks, was changed.

**RC:** l. 527: "...in a more consistent way and with larger...

 AR: Thanks, was changed.

**RC:** l. 543: "emulates well NEMO" should be "emulates NEMO well"

 AR: Thanks, was changed.

**RC:** l. 549-550: "...and therefore rather is a result of the circum-Antarctic tuning rather than of the resolution"

 AR: Thanks, was changed.

**RC:** l. 550 "...on the contrary" should be "in contrast"

 AR: Thanks, was changed.

**RC:** l. 568: "...require to account for..." should be "...require accounting for..."

 AR: Thanks, was changed.

**RC:** l. 616: "This relation is different from Eq. (34)..."

 AR: Thanks, was changed.

**RC:** l. 627: "On the opposite" should probably be "In contrast"

 AR: Thanks, was changed.

**RC:** l. 637: "Especially" should be something like "In particular"

 AR: Thanks, was changed.

**RC:** l. 648: "...used as end members" might be "...used to generate end members..."

 AR: Thanks, was changed.

**RC:** l. 660-661: "We therefore suggest, when possible, trying out..."

 AR: Thanks, was changed.

**RC:** **l. 694: "...any improvement in either of the two..."**

AR: Thanks, was changed.

**RC:** **l. 716: "...take into account, to some extent, the horizontal..."**

AR: Thanks, was changed.

**RC:** **l. 719: missing a comma after "geometry"**

AR: Thanks, was changed.

**References**

Holland, P. R.: The transient response of ice shelf melting to ocean change, Journal of Physical Oceanography, 47, 2101–2114, #10.5194/tc-2022-3210.1175/JPO-D-17-0071.1, 2017.

Jourdain, N., Mathiot, P., Merino, N., Durand, G., Le Sommer, J., Dutrieux, P., Spence, P., and Madec, G.: Ocean circulation and sea-ice thinning induced by melting ice shelves in the Amundsen Sea, J. Geophys. Res., 122, 2550–2573, #10.5194/tc-2022-3210.1002/2016JC012509, 2017.

Lazeroms, W., Jenkins, A., Gudmunsson, G., and van de Wal, R.: Modelling present-day basal melt rates for Antarctic ice shelves using a parametrization of buoyant meltwater plumes, The Cryosphere, 12, 49–70, #10.5194/tc-2022-3210.5194/tc-12-49-2018, 2018.

Reese, R., Gudmundsson, G., Levermann, A., and Winkelmann, R.: The far reach of ice-shelf thinning in Antarctica, Nature Climate Change, 8, 53–57, #10.5194/tc-2022-3210.1038/s41558-017-0020-x, 2018.

Rignot, E., Jacobs, S., Mouginot, J., and Scheuchl, B.: Ice-shelf melting around Antarctica, Science, 341, 266–270, #10.5194/tc-2022-3210.1126/science.1235798, 2013.

Roberts, D., Bahn, V., Ciuti, S., Boyce, M., Elith, J., Guillera-Arroita, G., Hauenstein, S., Lahoz-Monfort, J., Schröder, B., Thuiller, W., Warton, D., Wintle, B., Hartig, F., and Dormann, C.: Cross-validation strategies for data with temporal, spatial, hierarchical, or phylogenetic structure, Ecography, 40, 913–929, #10.5194/tc-2022-3210.1111/ecog.02881, 2017.

Seroussi, H., Nowicki, S., Payne, A., Goelzer, H., Lipscomb, W., Abe-Ouchi, A., Agosta, C., Albrecht, T., Asay-Davis, X., Barthel, A., Calov, R., Cullather, R., Dumas, C.,

Galton-Fenzi, B., Gladstone, R., Golledge, N., Gregory, J., Greve, R., Hattermann, T., Hoffman, M., Humbert, A., Huybrechts, P., Jourdain, N., Kleiner, T., Larour, E., Leguy, G., Lowry, D., Little, C., Morlighem, M., Pattyn, F., Pelle, T., Price, S., Quiquet, A., Reese, R., Schlegel, N.-J., Shepherd, A., Simon, E., Smith, R., Straneo, F., Sun, S., Trusel, L., van Breedam, J., van de Wal, R., Winkelmann, R., Zhao, C., Zhang, T., and Zwinger, T.: ISMIP6 Antarctica: a multi-model ensemble of the Antarctic ice sheet evolution over the 21st century, The Cryosphere, 14, 3033–3070, #10.5194/tc-2022-3210.5194/tc-14-3033-2020, 2020.

Wilks, D.: Statistical methods in the atmospheric sciences, 2nd ed., Elsevier, Amsterdam Paris, 2006.

---

## Author Response (AR2)

*EC: Editor comment*, AR: Author response

Dear Editor and reviewers,

We deeply thank you for the time and effort put into the review process of this paper. Here are the final responses to the last comments.

*EC: L76 remove "very"*
AR: Done

*EC: L90 " nonlinear " surface. Do you mean "curved" surface or something similar? Nonlinear has a mathematical connotation for me which I don't think is what is meant here.*
AR: « non-linear » free surface is a term widely used in the ocean modelling community: a non-linear free-surface implementation allows one to deal with large amplitude free-surface variations relative to the vertical resolution (see NEMO Team, 2019).

*EC: L122 are u_oc and v_oc the horizontal velocity components?*
AR: Yes, we have clarified:
"and the ocean horizontal and vertical velocity components $u_\text{oc}$ and $v_\text{oc}$"

*EC: L166 listed, e.g., in (commas)*
AR: Done

*EC: Consider making this paper more accessible by introducing all variables in a table or even up front (sort o an extension of table 2.) Clearly all variables are defined in the text, but some readers may only read details for a specific parameterization.*
AR: Thank you for this interesting suggestions. We have now added such a table in the Appendix. We argue that everything is explained in the text and the table is so large that it would break the flow of the paper, so putting it in the Appendix is more appropriate. We mention it in the beginning of Sec. 2.2:
"As a range of slightly different definitions of the variables are introduced, we provide two tables in Appendix C summarizing the main variables and different subscripts used in the following description."

*EC: I wonder if the melt $m$ should rather be marked with $\dot{m}$ and consistently be referred to as "(basal) melt rate" (units m ice per s)*
AR : In previous literature, \dot{m} and {m} have been used nearly interchangeably. We prefer sticking with m for the melt rate. We added "rate" to all occurrences where we mentioned melt in units m ice per s.

*EC: Eqs (20) and (26) appear to have some randomly placed $\mathbf{}$ notation. This bold fonts should be used for vectors. Double check.*
AR: The bold terms highlight the differences between the formulations, as explained in the text just before. We would like to keep it this way. We argue that it is clear that this is not describing vectors in this particular case.

*EC : I wonder if it would be better to have Fig. 3 as Fig. 1 as it attempts to explain the main approach of the paper.*
AR: We have now moved this figure to the beginning of Sec. 2.

*EC : Eq (34) are the $\cdot$ needed?*
AR: Ok, we removed it.

*EC : Fig. 5 have you tried showing the difference relative to the reference? This could possibly be more informative.*
AR: We have chosen not to show the difference for 2 reasons: (1) it is difficult to represent difference between two quantities that are best represented on a logarithmic scale and (2) we lose the information whether a negative difference stands for underestimated melt or for the difference between freezing and melting.

*EC : L633 here you refer to it as "melt rates". I suggest to do this everywhere.*
AR: We went through the manuscript to change this where appropriate.

*EC : Eq (36) remove \cdot*
AR: Done

*EC : L855 "A higher number of boxes leads to a slightly lower RMSE." Is this surprising, given that more free parameters are available to reduce the RMSE? I wonder if overfitting can play a role here.*
AR: Independently of the number of boxes, there are only two free parameters to tune, so we do not think that there is a risk of overfitting. Actually, we find it more surprising that it does not lower the RMSE even more, because having more boxes leads to a more heterogeneous pattern that might capture better spatial heterogeneities in the melt rates.

*EC : L866 "On the one ..." not (One)*
AR : Done